# Survey of Video Diffusion Models:
# Foundations, Implementations, and Applications

**Yimu Wang**[*]                                                                                    *yimu.wang@uwaterloo.ca*
*University of Waterloo*

**Xuye Liu**[*]                                                                                       *xuye.liu@uwaterloo.ca*
*University of Waterloo*

**Wei Pang**[*]                                                                                       *w3pang@uwaterloo.ca*
*University of Waterloo*

**Li Ma**[*]                                                                                        *li.ma@scanlinevfx.com*
*Netflix Eyeline Studios*

**Shuai Yuan**[*]                                                                                     *shuai@cs.duke.edu*
*Duke University*

**Paul Debevec**                                                                                   *debevec@scanlinevfx.com*
*Netflix Eyeline Studios*

**Ning Yu**[†]                                                                                    *ning.yu@scanlinevfx.com*
*Netflix Eyeline Studios*

**Reviewed on OpenReview:** *https://openreview.net/forum?id=2ODDBObKjH*

## Abstract

Recent advances in diffusion models have revolutionized video generation, offering superior temporal consistency and visual quality compared to traditional generative adversarial networks-based approaches. While this emerging field shows tremendous promise in applications, it faces significant challenges in motion consistency, computational efficiency, and ethical considerations. This survey provides a comprehensive review of diffusion-based video generation, examining its evolution, technical foundations, and practical applications. We present a systematic taxonomy of current methodologies, analyze architectural innovations and optimization strategies, and investigate applications across low-level vision tasks such as denoising and super-resolution. Additionally, we explore the synergies between diffusion-based video generation and related domains, including video representation learning, question answering, and retrieval. Compared to the existing surveys (Lei et al., 2024a;b; Melnik et al., 2024; Cao et al., 2023; Xing et al., 2024c) which focus on specific aspects of video generation, such as human video synthesis (Lei et al., 2024a) or long-form content generation (Lei et al., 2024b), our work provides a broader, more updated, and more fine-grained perspective on diffusion-based approaches with a special section for evaluation metrics, industry solutions, and training engineering techniques in video generation. This survey serves as a foundational resource for researchers and practitioners working at the intersection of diffusion models and video generation, providing insights into both the theoretical frameworks and practical implementations that drive this rapidly evolving field. A structured list of re-

---

[*]These authors contributed equally to this work.
[†]Corresponding author.

lated works involved in this survey is also available on GitHub: https://github.com/Eyeline-Labs/Survey-Video-Diffusion.

## Contents

# 1 Introduction

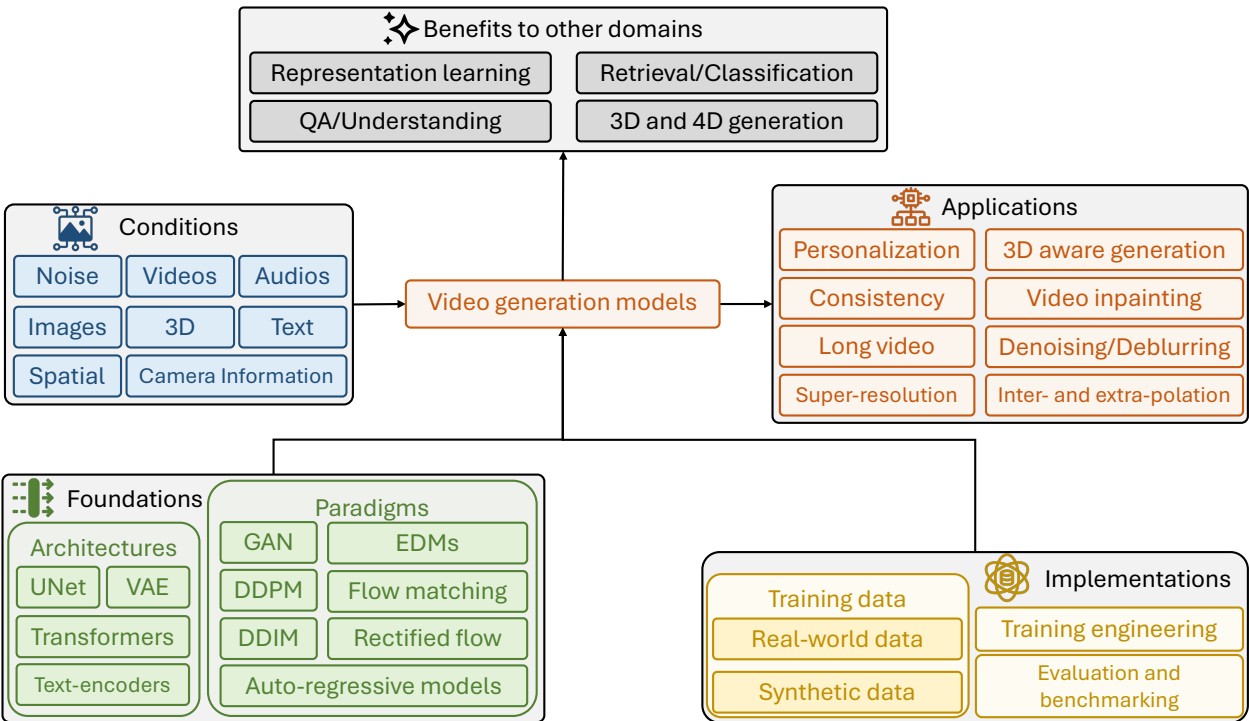

Figure 1: Overview of video generation methods. Generally, the input conditions can be noises, images, videos, audios, texts, and 3D point clouds. The architectures (UNet, VAE, and/or Transformers) are trained using GAN or diffusion models with training data of real-world data or synthetic data with different paradigms. The applications are in several folds, *e.g.*, video personalization, consistency-aware generation, and long video generalization. On the other side, video generation models can also benefit other video tasks, such as, video retrieval, understanding, and representation learning.

Video generation (Ren et al., 2024; Zheng et al., 2024c; Davtyan & Favaro, 2022) has emerged as a critical and transformative technology in recent years. The ability to generate high-quality, realistic videos has numerous applications, from entertainment and advertising (Wang & Shi, 2023) to virtual reality (Hu et al., 2021) and autonomous systems (Zhou et al., 2024c), further enabling enhanced user experiences, cost-effective content creation, and new avenues for creative expression.

In the past few years, video generation (Clark et al., 2019; Aldausari et al., 2022; Hong et al., 2022) has experienced remarkable progress, particularly with the adoption of Generative Adversarial Networks (GANs, (Goodfellow et al., 2014)). Researchers have implemented various strategies to improve the temporal coherence (Chai et al., 2023), realism, and diversity of generated videos. Despite these advancements, GAN-based methods often face challenges in training stability and achieving high-quality outputs consistently. The introduction of diffusion models (Ho et al., 2020; Nichol & Dhariwal, 2021; Sohl-Dickstein et al., 2015) has revolutionized this area by offering a probabilistic framework that overcomes many limitations of GANs. Diffusion-based models (Kwak et al., 2024; Chai et al., 2023; Wang & Yang, 2024; Ho et al., 2022c) have demonstrated superior performance in generating temporally consistent and visually compelling videos, motivating further research in the domain.

However, diffusion-based video generation presents several fundamental challenges. A primary concern lies in ensuring motion consistency across frames, a critical factor for producing temporally consistent and realistic videos. Moreover, generated videos must adhere to physical rules, such as accurate object dynamics and

environmental interactions, to maintain realism. Long video generation poses another challenge, requiring models to handle complex temporal dependencies over extended sequences. Furthermore, the computational demands of training diffusion-based models are substantial, often leading to inefficiencies that limit scalability. Similarly, inference acceleration remains a severe issue, as generating high-quality videos in real-time is critical for many applications. Beyond these technical challenges, ethical considerations also play a pivotal role, including mitigating biases in generated content and preventing the creation of harmful or misleading visual outputs.

In response to the rapid advancements and emerging challenges of diffusion-based video generation, this survey provides a systematic analysis of current methodologies, recent advances, and future directions in the field.

**Our contributions.** This survey presents a comprehensive analysis of diffusion-based video generation, focusing on both technical foundations and practical applications. While existing surveys (Lei et al., 2024a;b; Melnik et al., 2024; Cao et al., 2023; Xing et al., 2024c) have addressed specific aspects of video generation, such as human video synthesis (Lei et al., 2024a) or long-form content generation (Lei et al., 2024b), our work provides a broader, more updated, and more fine-grained perspective on diffusion-based approaches. Compared to related surveys (Xing et al., 2024c; Melnik et al., 2024), we offer more extensive coverage of diffusion models and their applications with a detailed overview of datasets, evaluation metrics, industry solutions, and training engineering techniques for video generation. The main contributions of this paper are as follows:

- To the best of our knowledge, this is the most comprehensive survey on diffusion-based video generation, including model paradigms, learning foundation, implementation details, applications, and relations to other domains.

- Compared to related surveys, our survey covers a broader scope of diffusion models and their applications with a detailed overview of datasets, evaluation metrics, industry solutions, and training engineering techniques for video generation.

The rest of the survey is organized as follows: Section 2 lays the groundwork by exploring foundational concepts, including video generation paradigms such as GAN-based models, auto-regressive models, and diffusion models. Section 3 focuses on implementation, addressing practical aspects such as datasets, training engineering techniques, and evaluation metrics, along with benchmarking findings to highlight model performance. Section 4 covers diverse applications, including conditional generation tasks, enhancement methods such as video denoising, inpainting, interpolation, extrapolation, and super-resolution, as well as personalization, consistency, long-video generation, and emerging 3D-aware diffusion models. Finally, Section 5 discusses the benefits of diffusion-based video generation to other domains, such as video representation learning, retrieval, QA, and 3D and 4D generation, emphasizing its broader impact in advancing related fields. This structure provides a comprehensive and cohesive understanding of diffusion-based video generation, from foundational principles to advanced applications.

## 2 Foundations

### 2.1 Video generative paradigms

#### 2.1.1 GAN video models

Generative Adversarial Networks (GANs) employ a two-player minimax game between a generator and discriminator, where the generator learns to produce realistic samples while the discriminator distinguishes between generated and real data. The objective function (Goodfellow et al., 2014) is formulated as:

$$\min_{G} \max_{D} V(D, G) = \mathbb{E}_{x \sim p_{data}(x)}[\log D(x)] + \mathbb{E}_{z \sim p_z(z)}[\log(1 - D(G(z)))] \tag{1}$$

where $G$ and $D$ are the generator and discriminator networks, $x$ is a real sample, and $z$ is a random noise vector. $D(x)$ indicates the probability that $x$ is real, while $D(G(z))$ reflects how real the generated sample

appears. For video generation, GANs have been extended to model temporal consistency through various architectural innovations and training strategies.

Early video GANs focused on temporal modeling through specialized architectures. TGAN (Saito et al., 2017) employs dual generators: a temporal generator creates motion features while an image generator produces video frames. TGAN-v2 (Saito et al., 2020) improves efficiency through multi-resolution generation using cascaded generator modules. MoCoGAN (Tulyakov et al., 2018) disentangles latent space into motion and content components, predicting motion vectors auto-regressively, while DVD-GAN (Clark et al., 2019) generates all frames in parallel with class conditioning. MoCoGAN-HD (Tian et al., 2020) advances motion-content disentanglement through latent motion trajectory prediction. DIGAN (Yu et al., 2021) leverages implicit neural representations for efficient long-sequence generation with frame-pair focused motion discrimination.

StyleGAN2 (Karras et al., 2020) introduced expressive latent spaces for photorealistic synthesis, establishing foundations for video generation through sparse motion cues. StyleGAN-V (Skorokhodov et al., 2022) models videos as continuous-time signals, enhancing temporal coherence for high-resolution, long-duration generation. StyleInv (Wang et al., 2023i) proposes non-autoregressive GAN inversion for smoother motion transitions and style transfer. AniFaceGAN (Wu et al., 2022c) adapts StyleGAN2 into 3D-aware GANs for multiview-consistent face animation. Recent advances include SIDGAN (Muaz et al., 2023) for high-resolution dubbed video generation through shift-invariant learning and pyramid-based decoding, and StyleLipSync (Ki & Min, 2023) for temporally consistent lip-synced videos with pose-aware masking and latent smoothing.

Large-scale video generative models have emerged as a significant advancement. Open and Advanced Large-Scale Video Generative Models (Wan et al., 2025) demonstrate the potential of scaling GAN architectures for complex video synthesis tasks. Hunyuan-Large (Sun et al., 2024b) introduces a 52-billion parameter mixture-of-experts model, demonstrating the ability of large-scale architectures to handle complex video generation tasks.These advances go beyond traditional GANs and bring models that are more powerful and efficient for large-scale video generation.

### 2.1.2 Auto-regressive video models

Auto-Regressive models generate data by modelling the conditional distribution of each data point given its preceding data points. In the context of video generation, Auto-Rregressive models generate realistic and coherent videos by predicting each frame based on previously generated frames. Generally, three strategies have been employed. The first is pixel-level auto-regression. Traditional representative methods, such as Video Pixel Networks (VPN) (Kalchbrenner et al., 2017), used LSTMs to capture temporal dependencies and PixelCNN (Reed et al., 2017) to model spatial and color dependencies within each frame. The second is frame-level auto-regression. (Huang et al., 2022a) proposed auto-regressive GAN to predict frames based on a single still frame. The third is latent-level auto-regression, which significantly saves processing time due to reduced data redundancy and achieves a good time quality trade-off (Rakhimov et al., 2020; Seo et al., 2022; Yan et al., 2021; Ge et al., 2022; Hong et al., 2023a).

### 2.1.3 Video diffusion models

GAN-based and Auto-Regressive models often face challenges in maintaining temporal consistency and computational efficiency when generating videos. While GANs struggle with maintaining temporal consistency and often produce unstable or inconsistent frames, diffusion models excel at generating smooth, coherent sequences, particularly for long videos, because their iterative denoising process ensures frame-to-frame consistency. Also, despite Auto-Regressive model can handle both continuous and discrete data, it normally requires high computational cost. In terms of frame quality, diffusion models generate more detailed and higher-resolution outputs compared to GANs, which are prone to artifacts from adversarial training, and autoregressive models, which face resolution limitations due to the compounding of errors in sequential prediction (Moser et al., 2024). Diffusion models also offer superior controllability and local editing capabilities because they can refine specific parts of the video during the denoising process without affecting other regions

while GANs and autoregressive methods generate frames in a more holistic or sequential manner (Huang et al., 2024a).

Diffusion models offer a more efficient solution by framing video generation as a denoising process. Early models, such as the Video Diffusion Model (VDM) (Ho et al., 2022c), adapted the U-Net architecture to 3D for spatio-temporal video generation but faced high computational costs and limited resolution.

Later, Imagen-Video (Ho et al., 2022a) uses a cascaded diffusion process (Ho et al., 2022b) to generate high-resolution videos. It introduces temporal attention layers between spatial layers to capture motion information effectively. Make-a-Video (Singer et al., 2022) is another powerful competitor in text-video synthesis by conditioning on the CLIP semantic space. It generates keyframes based on text input and uses a cascade of interpolation and upsampling diffusion models to ensure high consistency and fidelity. However, both of these pioneering models come with high computational costs. To address this, MagicVideo (Zhou et al., 2022) introduced low-dimensional latent embedding space in their diffusion process by a pretrained variational-auto-encoder (VAE), significantly reducing computational demands. Video LDM (Blattmann et al., 2023b) also extended the adaptation of the Latent Diffusion Models (Rombach et al., 2022) to text-to-video generation tasks by adding temporal attention layers to a pre-trained text-to-image diffusion model and fine-tune them on labeled video data. Their model is capable of generating long-driving car video sequences auto-regressively and producing videos of personalized characters. Recently, CCEdit (Feng et al., 2024a) leverages diffusion models with a trident network architecture that decouples structure and appearance control, ensuring precise and creative editing capabilities.

### 2.1.4 Auto-regressive video diffusion models

Some recent works further exploit auto-regressive framework in the context of video generation with diffusion models (Weng et al., 2024; Jin et al., 2024; Xie et al., 2024a). ART-V (Weng et al., 2024) first utilized autoregressive diffusion to generate frames sequentially, employing masked diffusion to enhance coherence and reduce inconsistencies. Subsequently, Pyramidal Flow Matching (Jin et al., 2024) introduced spatial and temporal pyramids to optimize autoregressive generation, reducing computational costs and improving scalability. Progressive Autoregressive Video Diffusion Models (Xie et al., 2024a) refined this by using a progressive noise schedule, enabling smoother transitions in long videos. CausVid (Yin et al., 2024) advanced the field by converting bidirectional diffusion models into causal autoregressive generators through asymmetric distillation, achieving real-time frame-by-frame generation with low latency and reduced error accumulation.

### 2.2 Learning foundations

Video generation relies on different methods to produce realistic and coherent results. As the section discussed above, common paradigms in Figure 1 include GANs, autoregressive models, and diffusion models, each with trade-offs in quality and efficiency. This section introduces the core learning principles behind these methods, starting with diffusion processes, which have become the leading approach for high-quality and temporally consistent video generation. These foundational concepts serve as the basis for the specific diffusion model frameworks and implementations discussed in subsequent sections.

### 2.2.1 Denoising diffusion probabilistic models (DDPM)

*Denoising diffusion probabilistic models* (DDPM) Ho et al. (2020); Nichol & Dhariwal (2021); Sohl-Dickstein et al. (2015) consist of two interconnected Markov chains: a forward diffusion process that gradually adds noise to the data, and a reverse process that removes noise to recover the data. The forward process transforms complex data distributions into a simpler prior, typically Gaussian noise, while the reverse process learns how to reconstruct the data from this noisy representation by modeling the reverse transitions. Data generation involves first sampling a random noise vector from the prior, and then applying the reverse chain, step-by-step, to iteratively refine the noise into meaningful data. The key challenge lies in training the reverse Markov chain so that it effectively mimics the true reversal of the forward process.

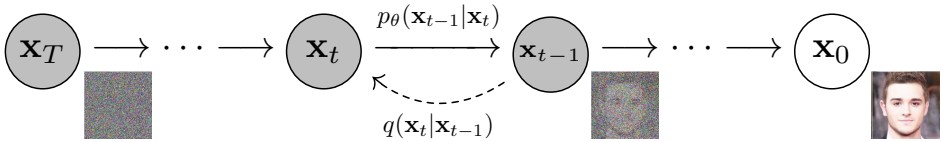

Figure 2: The directed graphical model for DDPM(Ho et al., 2020)

Mathematically, let the data be sampled from a distribution $x_0 \sim q(x_0)$. The forward Markov process produces a sequence of progressively noisier variables $x_1, x_2, ..., x_T$ through a transition kernel $q(x_t|x_{t-1})$. The joint distribution of these variables, conditioned on the original data $x_0$, can be expressed as:

$$q(x_1, ..., x_T|x_0) = \prod_{t=1}^{T} q(x_t|x_{t-1}). \tag{2}$$

Typically, the transition kernel is defined as a Gaussian distribution:

$$q(x_t|x_{t-1}) = \mathcal{N}(x_t; \sqrt{1 - \beta_t}x_{t-1}, \beta_t\mathbf{I}), \tag{3}$$

where $\beta_t \in (0, 1)$ is a pre-determined noise schedule controlling the amount of noise added at each step.

In the reverse process, the goal is to recover the data by sampling from a reverse Markov chain. The chain starts with a noise sample $x_T \sim p(x_T) = \mathcal{N}(0, \mathbf{I})$ from a simple Gaussian prior, and transitions through a learnable kernel $p_\theta(x_{t-1}|x_t)$, parameterized by neural networks. This kernel takes the form:

$$p_\theta(x_{t-1}|x_t) = \mathcal{N}(x_{t-1}; \mu_\theta(x_t, t), \Sigma_\theta(x_t, t)), \tag{4}$$

where $\mu_\theta(x_t, t)$ and $\Sigma_\theta(x_t, t)$ represent the learned mean and variance, respectively. By iteratively applying this reverse kernel, we generate the data $x_0$ through a sequence of updates starting from the noise sample $x_T$ and progressing through $x_{T-1}, x_{T-2}, \ldots, x_0$. Figure 2 illustrates the graphical model for DDPM.

This iterative sampling process captures the reverse dynamics of the original forward diffusion, gradually refining random noise into a meaningful data point, such as an image or text.

### 2.2.2 Denoising diffusion implicit models (DDIM)

This original DDPM formulation of the generation process in the form of a reverse Markov chain has more recently been complemented by a non-Markovian alternative denoted as denoising diffusion implicit models (DDIM), which offers a deterministic and more efficient generation process. In DDPM, the sampling process typically requires a large number of iterations to achieve satisfactory results. In contrast, the DDIM is designed to accelerate the sampling process by reducing the number of iterations needed. Here, a backward denoising step can be computed with

$$x_{t-1} = \frac{\sqrt{\bar{\alpha}_{t-1}}x_t - \sqrt{1 - \bar{\alpha}_t}\epsilon_\theta(x_t, t)}{\sqrt{\bar{\alpha}_t}} + \sqrt{1 - \bar{\alpha}_{t-1}}\epsilon_\theta(x_t, t). \tag{5}$$

One distinct advantage of this formulation of the denoising process is that it allows for accurate reconstruction of the original input video $x_0$ from the noise at time step $T$. This technique, called DDIM inversion, can be utilized for applications such as image and video editing.

### 2.2.3 Elucidated diffusion models (EDM)

EDM (Karras et al., 2022) introduce a unified and optimized framework for diffusion-based generative models by reinterpreting sampling and training processes. Unlike DDPMs, which rely on stochastic Markovian sampling, or DDIMs, which employ deterministic non-Markovian updates, EDMs aim to optimize both training and sampling efficiency by introducing tailored sampling schemes and reparameterization strategies. EDMs build upon the common framework of diffusion models by considering the data distribution $q(x_0)$ perturbed by Gaussian noise, leading to a sequence of noisy variables $\{x_t\}$ over time.

While DDPMs use a stochastic reverse process, EDMs leverage an ordinary differential equation (ODE) for deterministic sampling, as inspired by DDIMs. The probability flow ODE Song et al. (2020a) is expressed as:

$$\frac{dx}{dt} = -\sigma(t)\nabla_x \log p(x; \sigma(t)), \tag{6}$$

where $\sigma(t)$ is the noise schedule at time $t$. Unlike DDIMs, EDMs employ second-order solvers, such as Heun's method, to balance accuracy and computational efficiency.

To ensure stable training, EDMs redefine the loss function with noise-dependent preconditioning. The denoising target $D_\theta(x; \sigma)$ is expressed as:

$$D_\theta(x; \sigma) = c_{\text{skip}}(\sigma)x + c_{\text{out}}(\sigma)F_\theta(c_{\text{in}}(\sigma)x; c_{\text{noise}}(\sigma)), \tag{7}$$

where $F_\theta$ is a neural network, and $c_{\text{skip}}$, $c_{\text{out}}$, and $c_{\text{in}}$ are scale-dependent coefficients. The overall training objective becomes:

$$\mathcal{L} = \mathbb{E}_{\sigma,x_0,\epsilon}\left[\lambda(\sigma)\|D_\theta(x + \sigma\epsilon; \sigma) - x_0\|^2\right], \tag{8}$$

where $\lambda(\sigma)$ is a weighting function that emphasizes intermediate noise levels during training.

Compared to DDPMs, EDMs achieve faster sampling by reducing the number of neural function evaluations through efficient solvers and noise schedules. Unlike DDIMs, which rely on deterministic updates, EDMs integrate adaptive techniques to minimize truncation errors further, further improving fidelity and flexibility. This unified and modular design allows EDMs to achieve better results while significantly reducing computational overhead(Karras et al., 2022).

### 2.2.4 Flow matching and rectified flow

Despite denoising diffusion being powerful for video generation tasks, another branch in the family of differential-equation-based generative models began to arise recently, namely the flow matching generative models(Lipman et al., 2023; Liu et al., 2022c; Tong et al., 2023). While diffusion models learn the score function of a specific SDE, flow matching aims to model the vector field implied by an arbitrary ODE directly. A neural network is used for approximating the vector field, and the ODE can also be numerically solved to obtain data samples. The design of such ODE and vector field often considers linearizing the sampling trajectory and minimizing the transport cost(Tong et al., 2023). As a result, flow matching models have simpler formulations and fewer constraints but better quality. VoiceBox(Le et al., 2023) shows the potential of flow matching in fitting large-scale speech data, and LinDiff(Liu et al., 2023b) shares a similar concept in the study of vocoders. More importantly, the rectified flow(Liu et al., 2022c) technique in flow matching models further straightens the ODE trajectory in a concise way. By training a flow matching model again but with its own generated samples, the sampling trajectory of rectified flow theoretically approaches a straightforward line, which improves the efficiency of sampling. *Rectified Flow*(Liu et al., 2022c; 2023d) is a generative model that efficiently transitions between two distributions $\pi_0$ and $\pi_1$ by solving ordinary differential equations (ODEs). Building on flow matching(Lipman et al., 2023) and stochastic interpolants(Albergo et al., 2023), Rectified Flow introduces a rectification mechanism to optimize the ODE path, improving the efficiency and stability of the sampling process. This approach establishes a strong theoretical baseline for diffusion acceleration and provides a unified perspective on generative models.

The core dynamics of Rectified Flow are governed by the following ODE:

$$\mathrm{d}Z_t = v(Z_t, t)\mathrm{d}t,$$

where $Z_t$ represents the data point at time $t \in [0, 1]$, which evolves continuously along the flow. The vector field $v(Z_t, t)$ determines the direction and velocity of $Z_t$'s movement. This vector field is parameterized using a neural network, which is optimized to align the trajectories with the target distributions. The term $\mathrm{d}t$ represents an infinitesimal time step, enabling a smooth transition between $\pi_0$ and $\pi_1$.

To learn the vector field $v$, Rectified Flow minimizes the following objective:

$$\min_v \int_0^1 \mathbb{E}\left[\|(X_1 - X_0) - v(X_t, t)\|^2\right] \mathrm{d}t,$$

where $X_0$ and $X_1$ are sampled from the initial distribution $\pi_0$ and target distribution $\pi_1$, respectively. The term $X_t = tX_1 + (1-t)X_0$ represents a linear interpolation between $X_0$ and $X_1$. The loss function measures the squared difference between the actual direction of interpolation $(X_1 - X_0)$ and the vector field $v(X_t, t)$. This ensures that $v$ effectively captures the necessary dynamics to transition samples between the two distributions while preserving their structure.

A key theoretical advantage of Rectified Flow is that the ODE formulation guarantees that paths do not cross. The vector field $v(Z_t, t)$ ensures unique and deterministic trajectories for all starting points, avoiding ambiguities in sampling and maintaining stability during inference. With its focus on optimizing vector fields and straightening trajectories, Rectified Flow offers a powerful and efficient alternative to traditional diffusion models. By combining a strong theoretical foundation with practical efficiency, it sets a new benchmark for generative modeling tasks.

### 2.2.5 Learning from feedback and reward models

The integration of human feedback into video diffusion models has evolved significantly, beginning with foundational ideas in reinforcement learning and progressing to advanced mechanisms leveraging multimodal and AI-driven annotations. Early work C2M(Ardino et al., 2021) user interaction by selecting objects in the scene and specifying their final location through mouse clicks to generate video in complex scenes. InstructVideo (Yuan et al., 2023) further emphasized aligning video generation with human preferences by incorporating temporal coherence loss:

$$\mathcal{L}_{\text{coherence}} = \mathbb{E}\theta \left[ \sum_{j=1}^{F-1} |(v_{j+1} - v_j) - (o_{j+1} - o_j)|_2^2 \right], \tag{9}$$

where $v_j$ and $o_j$ denote predicted and ground truth frames. This method enhanced temporal consistency but primarily focused on supervised fine-tuning with explicit human feedback signals. Subsequent research introduced more scalable methods for integrating feedback. (Furuta et al., 2024b) demonstrated the use of binary AI feedback from vision-language models (VLMs) to optimize dynamic object interactions in videos. Their unified reinforcement learning framework leveraged AI annotations to align outputs with human-like quality assessments, overcoming the limitations of manual data annotation. This concept was further extended in T2V-Turbo(Li et al., 2024d) and its subsequent version T2V-Turbo-V2(Li et al., 2024e), which significantly improved the quality and speed of text-to-video generation through the incorporation of AI feedback, enhancing dynamic object interaction.(Furuta et al., 2024a) In parallel, efforts to construct comprehensive preference datasets began to take shape. (Wu et al., 2024g) presented VIDEOPREFER, a large-scale dataset created using multimodal large language models (MLLMs) such as GPT-4 Vision. This dataset provided 135,000 preference annotations, allowing the training of VIDEORM, the first general-purpose reward model tailored for video preferences. VIDEORM incorporated temporal dynamics, significantly enhancing the alignment of video outputs with human expectations. (Furuta et al., 2024a) further proposed a robust framework for improving dynamic interactions in video generation, showcasing AI feedback as a critical factor for realistic object behavior in generated content. Other works, such as FreeScale(Qiu et al., 2024c)indirectly addressed alignment by improving high-resolution visual generation in video diffusion models. Their approach, while not explicitly focused on feedback, laid the groundwork for integrating multi-scale visual consistency with human-preferred attributes in video content. Some advanced methods, like VINE(Lu et al., 2024b) also highlighted the robustness of feedback-aligned generative priors for specific tasks like watermarking and video editing. These innovations showed how feedback mechanisms could extend beyond video generation to broader applications, ensuring fidelity and alignment with user intent.

### 2.2.6 One-shot and few-shot learning

One-shot and few-shot fine-tuning techniques enable generative models to adapt to new tasks or domains with minimal training data. These approaches are particularly valuable in scenarios where acquiring large datasets is infeasible, yet high-quality outputs are required. By leveraging either a single example (one shot) or a small set of examples (few shots), these methods refine pretrained generative models to specialize in specific tasks while maintaining generalization capabilities.

One-shot learning refers to a model's ability to learn from just a single example. In generative modelling, this is particularly useful when only one instance of the target data is available. For example, Tune-A-Video (Wu et al., 2023b) demonstrates how a pre-trained text-to-image diffusion model can be fine-tuned on a single text-video pair, enabling it to generate new videos that maintain the temporal consistency of the original while adhering to text prompts. However, Tune-A-Video is based on the template video and edits it with different content prompts which will restrict the freedom of generative video.

Few-shot fine-tuning extends this capability by adapting models to new domains or tasks using a small number of training samples. Several techniques have emerged to enhance the generative capacity of models under few-shot scenarios. LAMP (Learn A Motion Pattern) (Wu et al., 2023c) proposes a motion learning model to capture the motion pattern from the training data and utilizes about $8\tilde{1}6$ videos to tune the pretrained T2I model. However, it is constrained by their incomplete capture of image features. MAIM (Huang et al., 2025) addresses this by integrating image features via the CLIP encoder, enabling videos with intricate scenes and multiple objects.

### 2.2.7 Training-free methods

Training-free approaches enable direct video generation without additional training or fine-tuning, making them valuable for black-box scenarios.

Text2Video-Zero (Khachatryan et al., 2023) introduced cross-frame attention based on pretrained text-to-image diffusion model. FateZero (Qi et al., 2023) enhanced this through intermediate attention maps, and Free-Bloom (Huang et al., 2023a) combined GPT-3 with image diffusion models for improved semantic coherence. PEEKABOO (Jain et al., 2024b) enabled spatio-temporal control through masked diffusion using a latent diffusion model-based video generation model, while ControlVideo (Zhang et al., 2023e) extended ControlNet with an interleaved-frame smoother to generate consistent controllable and structurally smooth video. DreamVideo-2 (Wei et al., 2024b) proposed zero-shot subject-driven video customization with precise motion control, enabling subject-preserving and motion-specific video editing. Following this, VideoElevator (Zhang et al., 2024g) encapsulates T2V to enhance temporal consistency and harnesses T2I to provide more faithful details. Recent innovations include GPT4Motion (Lv et al., 2024)'s combination of GPT-4, Blender, and text-to-image diffusion models for physics-aware generation, AnyV2V (Ku et al., 2024)'s universal editing framework built on an image editing model combined with an existing image-to-video generation model to generate the edited video through temporal feature injection. Video Custom Diffusion (Magic-Me) (Ma et al., 2024e) built a training-free 3D Gaussian Noise Prior for video frame initialization, reconstructing inter-frame correlation to achieve identity-preserved generation.

### 2.2.8 Token learning

Token learning, such as Textual Inversion Gal et al. (2022), is another paradigm within generative modeling. Instead of fine-tuning the entire model, token learning focuses on training specific embeddings or tokens that represent new concepts or ideas. By using textual inversion, the model learns a new token that can be combined with existing tokens to guide generation without requiring large datasets or significant changes to the model's architecture. However, optimizing the single token embedding vector has limited expressive capacity because of its limited optimized parameter size. In addition, using one word to describe concepts with rich visual features and details is very hard and insufficient. Animate-A-Story(He et al., 2023a) extended traditional textual inversion to timestep-variable textual inversion (TimeInv) and adapted it to video generation task to ensure flexible controls over structure and characters. DreamVideo(Wei et al., 2024c) further incorporated textual inversion to capture the fine appearance of the subject from provided images to generate videos with customized subjects.

The learning foundations discussed in this section provide the core principles behind modern video generation. These fundamental concepts provide the mathematical and conceptual paradigm, as shown in Figure 1, which are necessary for understanding the specific diffusion model implementations and architectural designs explored in the following sections. Building upon these foundations, subsection 2.4 examines the practical frameworks and implementations to use these theoretical principles into real-world video generation systems.

### 2.3 Guidances

### 2.3.1 Classifier guidance

In generative modeling, diffusion models such as DDPM and DDIM can be guided using classifiers to improve the quality of conditional generation. This approach, called *classifier guidance* (Dhariwal & Nichol, 2021), integrates class information into the diffusion process by leveraging the gradients of a classifier trained on noisy data. The classifier's gradients help the model generate data conditioned on a given label, improving the overall sample quality for specific categories.

Mathematically, the classifier-guided diffusion process modifies the reverse noising process by incorporating the classifier's log-probability gradients. This guidance is applied at each timestep of the diffusion process, allowing the model to generate more accurate conditional samples.

The reverse noising process is represented as follows:

$$p_{\theta,\phi}(x_t|x_{t+1}, y) = Z \cdot p_\theta(x_t|x_{t+1}) \cdot p_\phi(y|x_t) \tag{10}$$

where $p_\theta$ is the reverse noising process from the original diffusion model, $p_\phi(y|x_t)$ is the classifier's conditional probability, and $Z$ is a normalizing constant.

We now describe how this approach can be applied to both DDPM and DDIM frameworks. In particular, they(Dhariwal & Nichol, 2021) train a classifier $p_\phi(y|x_t, t)$ on noisy images $x_t$, and then use gradients $\nabla_{x_t} \log p_\phi(y|x_t, t)$ to guide the diffusion sampling process towards an arbitrary class label $y$.

**Algorithm for classifier-guided DDPM sampling.** In the DDPM framework, classifier guidance can be incorporated by modifying the reverse diffusion process. The update at each timestep involves sampling from a Gaussian distribution with a mean adjusted by the classifier gradients. Detailed workflow can be referred to Algorithm algorithm 1.

---

**Algorithm 1** Classifier-guided DDPM sampling, given a diffusion model $(\mu_\theta(x_t), \Sigma_\theta(x_t))$, classifier $p_\phi(y|x_t)$, and gradient scale $s$.(Dhariwal & Nichol, 2021)

---

1: **Input:** class label $y$, gradient scale $s$
2: $x_T \leftarrow$ sample from $\mathcal{N}(0, I)$
3: **for** $t = T$ to 1 **do**
4: $\quad \mu, \Sigma \leftarrow \mu_\theta(x_t), \Sigma_\theta(x_t)$
5: $\quad x_{t-1} \leftarrow$ sample from $\mathcal{N}(\mu + s\Sigma\nabla_{x_t} \log p_\phi(y|x_t), \Sigma)$
6: **end for**
7: **return** $x_0$

---

**Algorithm for classifier-guided DDIM sampling.** For the DDIM framework, the deterministic nature of the sampling process requires a slightly different approach. The classifier gradient is used to modify the epsilon prediction, which guides the reverse sampling process in DDIM. Deatailed workflow can be referred to Algorithm algorithm 2.

---

**Algorithm 2** Classifier-guided DDIM sampling, given a diffusion model $\epsilon_\theta(x_t)$, classifier $p_\phi(y|x_t)$, and gradient scale $s$. (Dhariwal & Nichol, 2021)

---

1: **Input:** class label $y$, gradient scale $s$
2: $x_T \leftarrow$ sample from $\mathcal{N}(0, I)$
3: **for** $t = T$ to 1 **do**
4: $\quad \epsilon \leftarrow \epsilon_\theta(x_t) - \sqrt{1 - \bar{\alpha}_t}\nabla_{x_t} \log p_\phi(y|x_t)$
5: $\quad x_{t-1} \leftarrow \frac{\sqrt{\bar{\alpha}_{t-1}}x_t - \sqrt{1-\bar{\alpha}_t}\epsilon}{\sqrt{\bar{\alpha}_t}} + \sqrt{1 - \bar{\alpha}_{t-1}}\hat{\epsilon}$
6: **end for**
7: **return** $x_0$

---

In the DDIM case, it defines a new epsilon prediction $\hat{\epsilon}(x_t)$, which incorporates the classifier's guidance into the joint distribution of the diffusion model and the classifier:

$$\hat{\epsilon}(x_t) := \epsilon_\theta(x_t) - \sqrt{1 - \bar{\alpha_t}} \nabla_{x_t} \log p_\phi(y|x_t) \tag{11}$$

This modified epsilon prediction is then used to guide the reverse sampling process, replacing the standard noise predictions from the original model.

The development of classifier guidance for diffusion models has sparked significant advances in video generation applications. (Shi et al., 2023a) explored compositionality in visual generation by using latent classifiers to guide diffusion models in semantic space, demonstrating effective control over multiple attributes and their relationships. (Chung et al., 2024) further reformulated classifier guidance to better respect manifold constraints crucial for temporal consistency in videos. (Lian et al., 2024) then incorporated large language models to generate dynamic scene layouts as intermediate representations, effectively guiding the diffusion process for coherent motion and interactions. These developments demonstrate the field's progression from basic guidance mechanisms to more sophisticated approaches specifically tailored for video generation challenges.

### 2.3.2 Classifier-free guidance

While classifier-based guidance successfully allows for the conditional generation of data by leveraging the gradients from a pre-trained classifier, it has several limitations. For instance, classifier guidance requires the maintenance of a separate classifier, and its performance is closely tied to the quality of this classifier. To address these concerns, *classifier-free guidance*(Ho & Salimans, 2022) offers an alternative that achieves similar results without the need for a dedicated classifier.

Classifier-free guidance modifies the model's predicted noise directly by using conditioning information, allowing for conditional generation without relying on external classifier gradients. This approach avoids the complexities of training and maintaining an additional classifier.

Rather than training a separate classifier model, they(Ho & Salimans, 2022) adopt an approach that jointly trains an unconditional denoising diffusion model, $p_\theta(x)$, and a conditional model, $p_\theta(x|y)$. Both models are parameterized through a shared score estimator: $\epsilon_\theta(x_t)$ for the unconditional case and $\epsilon_\theta(x_t, y)$ for the conditional case. They use a single neural network to parameterize both models and then perform sampling using a linear combination of conditional and unconditional scores, as shown in Equation 12.

$$\hat{\epsilon}_\theta(x_t, y) = (1 + w)\epsilon_\theta(x_t, y) - w\epsilon_\theta(x_t) \tag{12}$$

Here, $w$ is a guidance scale that controls the strength of the conditional information. This formulation allows the model to perform conditional sampling without relying on classifier gradients.

**Algorithm for Classifier-Free Guidance** describes the sampling procedure for classifier-free guidance, which adjusts the score estimates directly within the diffusion model without requiring any classifier gradients. Algorithms algorithm 3 and algorithm 4 describe training and sampling with classifier-free guidance in detail.

This algorithm leverages the classifier-free guidance to adjust the sampling procedure by computing the guided score at each timestep. Instead of using gradients from a classifier, the model directly modifies its own score estimates to account for the conditional information.

### 2.4 Diffusion model frameworks

This section explores several key frameworks within diffusion models, highlighting their distinctive approaches and applications.

This section builds on the theoretical foundations from subsection 2.2 by introducing practical diffusion model frameworks for video generation. While the previous section focused on core principles, this section

---

**Algorithm 3** Joint training a diffusion model with classifier-free guidance

---

**Require:** $p_{\text{uncond}}$: probability of unconditional training
1: **repeat**
2:     $(x, y) \sim p(x, y)$                          ▷ Sample data with conditioning information from the dataset
3:     $y \leftarrow \emptyset$ with probability $p_{\text{uncond}}$            ▷ Randomly discard conditioning to train unconditionally
4:     $\lambda \sim p(\lambda)$                                ▷ Sample log SNR value
5:     $\epsilon \sim \mathcal{N}(0, I)$
6:     $x_\lambda = \alpha_\lambda x + \sigma_\lambda \epsilon$                        ▷ Corrupt data to the sampled log SNR value
7:     Take gradient step on $\nabla_\theta \|\epsilon_\theta(x_\lambda, y) - \epsilon\|^2$            ▷ Optimize the denoising model
8: **until** converged

---

**Algorithm 4** Conditional sampling with classifier-free guidance

---

**Require:** $w$: guidance strength
**Require:** $c$: conditioning information for conditional sampling
**Require:** $\lambda_1, \ldots, \lambda_T$: increasing log SNR sequence with $\lambda_1 = \lambda_{\min}, \lambda_T = \lambda_{\max}$
1: $x_0 \sim \mathcal{N}(0, I)$
2: **for** $t = 1, \ldots, T$ **do**
3:     $\tilde{\epsilon}_{t-1} \leftarrow (1 + w)\epsilon_\theta(x_{t-1}) - w\epsilon_\theta(x_{t-1})$          ▷ Form the classifier-free guided score at log SNR $\lambda_{t-1}$
4:     $x_t \leftarrow \frac{(x_{t-1} - \alpha_{\lambda_{t-1}} \tilde{\epsilon}_{t-1})}{\alpha_{\lambda_{t-1}}}$          ▷ Sampling step (could be replaced by another sampler, e.g., DDIM)
5:     $x_t \sim \mathcal{N}(\tilde{\mu}_{\lambda_{t-1}, \lambda_t}(x_{t-1}, x_t), \tilde{\sigma}^2_{\lambda_{t-1}, \lambda_t})$
6: **end for**
7: **return** $x_T$

---

focus on illustrating how diffusion-based video generation builds on theoretical foundations through a range of practical strategies. The following sections examine four key implementation approaches: pixel-based and latent-based modeling, optical flow integration, adaptive noise scheduling, and agent-driven frameworks, each addressing specific challenges in video synthesis such as temporal consistency, efficiency, and controllability.

### 2.4.1 Pixel diffusion and latent diffusion

The theoretical diffusion processes introduced in subsection 2.2 can be implemented through different architectural approaches, each offering distinct advantages for video generation. Pixel-based and latent-based diffusion models represent two fundamental implementation strategies that directly translate the mathematical paradigm of DDPM, DDIM, and other diffusion variants into practical video generation systems as shown in Figure 1.

Remarkable progress has been made in developing large-scale pre-trained text-to-Video Diffusion Models (VDMs), including proprietary models (*e.g.*, Make-A-Video (Singer et al., 2023), Imagen Video (Ho et al., 2022c), Video LDM (Blattmann et al., 2023b), Gen-2 (Esser et al., 2023)) and open-sourced ones (*e.g.*, VideoCrafter (He et al., 2023b), ModelScopeT2V (Wang et al., 2023c). These VDMs can be categorized into two primary architectures: (1) Pixel-based VDMs and (2) Latent-based VDMs. The former directly operates on pixel values for denoising, as exemplified by Make-A-Video Singer et al. (2023), Imagen Video Ho et al. (2022c), and PYoCo Ge et al. (2023). The latter manipulates the compressed latent space within a variational autoencoder (VAE), as demonstrated by Video LDM Blattmann et al. (2023b) and MagicVideo Zhou et al. (2022). However, both of them have pros and cons. **Pixel-based VDMs** can generate motion accurately aligned with the textual prompt but typically demand expensive computational costs in terms of time and GPU memory, especially when generating high-resolution videos. **Latent-based VDMs** are more resource-efficient because they work in a reduced-dimension latent space. However, it is challenging for such a small latent space (*e.g.*, $8 \times 5$ for $64 \times 40$ videos) to cover rich yet necessary visual semantic details as described by the textual prompt. Thus, if the generated videos are often not well-aligned with the textual prompts or the generated videos are of relatively high resolution (*e.g.*, $256 \times 160$ videos), the latent model will focus more on spatial appearance but may also ignore the text-video alignment. Recently, Show-1(Zhang et al.,

2023a) integrates the strengths of both pixel and latent VDMs, resulting in a novel video generation model that can produce high-resolution videos of precise text-video alignment at low computational cost (15G GPU memory during inference). LATTE Ma et al. (2024c) further proposes a novel latent diffusion transformer for video generation, which extracts spatio-temporal tokens from input videos and models video distribution in the latent space using Transformer blocks, achieving state-of-the-art performance across multiple video generation datasets by exploring optimal design choices such as video clip patch embedding, model variants, timestep-class information injection, and temporal positional embedding.

### 2.4.2 Optical-flow-based diffusion models

Another distinct approach within diffusion models leverages optical flow to maintain temporal coherence between frames. Optical flow is a common approach to represent motion by estimating the displacement field between consecutive frames. Early approaches treated it as an optimization problem using handcrafted features to maximize visual similarity(Horn & Schunck, 1981; Black & Anandan, 1993; Bruhn et al., 2005; Sun et al., 2014). Incorporating deep learning-based methods revolutionized this field. FlowNet(Dosovitskiy et al., 2015) pioneered end-to-end optical flow estimation, demonstrating the potential of deep learning in this domain. Subsequent advancements on improved architectures and synthetic datasets further promoted the progress of optical flow estimation(Ilg et al., 2017; Ranjan & Black, 2017; Sun et al., 2018; 2021; Hui et al., 2018; 2020; Yang & Ramanan, 2019). RAFT(Teed & Deng, 2020) included iterative refinement with correlation volumes, significantly boosting performance, while FlowFormer(Huang et al., 2022b; Shi et al., 2023c) applied attention mechanisms within optical flow usage. VideoFlow(Shi et al., 2023b) extended optical flow to video generation scenarios and utilized temporal information across multiple frames, achieving high accuracy. (Hu et al., 2023)propose a Dynamic Multi-scale Voxel Flow Network (DMVFN) to explicitly model the complex motion cues of diverse scales between adjacent video frames by dynamic optical flow estimation to generate videos at lower computational costs. Also, Motion-I2V(Shi et al., 2024) further refined the motion modelling by a diffusion-based motion field predictor and motion-augmented temporal attention to generate more consistent videos even in the presence of large motion and viewpoint variation. By conditioning the optical flow between a previous and a subsequent frame, this framework achieves realistic motion continuity, which is critical for applications in video generation and motion synthesis. Additionally, recent advancements explore upgrading pretrained image diffusion models to video generation by temporally warping input noise. How I Warped Your Noise (Chang et al., 2024a) proposed a novel noise representation called integral noise ($\int$-$noise$), preserving temporal correlations through an optical-flow-based noise warping algorithm. This foundational approach mitigated issues like high-frequency flickering and texture-sticking artifacts, laying the groundwork for temporally coherent video diffusion. Building upon this, Infinite-Resolution Integral Noise Warping (Deng et al., 2025) method introduces a training-free approach where temporal correlations in noise are preserved by warping input noise using integral representations, enabling consistent frame generation while reducing computational overhead. These diffusion model frameworks demonstrate the versatility and adaptability of diffusion processes across different domains, from high-resolution static images to complex, temporally coherent videos. As shown in Figure 3, Go-with-the-Flow (Burgert et al., 2025) combined optical flow extraction from training videos with real-time noise warping and fine-tuning video diffusion models on paired warped noise and video data to enable robust and motion-controllable generation. Together, these methods demonstrate the progression from temporal correlation preservation to advanced motion control, showcasing the versatility and adaptability of diffusion models in video generation.

### 2.4.3 Noise scheduling

Diffusion models (Song et al., 2020b; Ho et al., 2020) perturb data with Gaussian noise through a diffusion process for training, and the reverse process is learned to transform the Gaussian distribution back to the data distribution. Perturbing data points with noise populates low data density regions to improve the accuracy of estimated scores, resulting in stable training and image sampling. The forward process is controlled by the handcrafted noise schedule. However, current noise scheduling strategies remain handcrafted for each dataset, such as VP (Ho et al., 2020), VE (Song et al., 2020b), Cosine (Nichol & Dhariwal, 2021) and EDM (Karras et al., 2022). These noise schedules perform well in low-resolution RGB spaces but yield poorer results in higher resolutions (Dhariwal & Nichol, 2021; Hoogeboom et al., 2023). Recent studies (Chen, 2023; Hoogeboom et al., 2023) propose carefully designed noise schedules that outperform VP, VE, and

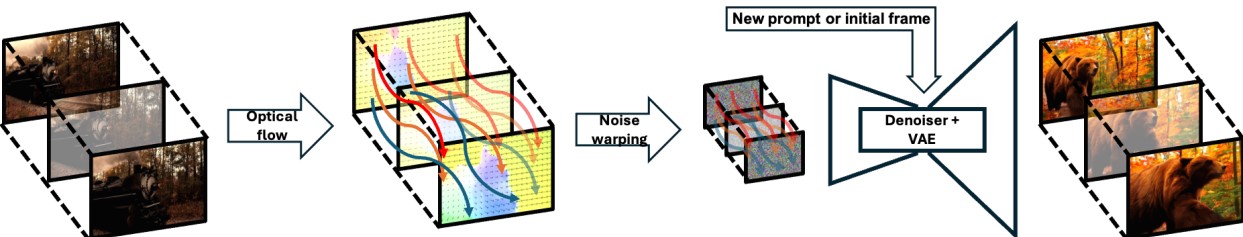

Figure 3: An example for optical flow usage in video scenario, Go-with-the-Flow (Burgert et al., 2025), a novel framework that predicts 3D scene dynamics across sequential frames, outperforming conventional single-frame approaches. By integrating multi-frame spatial-temporal relationships, it improves depth accuracy and visual fidelity in scene reconstruction.

Cosine schedules in RGB space. Besides, (Guo et al., 2023a) analyzed the power spectrum, introduced a numerical quantification of noise levels, and proposed the weighted signal-to-noise ratio (WSNR) as a unified metric for noise levels in both RGB and latent spaces. WSNR-equivalent training noise schedules significantly enhance high-resolution model performance across both latent and RGB spaces.

### 2.4.4 Agent-based diffusion models

Agent-based approaches have revolutionized diffusion models for multimedia generation, where agents are implemented as specialized neural modules with distinct functionality and decision-making capabilities. Drive-GAN (Kim et al., 2021) pioneered this direction by utilizing multi-agent frameworks for video generation through differentiable simulation. UniSim (Yang et al., 2024g) contributed by developing a neural closed-loop sensor simulator using coordinated agents for scene reconstruction and sensor data generation. MORA (Yuan et al., 2024c) established a comprehensive framework with five specialized agents for different generation stages, introducing self-modulated fine-tuning for dynamic agent contribution adjustment. VideoAgent (Soni et al., 2024b) advanced the framework by integrating multimodal LLM feedback and real-world execution for iterative refinement. AutoGen (Wu et al., 2024d) and MetaGPT (Hong et al., 2024a) extended these principles to broader generative tasks, implementing standardized communication protocols to enable scalable multi-agent collaboration. This progression shows how agent-based diffusion models have evolved from simple architectures to sophisticated multi-agent systems capable of handling complex generation tasks while maintaining consistency through structured collaboration and feedback mechanisms.

## 2.5 Architectures

As shown in Figure 4, most text-conditional visual generation models comprise three key modules: a variational autoencoder (VAE) compresses images and videos into a latent space. Second, a neural network, typically incorporating a U-Net or Transformer-based backbone, performs the denoising within this latent space. The denoising process can be enhanced using optical flow or human feedback to improve generation quality. A text encoder generates embeddings to guide the image or video generation process. These are the main architectures in video generation tasks as shown in Figure 1.

### 2.5.1 UNet

The UNet(Ronneberger et al., 2015) architecture has become foundational in designing denoising models for visual diffusion applications, originally developed for medical image segmentation and now adapted widely for generative tasks across images, video, and audio. In image generation tasks, a UNet typically encodes an input into progressively lower-resolution latent representations while increasing feature channels through a series of encoding layers. This encoded latent is then upsampled back to the original resolution by corresponding decoding layers.

In diffusion models, UNet can operate in either the pixel space or the latent space. For example, Latent Diffusion Models (LDMs) (Rombach et al., 2022) utilize UNet within a lower-dimensional latent space, which

| Models | Backbone | VAE | Text Encoder | # Params | Training dataset and size | Resolution | Duration | # Frames | GPU size | GPU time |
|---|---|---|---|---|---|---|---|---|---|---|
| **Academia open-source models** | | | | | | | | | | |
| VideoGPT(Yan et al., 2021) | - | VQVAE | - | - | UCF-101, BAIR | 64 × 64 | - | - | 8 × RTX 6000 | - |
| CogVideo(Hong et al., 2023a) | DiT | 2D VAE | GPT-3 | 15.5B | WebVid-5.4M | 480 × 480 | - | - | - | - |
| CogVideoX(Yang et al., 2024j) | STDIT | 3D VAE | - | 2-5B | LAION-5B, COYO-700M | 768 × 1360 | 10s | 160 | - | - |
| MagicVideo(Zhou et al., 2022) | 3D UNet | VideoVAE | CLIP | - | WebVid-10M, HD-VG-130M | 1024 × 1024 | - | - | 1 × A100 | - |
| Make-A-Video(Singer et al., 2022) | UNet | 2D VAE | CLIP | 9.7B | WebVid-10M, HD-VG-100M | 768 × 768 | - | 76 | - | - |
| LVDM(Rakhimov et al., 2023) | 3D UNet | 3D VAE | CLIP | 1.6B | UCF-101, TaiChi | 128 × 128 | - | 1024 | 8 × V100s | 4.5 days |
| Video-LDM(Blattmann et al., 2023b) | 3D UNet | - | - | - | WebVid-10M | 512 × 1024 | ∼ 4.7s | 113 | 2 × A100s | - |
| Latent-shift(Sun et al., 2023c) | ViT | 2D VAE | CLIP ViT-L | 1.5B | WebVid-10M | 256 × 256 | 2-8s | - | 1 × A100 | - |
| MagViT(Yu et al., 2023a) | UNet | 2D VAE | - | 464M | UCF-101 | 128 × 128 | 2-8s | 16 | 1 × V100 | - |
| MagViT-V2(Yu et al., 2024b) | - | 3D VAE | MLM | 307M | UCF-101 | - | - | - | - | - |
| VideoFusion(Zhou et al., 2023) | UNet | - | CLIP | 2B | UCF101, Taichi-HD, SkyTimelapse | 128 × 128 | - | 16 | - | - |
| VideoComposer(Jiang et al., 2023a) | 3D UNet | - | CLIP | - | WebVid-10M | 256 × 256 | - | 16 | - | - |
| InstructVideo(Yuan et al., 2023) | UNet | - | CLIP | - | WebVid-10M | - | - | - | 4 × A100s | - |
| ModelScope(Wang et al., 2023c) | 3D UNet | VQGAN | T5 | ∼1.7B | WebVid | 256 × 256 | - | - | A100 | - |
| HiGen(Qing et al., 2024) | 3D UNet | - | CLIP | - | WebVid-10M | 448 × 256 | - | 32 | 8 × A100s | - |
| Dysen-VDM(Zhao et al., 2023a) | 3D UNet | 2D VAE | CLIP | - | UCF-101, MSRVTT | 256 × 256 | 2s | 16 | 16 × A100s | - |
| VideoGen(Chen & Sun, 2023) | UNet | - | CLIP | - | WebVid-10M | 256 × 256 | - | 16 | 64 × A100s | - |
| Animate-A-Story(He et al., 2023a) | 3D UNet | - | CLIP | - | WebVid-10M | 256 × 256 | - | 16 | - | - |
| AnimateDiff(Guo et al., 2023c) | UNet | 2D VAE | - | - | Web Vid | 1024 × 576 | - | - | -, - | - |
| AnimateDiff-V2 | UNet | - | CLIP | - | - | 1024 × 1024 | - | - | -, - | - |
| SimDA(Xing et al., 2024b) | UNet | - | T5-XXL | 1.1B | WebVid-10M | 256 × 256 | - | - | 1 × A100 | - |
| AnimateLCM(Wang et al., 2024a) | - | - | CLIP | - | UCF-101 | 512 × 512 | 2s | 16 | 1 × A100 | - |
| Snap Video(Menapace et al., 2024) | DiT | 3D VAE | T5-XXL | 3.90B | UCF-101, MSRVTT | 288 × 512 | - | 16 | A100 | - |
| VideoDirGPT(Lin et al., 2024c) | UNet | 2D VAE | CLIP | - | UCF-101, MSR-VTT | - | - | - | 16 × A6000 | - |
| SiT(Ma et al., 2024a) | DiT | 2D VAE | CLIP ViT-L | 675M | LAION-5B | 512 × 512 | - | - | - | - |
| Open-Sora(Zheng et al., 2024c) | STDiT | 3D VAE | T5-XXL | 1.1B | WebVid-10M, Panda-70M HD-VG-130M, MiraData Vript, Inter4K | 1280 × 720 | 2-16s | 16 | 8 × H100s | 3.5k hrs |
| Open-Sora-Plan(Lab & etc., 2024) | DiT | WF-VAE | mT5-XXL | - | Panda-70M | 256 × 256 | - | 25-49 | 8× NPUs | - |
| I4VGEN(Guo et al., 2024a) | UNet | 2D VAE | CLIP | - | - | 512 × 512 | 2-8s | 16 | 1 × V100 | - |
| Zeroscope-v2 | - | - | - | ∼1.7B | - | 1024 × 576 | 3-12s | - | - | - |
| CausVid(Yin et al., 2024) | DiT | - | - | - | - | 640 × 352 | 10s | 128 | 64 × H100s | 2 days |
| RepVideo(Si et al., 2025) | - | - | - | - | - | - | - | - | 32 × H100s | - |
| **Industry commercial models** | | | | | | | | | | |
| GEN-1 | - | - | - | - | - | 1280 × 720 | ∼8s | - | - | - |
| GEN-2 | - | - | - | - | - | 2048 × 1080 | ∼ 4s | - | - | - |
| GEN-3 Alpha | - | - | - | - | - | 4096 × 2160 | Variable | - | - | - |
| Imagen video(Ho et al., 2022a) | UNet | - | T5-XXL | 16.2B | - | Up to 1280 × 768 | Up to 5.3s | - | - | - |
| W.A.L.T(Gupta et al., 2023) | DiT | - | T5-XXL | 3.00B | UCF-101, MSRVTT | 512 × 896 | 2-5s | - | 8× A6000s | - |
| Phenaki(Villegas et al., 2022) | C-ViViT | 2D VAE | T5-XXL | 0.8B | - | 64 × 64 | - | 11-15 | - | - |
| MiracleVision | - | - | - | - | - | 1920 × 1080 | 60s | 1,440 | - | - |
| Lavie(Wang et al., 2023g) | UNet | 2D VAE | CLIP | ∼ 3B | Vimeo25M | 1280 × 2048 | ∼ 10-20s | 16-61 | - | - |
| Seine (Chen et al., 2024e) | UNet | VQGAN | CLIP | - | WebVid-10M | 320 × 512 | - | 16 | - | - |
| VLogger (Zhuang et al., 2024) | UNet | VQVAE | CLIP | 1.3B | WebVid-10M | 320 × 512 | ∼ 2-3s | 16 | - | - |
| Optis (Chen et al., 2024e) | - | - | CLIP | 1.3B | WebVid-10M | 320 × 512 | 5min | 320 | A100 | - |
| Vchitect-2.0 | - | - | - | ∼ 2B | - | 720 × 480 | ∼ 10-20s | - | 8 × A100s | - |
| Pika | - | - | - | - | - | 1920 × 1080 | 3-7s | - | - | - |
| MovieGen | DiT | TAE | MetaCLIP+UL2+ByT5 | 30B | - | 1920 × 1080 | 16s | - | 6,144 H100s | 1000k hrs |
| Show-1 | UNet | - | T5-XXL | - | WebVid-10M | 576 × 320 | - | - | 48 × A100s | - |
| Kling | DiT | 3D VAE | - | - | - | 1920 × 1080 | up to 3min | - | - | - |
| Wanx 2.1 | DiT | - | - | - | - | 1920 × 1080 | - | - | - | - |
| Nova Real | - | - | - | - | - | 1280 × 720 | ∼ 6s | - | - | - |
| Veo-1 | - | - | - | - | - | 1920 × 1080 | ∼ 1min | - | - | - |
| Veo-2 | - | - | - | - | - | 4096 × 2160 | ∼ 2min | - | - | - |
| LumaAI Dream Machine | - | - | - | - | - | 1360 × 752 | ∼ 5s | - | - | - |
| LumaAI Ray 2 | - | - | - | - | - | 1280 × 720 | ∼ 5-9s | - | - | - |
| VideoPoet | - | - | T5 XL | 1.1 B | 270M videos | 896 × 512 8 fps | - | - | - | - |
| Lumiere | UNet | - | T5 XL | - | 30M videos | - 16 fps | 5s | - | - | - |
| Hailuo AI | - | - | - | - | - | 1280 × 720 | 3- 5s | - | - | - |
| Mira | - | - | - | - | MiraData | - | 10s-2min | - | - | - |
| VideoCrafter1(Chen et al., 2023a) | U-ViT | Video VAE | CLIP | - | Laion-coco-600M, Webvid-10M | 1024 × 576 | - | - | - | - |
| VideoCrafter2(Chen et al., 2024b) | U-ViT | Video VAE | - | - | WebVid-10M | 512 × 320 | - | - | - | - |
| Vidu | - | - | - | - | - | 1920 × 1080 | 4-8s | - | - | - |
| EasyAnimate (Xu et al., 2024c) | DiT | 3D VAE | T5-XXL | - | - | 1024 × 576 | - | up to 144 | - | - |
| Mochi 1 (Team, 2024a) | DiT | 3D VAE | T5-XXL | ∼10B | - | 1280 × 720 | ∼ 5.4s | - | 4 × H100s | - |
| Jimeng (Lin et al., 2024b) | - | - | - | - | - | - | 5s | - | - | - |
| Allegro | - | - | - | - | - | 1280 × 720 | - | - | - | - |
| LTX-Video (HaCohen et al., 2024) | DiT | Video-VAE | T5-XXL | - | - | 768 × 512 | 5s | 121 | 1 × H100 | - |
| STIV (Lin et al., 2024b) | DiT | - | - | 8.7B | - | 512 × 512 | - | - | - | - |
| Sora | DiT | 3D VAE | Multiple LLMs | ∼30B | - | 1920 × 1080 | 5-20s | - | - | - |
| HunyuanVideo (Kong et al., 2025) | DiT | 3D VAE | MLLM | ∼13B | - | 1280 × 720 | ∼5s | - | - | - |
| Step-Video-T2V (Ma et al., 2025) | DiT | Video VAE | CLIP, Step-LLM | ∼30B | - | 768 × 768 | - | 204 | H100 | - |
| SkyReels-V1 (SkyReels-AI, 2025) | DiT | 3D VAE | MLLM | - | - | 544 × 960 | - | 97 | - | - |
| Magic-1-For-1 (Yi et al., 2025) | DiT | Video VAE | CLIP, LLM | - | - | - | - | - | - | - |

Table 1: Comparison of modules and parameters in different diffusion generative models and their industry applications. New models can be found in video generation arena leaderboard and vbench leaderboard. (TAE refers to temporal autoencoder, and MLLM refers to multimodal large language model). Papers about related component and dataset: VideoVAE(Zhou et al., 2022), mT5-XXL(Xue, 2020), WebVid(Bain et al., 2021), UCF-101(Soomro et al., 2012), MSR-VTT(Xu et al., 2016), MiraData(Ju et al., 2024), Laion-coco 600M(LAION, 2023) Taichi-HD(Siarohin et al., 2019), SkyTimelapse(Xiong et al., 2018), Panda-70M(Chen et al., 2024d), FaceForensics(Rössler et al., 2018), Inter4K(Stergiou & Poppe, 2023), Vimeo25M(Huang et al., 2023b), MiraData(Zhao et al., 2023d), Vript(Yang et al., 2024a).

allows for more efficient, high-resolution generation. In pixel space, UNet implementations are used for direct denoising, maintaining the input's spatial resolution throughout the process.

Modern implementations of UNet in diffusion models build upon the original by replacing ResNet blocks with Vision Transformer (ViT) layers(Dosovitskiy et al., 2020). ResNet blocks apply 2D-Convolutions, focusing on extracting spatial features, while the ViT blocks introduce spatial self-attention and cross-attention mechanisms. This configuration allows the model to condition generation based on text prompts or specific

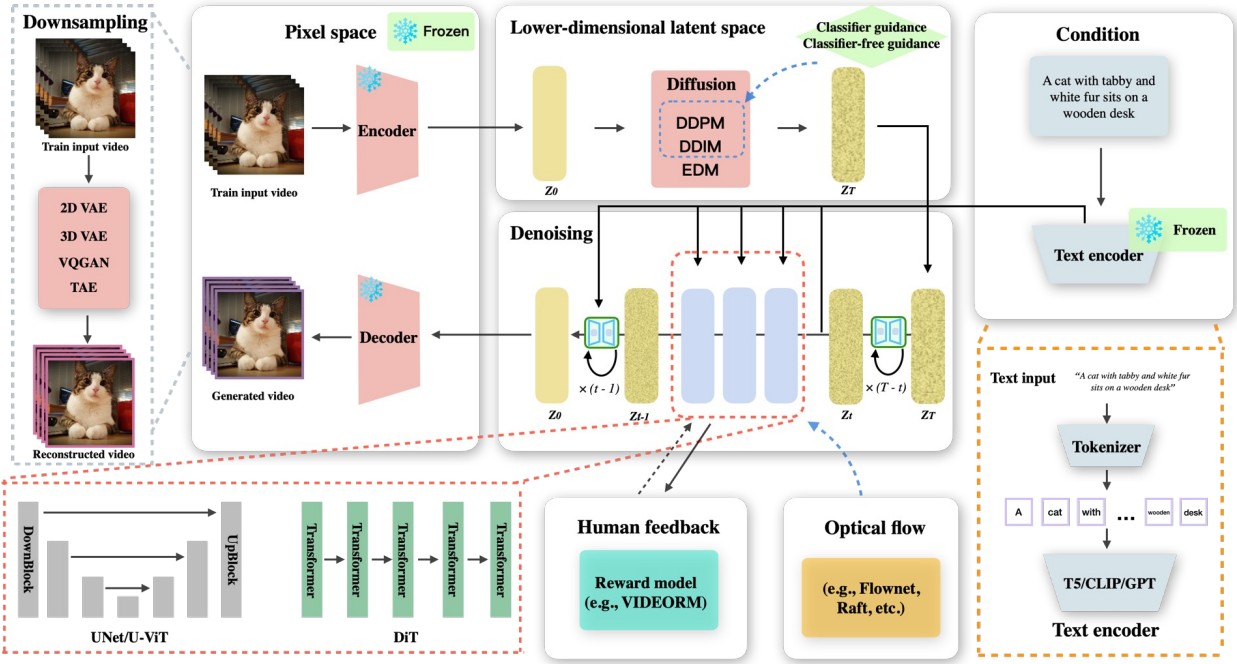

Figure 4: A pipeline for diffusion-based visual content generation leverages a pre-trained variational autoencoder (VAE), such as 2D VAE, 3D VAE, VQGAN, or TAE, to encode input images or videos into a lower-dimensional latent representation. Within this latent space, diffusion models (e.g., DDPM, DDIM, EDM) iteratively introduce noise and employ neural architectures, such as U-Net or Transformer-based models, to learn a denoising process that reconstructs high-fidelity outputs. User-provided textual prompts undergo refinement through large language models (e.g., T5, CLIP, GPT) before being mapped into an embedding space via a trained text encoder. This embedding space serves as a conditioning mechanism, guiding the diffusion process to ensure semantic coherence with the input prompt. Furthermore, the framework integrates optical flow estimation methods (e.g., FlowNet, Raft) to enhance motion consistency in generated video sequences and incorporates human feedback mechanisms (e.g., VIDEORM) to iteratively improve generation quality.

time steps. Matching resolution layers in the encoder and decoder are connected by residual connections, ensuring efficient information flow.

UNet has been further adapted to handle temporal information for video-based diffusion tasks. Extensions like AnimateDiff incorporate temporal attention layers within the UNet to capture dependencies across frames. (An et al., 2023) keep using 2D UNet and enable motion learning by shifting the feature channels along the temporal dimension. Other modifications include transitioning from classic 2D-UNet to spatial-temporal factorized 3D UNet architecture. VDM (Ho et al., 2022c) propose a spatial-temporal factorized 3D UNet by adding temporal blocks for video generation, as a natural extension of the standard image diffusion model. Similarly, (Ho et al., 2022a) builds on the similar video U-Net architecture to the cascaded image diffusion model Imagen Saharia et al. (2022), while (He et al., 2022), MagicVideo(Zhou et al., 2022), and ModelScope(Wang et al., 2023c)apply it to the latent space (Rombach et al., 2022), as shown in Table 1. MoVideo(Liang et al., 2025a) further generates the depth and optical flow of the whole video by utilizing a 3D UNet architecture by adding extra-temporal modules, including temporal convolution and temporal attention layers after spatial convolution and spacial attention layers, improving frame consistency and visual quality for video generation.

To address the challenge of maintaining temporal consistency across video frames, modern video diffusion models incorporate temporal attention mechanisms within their UNet architectures. Temporal attention enables the model to capture dependencies between frames by computing attention weights across the tem-

poral dimension. This mechanism is particularly crucial for video generation tasks where maintaining object consistency and smooth motion transitions is essential. The temporal attention operation can be formulated as:

$$\text{Attention}(Q, K, V) = \text{softmax}\left(\frac{QK^T}{\sqrt{d_k}}\right) V \tag{13}$$

where $Q$, $K$, and $V$ represent query, key, and value tensors derived from different temporal positions, and $d_k$ is the dimension of the key vectors. This formulation allows the model to selectively attend to relevant temporal information, enhancing the quality of generated videos by ensuring coherent motion and consistent object appearances across frames.

One notable implementation is Align Your Latents(Blattmann et al., 2023b), which introduces a sophisticated temporal attention mechanism that operates in the latent space, significantly improving the temporal coherence of generated videos while maintaining high spatial resolution. As illustrated in Figure 5, this approach demonstrates how temporal attention can be effectively integrated into existing diffusion architectures to enhance video generation capabilities.

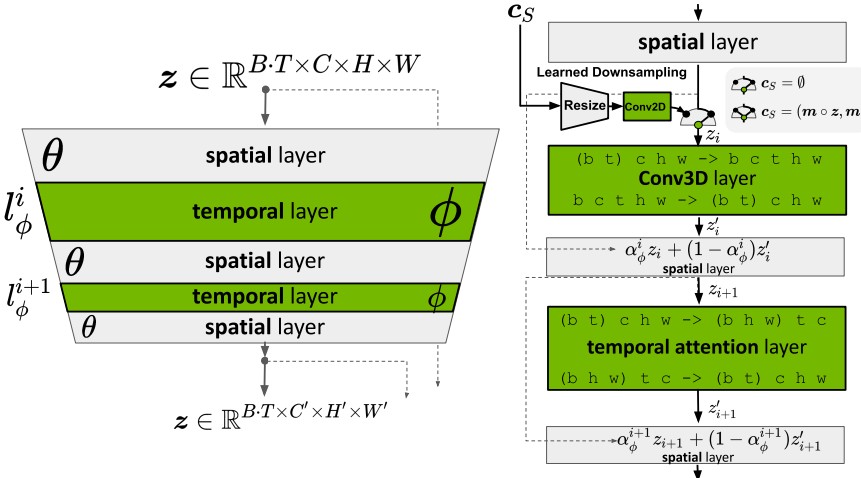

Figure 5: Classic temporal attention design from (Blattmann et al., 2023b), showing how temporal attention mechanisms improve video generation quality by maintaining temporal coherence across frames.

### 2.5.2 Diffusion transformers

The Diffusion Transformer (DiT) replaces UNet's traditional convolutional design with a Vision Transformer (ViT) framework, offering improved spatial attention capabilities suited for generative tasks across images and videos (Peebles & Xie, 2023). DiT achieves this through a 2D patchification process, where individual image frames are divided into patches that are then treated as tokens, allowing the model to capture detailed spatial dependencies within each frame. GenTrone(Chen et al., 2024c) adapted DiT from class to text conditioning and scaled GenTrone from approximately 900M to over 3B parameters demonstrating Transformer-based diffusion models can be used in the visual generative domain. From Table 1, models including CausVid(Yin et al., 2024), SiT(Ma et al., 2024a), and Sora also use DiT in video generation settings. However, for video generation, 2D tokenization alone is insufficient to capture the sequential dependencies critical for temporal coherence across frames. Approaches such as ViViT(Arnab et al., 2021) and STDiT(Zheng et al., 2024c) address this limitation by extending DiT's patchification from 2D to 3D, introducing a 3D tokenization approach that simultaneously captures both spatial and temporal relationships. One representative work is CogVideoX (Hong et al., 2023a), which uses STDiT to model spatial-temporal evolution in video generation, capturing long-range dependencies for improved motion consistency and semantic fidelity. In this structure, consecutive frames are processed as spatiotemporal patches, enabling the model to understand dynamic changes across frames. Building on these approaches, frameworks

like VDT(Lu et al., 2024a) incorporate both causal attention, which restricts attention to previous frames, and sparse causal attention, which further limits the focus to a subset of recent frames. VersVideo(Xiang et al., 2023) incorporated multi-excitation paths for spatial-temporal convolutions with dimension pooling across different axes and multi-expert spatial-temporal attention blocks to further boost the model's spatial-temporal performance instead of simply extending 2D operations with temporal operations to significantly escalate training and inference costs. Text2Performer(Jiang et al., 2023b)proposes ae continuous VQ-diffuser to directly output the continuous pose embeddings for better motion modeling. These mechanisms enhance model's effectiveness for video-based diffusion tasks.

The transition from 2D to 3D patchification represents a fundamental architectural innovation for video generation. ViViT (Arnab et al., 2021) introduces a factorized encoder architecture that separates spatial and temporal modeling. As shown in Figure 6, this model first extracts patch tokens from each frame using spatiotemporal tokenization. A spatial transformer encoder then processes the tokens within each frame independently to model local structure and appearance.

The model passes the outputs of each frame to a temporal transformer that captures how frames relate over time. This two-step design learns both spatial and temporal patterns clearly and separately. For video generation, this setup helps control motion and appearance more precisely and keeps the frames consistent over time. Separating spatial and temporal steps also makes the model easier to train and understand.

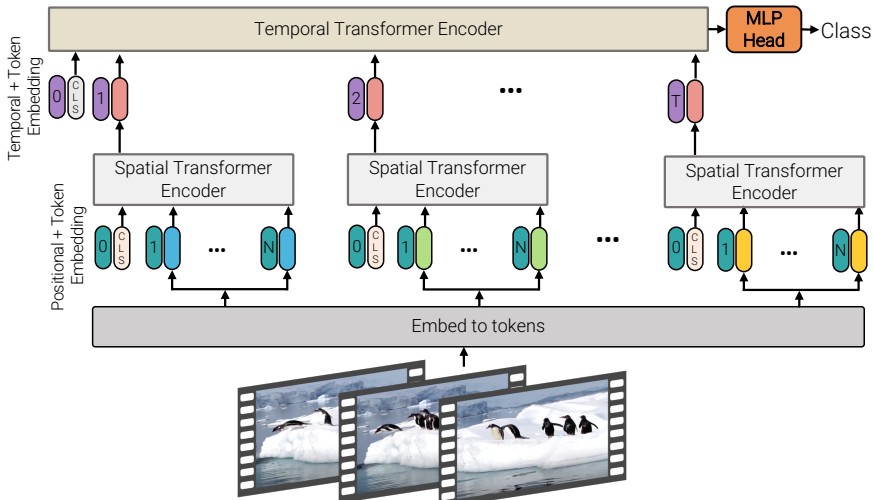

Figure 6: Factorized encoder (Model 2) in ViViT (Arnab et al., 2021). The first transformer encoder models spatial interactions within each frame. The second transformer encoder captures temporal dependencies between frames. This factorized structure enables a late fusion of spatial and temporal information, which supports temporally consistent video generation.

### 2.5.3 VAE for latent space compression

Diffusion and denoising in RGB pixel space, as demonstrated in (Ho et al., 2020; Dhariwal & Nichol, 2021; Saharia et al., 2022; Ramesh et al., 2022; Ho et al., 2022b), require high computational resources, significantly increasing both training cost and inference latency. To mitigate its resource consumption, Latent Diffusion Models (LDM)(Rombach et al., 2022) leverage variational autoencoders (VAEs) to compress images from pixel space into a more compact latent space enabling the diffusion process to occur in this optimized space. This approach enhances both training and inference efficiency by reducing the complexity of the data representation.

Classical VAEs for image and video compression include standard VAEs(Kingma, 2013), quantized versions like VQVAE(Esser et al., 2021), VQGAN(Van Den Oord et al., 2017), and their GAN-enhanced variants, which improve reconstruction quality for higher compression. For example, VideoGPT(Yan et al., 2021) use a 3D-VQVAE in the latent space for video generation. Seine(Chen et al., 2024e) and MAGVIT(Yu et al.,

2023a) integrate 3D-VQGAN with discriminators for latent space encoding to achieve better visual quality. The VAE framework not only compresses effectively but also enable the training of multiple generative tasks across downstream tasks.

By integrating VAE in model building, some image generation models(Ho et al., 2022a; Li et al., 2024b; Podell et al., 2023; Bao et al., 2023b; Peebles & Xie, 2023; Lu et al., 2024c; Ma et al., 2024a; Chen et al., 2023c; Li et al., 2024l; Team, 2024b; Esser et al., 2024) also leverage latent space encoding/decoding, freezing VAE parameters during diffusion training and inference. Some diffusion models, including Make-A-Video(Singer et al., 2022), Imagen(Ho et al., 2022a), Show-1(Zhang et al., 2023a), directly learn pixel distributions generating videos, as shown in Table 1. However, video generation involves both spatial and temporal complexity, leading to higher computational costs. Besides, some representative diffusion video generation models, including Sora(Liu et al., 2024e), Phenaki(Villegas et al., 2022), VideoCrafter(Chen et al., 2023a), Animate-Diff(Guo et al., 2023c), and VideoPoet(Kondratyuk et al., 2023), start using VAE compression in video settings before training and inference in latent space. Many video generation models, including Latte(Ma et al., 2024c), Lavie(Huang et al., 2023b), MagicVideo(Zhou et al., 2022), Align-your-latent(Blattmann et al., 2023b), and AnimateDiff(Guo et al., 2023c), are devrived from Stable Diffusion's image 2D VAEs, as training a full 3D latent space from scratch is challenging. However, temporal compression in 2D VAE-based video models relies on uniform frame sampling, which ignores motion information, leading to low FPS and less smooth video generation.

Some approaches also start to utilize hybrid 2D-3D or fully 3D VAEs. For example, MagViT (Yu et al., 2023a), EasyAnimate (Xu et al., 2024c), Open-Sora (Zheng et al., 2024c; Lab & etc., 2024), and CogVideo-X (Yang et al., 2024j) employ hybrid 2D-3D VAEs, leveraging volumetric encoding for video generation. MAGViT(Yu et al., 2023a) adopting 3D VQGAN structures integrating both 3D and 2D downsampling, and MAGViT-V2(Yu et al., 2024b) utilizing a fully 3D VAE with 3D convolutional encoder and overlapping downsampling for enhanced spatial-temporal fidelity. The MAGVIT architecture improves video compression by combining spatial and temporal information in one model. This helps reduce data size while keeping good video quality.

HunyuanVideo(Kong et al., 2025), Kling, Mochi 1 (Team, 2024a), and SkyReels-V1(SkyReels-AI, 2025) continue leveraging 3D VAEs for compressed training in latent space in industry scenarios. Step-Video-T2V(Ma et al., 2025) and Magic-1-For-1(Yi et al., 2025) introduce novel Video VAE structures for efficient generation. However, training a video VAE independently without ensuring compatibility often leads to a latent space gap, preventing accurate projection into pixel space (Zhao et al., 2024b). CV-VAE (Zhao et al., 2024b) introduces a novel approach using latent space regularization to train a video VAE that extracts continuous latent for generative video models while maintaining compatibility with existing pretrained image and video models. Also, to trade off lower memory and computational cost with slightly lower reconstruction quality, the latest video generation model, Moive Gen(teamMeta, 2024) further uses interleaved 2D-1D convolutional encoders in its VAE. This approach balances efficiency and quality by reducing the redundancy of temporal information while maintaining spatial coherence.

As video generation increasingly demands temporal consistency and causal modeling, recent approaches have introduced causal 3D VAE structures that process frames in a strictly sequential, past-dependent manner. As shown in Figure 7, **CogVideoX** (Yang et al., 2024j) uses a 3D VAE with temporal causality to compress spatial and temporal information, ensuring that each output frame is based only on current and earlier frames.

In the encoder-decoder pipeline, the model applies downsampling across both space and time, then reconstructs frames with mirrored upsampling layers. The VAE operates in a causal, frame-by-frame fashion: it never uses future frames when encoding or decoding the present one, which helps ensure temporal coherence in longer video outputs.

### 2.5.4 Text encoders

In text-conditional visual generation models, the text encoder plays a crucial role in capturing semantic information from input text prompts, directly affecting the generated content. Early text-to-image models used text encoders trained on paired text-image datasets, either trained from scratch(Nichol et al., 2021;

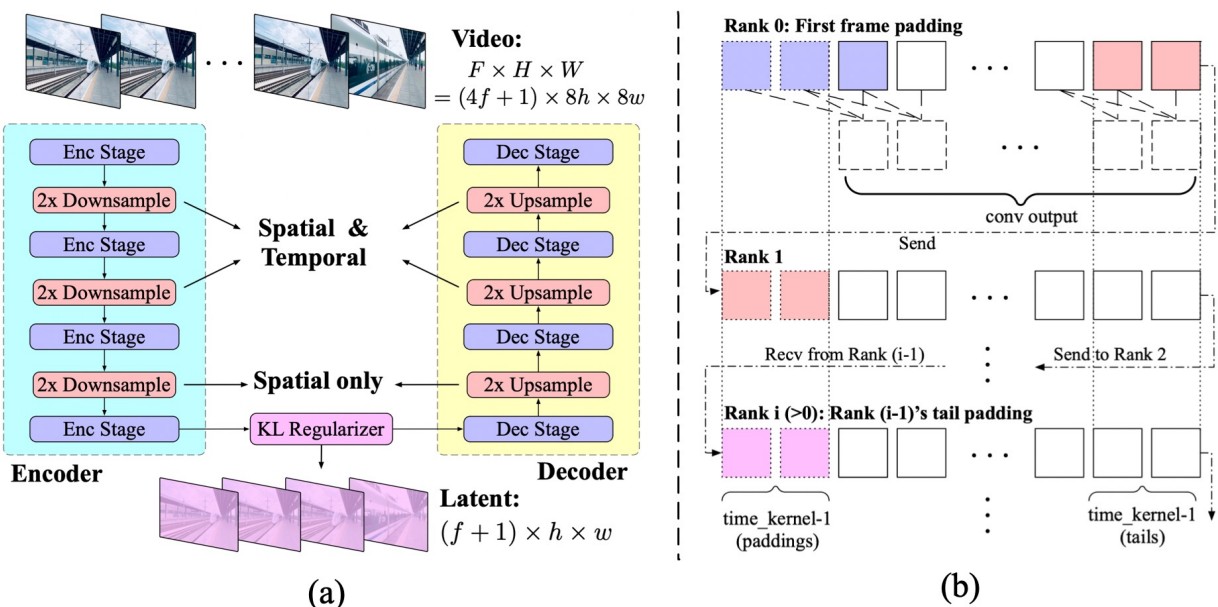

Figure 7: 3D causal VAE structure in CogVideoX (Yang et al., 2024j). (a) The encoder compresses spatial and temporal features, followed by spatial-only layers and a KL regularizer to generate latent representations. The decoder mirrors this structure in reverse. (b) Temporally causal 3D convolutions are applied so each output frame depends only on the current and earlier frames, maintaining motion consistency.

Ramesh et al., 2021) or fine-tuned from pre-trained models such as CLIP(Radford et al., 2021). CLIP, which employs contrastive learning to align text and image embedding spaces, enables the text encoder to effectively represent both visual and linguistic semantics after being trained on large-scale multimodal datasets. Once an input text prompt is tokenized and embedded, it serves as a conditioning mechanism for the diffusion model's generative backbone. CLIP-based text encoders are widely used in text-to-image diffusion models, as shown in Table 1, including DALLE-2(Ramesh et al., 2022), Stable Diffusion(Rombach et al., 2022), DiT(Bao et al., 2023a; Peebles & Xie, 2023), Fit(Lu et al., 2024c), SiT(Ma et al., 2024a), HunyuanDiT(Li et al., 2024l), and Scaling Diffusion (Esser et al., 2024; Labs, 2024). In these models, the text encoder's parameters are often frozen to reduce computational and memory overhead during training, ensuring that most training resources are allocated to the diffusion process itself.

However, CLIP often struggle with understanding detailed text descriptions. To address this, large language models (LLMs) trained on extensive text corpora provide stronger text comprehension and generation capabilities. Imagen(Ho et al., 2022a) compared CLIP with pre-trained LLMs such as BERT(Devlin et al., 2019) and T5(Raffel et al., 2020) as text encoders. Their findings showed that scaling the size of the text encoder improves the quality of text-to-image generation. Notably, (Ho et al., 2022a) found that the T5-XXL encoder significantly enhances image-text fidelity, leading to its adoption in several models such as Latte(Ma et al., 2024c), Open-Sora(Zheng et al., 2024c), and SimDA(Xing et al., 2024b). Some approaches further combine CLIP and T5 to enhance text comprehension. Step-Video-T2V(Ma et al., 2025) integrates CLIP, Step-LLM, and Video-VAE to refine alignment between textual descriptions and generated video content. To improve text encoding, ByT5(Xue et al., 2022b) introduced a byte-level tokenization-free approach, effectively handling diverse text inputs. This has been leveraged in Movie-Gen, where ByT5 combined with MetaCLIP and UL2 jointly enhances text understanding for improved video generation. Multilingual contexts also benefit from LLM-based encoders. Open-Sora-Plan (Lab & etc., 2024) incorporates multilingual T5 (mT5-XXL) alongside WF-VAE, optimizing text representations for video synthesis across diverse linguistic structures. Recent image(Tan et al., 2024c; Team, 2024b) and video(Yang et al., 2024j; Yi et al., 2025), generation models employ large language models such as Baichuan(Yang et al., 2023a), Llama(Touvron et al., 2023a;b), and ChatGLM(Du et al., 2021) to enhance the semantic understanding of complex text. SkyReels-V1(SkyReels-

AI, 2025) and HUnyuanVideo(Kong et al., 2025) further integrate pre-trained Multimodal Large Language Model (MLLM) as their text encoders to enhance visual-text alignment and improve instruction adherence in diffusion models.

## 3 Implementations

In this part, we present a detailed analysis of the datasets, training engineering techniques, and evaluations of video generation.

### 3.1 Datasets

In this part, we review the most popular datasets used for video generation, as summarized in table 2. Specifically, we split the datasets into academic and commercial datasets.

**Academic benchmark datasets**. As a long-standing problem, there exist numerous traditional text-to-video benchmark datasets, such as MSR-VTT (Xu et al., 2016), DiDeMo (Hendricks et al., 2017), LSMDC (Rohrbach et al., 2015), and VATEX (Wang et al., 2019), which consist of videos and corresponding captions collected from the real world or annotators. However, those datasets are always small, low quality, and cover only a small number of different areas, such as sports, cooking, and movies.

Building upon those traditional benchmark datasets, researchers have recently introduced datasets (Sanabria et al., 2018; Zellers et al., 2021; Lee et al., 2021; Chen et al., 2024d; Ju et al., 2024) with a large number of videos. For example, YT-Temporal-180M (Zellers et al., 2021) contains 180M videos collected from YouTube. InternVid (Wang et al., 2023h) takes over 7 million videos lasting nearly 760K hours, yielding 234M video clips accompanied by detailed descriptions of a total of 4.1B words. Also, researchers noticed that video quality, such as resolution, is also important. Inter4k-1k (Stergiou & Poppe, 2023), as a pioneer, introduces 1k videos with a resolution of 4k, enabling the understanding and generation of high-resolution videos.

On the other side, as manually collecting datasets is time-consuming and expensive, researchers started to generate datasets using off-the-shelf models. VidProM (Wang & Yang, 2024) contains 1.67 million unique text-to-video prompts generated by Pika, VideoCraft2 (Chen et al., 2024b), and Text2Video-Zero (Khacha-tryan et al., 2023).

On the other side, other researchers focus on providing comprehensive video generation benchmark datasets (Huang et al., 2024b; Wang & Yang, 2024) from different perspectives, *e.g.*, temporal consistency, style consistency, semantic consistency, and video quality. VBench takes 100 prompts to evaluate video generation models from different perspectives. FETV (Liu et al., 2023e) reuses the texts from the MSR-VTT test set and WebVid for open-domain text-video pairs, guaranteeing the diversity of prompts. It also contains manually written prompts describing scenarios that are unusual in the real world, and cannot be found in existing text-video datasets.

Except for the benchmark dataset for video generation, there are also some datasets targeting face video generation, video inpainting, and video restoration. For example, VFHQ (Xie et al., 2022) is proposed for the face super-resolution, which contains over $16,000$ high-fidelity clips of diverse interview scenarios.

**Commercial benchmark datasets.** There are also some companies providing high-quality image/video-text paired data such as Pond5, Adobe Stock, Shutterstock, Getty, Coverr, Videvo, Depositphotos, Storyblocks, Dissolve, Freepik, Vimeo, and Envato, which are widely used in training video generation methods (Zheng et al., 2024c).

### 3.2 Training engineering

A comprehensive pipeline for data curation, preprocessing, and training is needed to train a large foundation model for video generation. In this part, we present an overview of data preprocessing, training techniques, and acceleration methods.

| Datasets | Modalities | Vision | Text | Duration | Resolution | Domains | # of samples | Sources | License |
|---|---|---|---|---|---|---|---|---|---|
| **Academia datasets** | | | | | | | | | |
| UCF-101 (Soomro et al., 2012) | V, A | Real | - | AVG. 7 s | 320x240 | - | V (13,320) | YouTube | CC-BY |
| Kinetics-400 (Carreira & Zisserman, 2017) | V, A | Real | - | AVG. 10 s | - | - | V (306,245) | YouTube | - |
| BSCV (Liu et al., 2023c) | V | Real | - | - | 480P | - | V (28 k) | - | - |
| VFHQ (Xie et al., 2022) | V | Real | - | - | 700x700 | - | V (16,827) | FFHQ, VoxCeleb1 | - |
| DMLab-40k (Yan et al., 2023b) | V | Synthetic | - | - | - | 3D Rendered | V (40 k) | - | - |
| Habitat (Yan et al., 2023b) | V | Synthetic | - | - | - | 3D Rendered | V (200 k) | - | - |
| Minecraft (Yan et al., 2023b) | V | Synthetic | - | - | - | 3D Rendered | V (200 k) | - | - |
| DEVIL (Szeto & Corso, 2022) | V | Real | - | - | - | Camera Infos | V (1,250) | Flickr | MIT |
| Inter4K-1k (Stergiou & Poppe, 2023) | V | Real | - | AVG. 5 s | 4K | - | V (1 k) | - | - |
| MSR-VTT (Xu et al., 2016) | V, T, A | Real | Real | AVG. 15 s | 320x240 | - | T/V (10 k) | Youtube | - |
| DiDeMo (Hendricks et al., 2017) | V, T, A | Real | Real | AVG. 7 s | - | - | T/V (27 k) | Flickr | BSD 2-Clause |
| LSMDC (Rohrbach et al., 2015) | V, T, A | Real | Real | AVG. 5 s | 1080p | - | T/V (118 k) | - | - |
| VATEX (Wang et al., 2019) | V, T, A | Real | Real | AVG. 10 s | - | - | T/V (41 k) | Youtube | CC-BY-4.0 |
| YouCook2 (Zhou et al., 2018a) | V, T, A | Real | Real | AVG. 5 mins | - | - | T/V (14 k) | Youtube | - |
| How2 (Sanabria et al., 2018) | V, T, A | Real | Real | AVG. 90 s | - | - | T/V (79 k) | Youtube | Creative Commons BY-SA 4.0 |
| ActivityNet Caption (Heilbron et al., 2015) | V, T, A | Real | Real | AVG. 120 s | - | - | T/V (100 k) | Youtube | - |
| VideoCC3M (Nagrani et al., 2022) | V, T, A | Real | Real | AVG. 10 s | - | - | T/V (10.3 M) | - | - |
| WebVid10M (Bain et al., 2021) | V, T | Real | Real | AVG. 18 s | 360p | - | T/V (10.7 M) | - | AGPL-3.0 |
| WTS70M (Stroud et al., 2021) | V, T | Real | Real | AVG. 10 s | - | - | T/V (70 M) | - | - |
| HowTo100M (Miech et al., 2019) | V, T, A | Real | Auto Captioning | AVG. 4 s | 240p | - | T/V (136 M) | Youtube | Apache License 2.0 |
| HD-VILA-100M (Xue et al., 2022a) | V, T | Real | Auto Captioning | AVG. 13 s | 720p | - | T/V (100 M) | - | See license |
| YT-Temporal-180M (Zellers et al., 2021) | V, T, A | Real | Real | - | - | - | T/V (180 M) | Youtube | - |
| ACAV100M (Lee et al., 2021) | V, T, A | Real | Real | AVG. 10 s | - | - | T/V (100 M) | - | MIT License |
| Vript-400k (Yang et al., 2024a) | V, T, A | Real | Real | AVG. 11 s | 720p | - | T/V (420 k) | - | - |
| VidProM (Wang & Yang, 2024) | V, T | Synthetic | Real | AVG. 2 s | - | - | T (1.67 M), V (6.69 M) | - | CC-BY-NC 4.0 |
| FETV (Liu et al., 2023e) | V, T | Real | Real | - | - | - | T (619), V (541) | MSR-VTT and WebVid | CC-BY-NC 4.0 |
| InternVid (Wang et al., 2023h) | V, T | Real + Synthetic | Auto Captioning | AVG. 12 s | 720p | - | T/V (234 M) | - | CC BY-NC-SA 4.0 |
| AIGCBench (Fan et al., 2024) | I, V, T | Real | Real | - | - | - | T/I (2,928), T/V (1,000) | - | Apache License 2.0 |
| AVE (Argaw et al., 2022) | V, T | Real | Real | AVG. 4 s | - | - | V (196k ) | - | - |
| Panda-70M (Chen et al., 2024d) | V, T | Real | Auto Captioning | AVG. 9 s | 720p | - | T/V (70 M) | - | See license |
| HD-VG-130M (Wang et al., 2024f) | V, T | Real | Auto Captioning | - | 720p | - | T/V (130 M) | - | See license |
| MiraData-77k (Ju et al., 2024) | V, T | Real | Auto Captioning | AVG. 72 s | 720p | Camera Infos | V (330 k) | - | GPL-v3 |
| LAION-AESTHETICS 6.5+ | I, T | Synthetic | Real | - | - | - | T/I (625 k) | - | - |
| Animate bench | I, T | Synthetic | Real | - | - | - | T/I (105) | - | Apache License 2.0 |
| DrawBench (Saharia et al., 2022) | I, T | Real | Real | - | - | - | I/T (200) | - | - |
| PartiPrompts (Yu et al., 2022b) | I, T | Real | Real | - | - | - | T (1.6 k) | - | Apache License 2.0 |
| **Commercial datasets** | | | | | | | | | |
| Midjourney-v5-1.7M | I, T | Synthetic | Real | - | - | - | T/I (1.7 M) | - | Apache License 2.0 |
| Midjourney-Kaggle-Clean | I, T | Synthetic | Real | - | - | - | T/I (250 k) | - | cc0-1.0 |
| Unsplash-lite | I, T | Synthetic | Real | - | - | - | T/I (250 k) | - | See license |
| Mixkit | V, T | Real | Real | AVG. 18 s | 720p | - | T/V (1,234) | - | Commercial and non-commercial |
| Pixabay | I, V, T | Real | Real | AVG. 25 s | 720p | - | T/V (31,616) | - | Commercial and non-commercial |
| Pexels-400k | V, T | Real | Real | - | 720p | - | T/V (400,476) | - | MIT |

Table 2: The overview of most popular datasets used in training video generation models. We also include image datasets as they are usually used in training. "I", "V", "T", and "A" represent image, video, text, and audio. Other commercial datasets include those released by Pond5, Adobe Stock, Shutterstock, Getty, Coverr, Videvo, Depositphotos, Storyblocks, Dissolve, Freepik, Vimeo, and Envato.

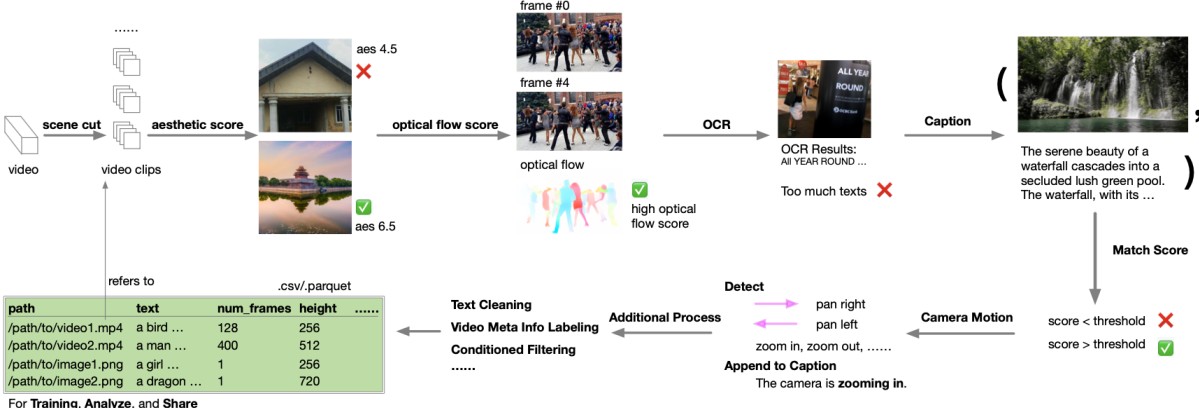

Figure 8: The data preprocessing pipeline of Open-Sora (Zheng et al., 2024c).

**Data preprocessing**. Instead of directly using the original data from the dataset, more and more methods (Zheng et al., 2024c; Li et al., 2024l; Yang et al., 2024j; Blattmann et al., 2023a) have started to select, filter, and enhance video-text paired data before training the model as shown in fig. 8. OpenSora (Zheng et al., 2024c), as a representative open-sourced method, first splits videos into shorter clips using scene detection (Ravi et al., 2025). Later, clips are evaluated using the aesthetic and optical flow scores, and the clips with low scores are removed from training. After that, the qualified clips are captioned using machine captioners (Hong et al., 2024b; Li et al., 2025a; Xu et al., 2024d) and a matching score will be employed to evaluate the alignment between clips and their captions. Finally, samples with adequate scores (high

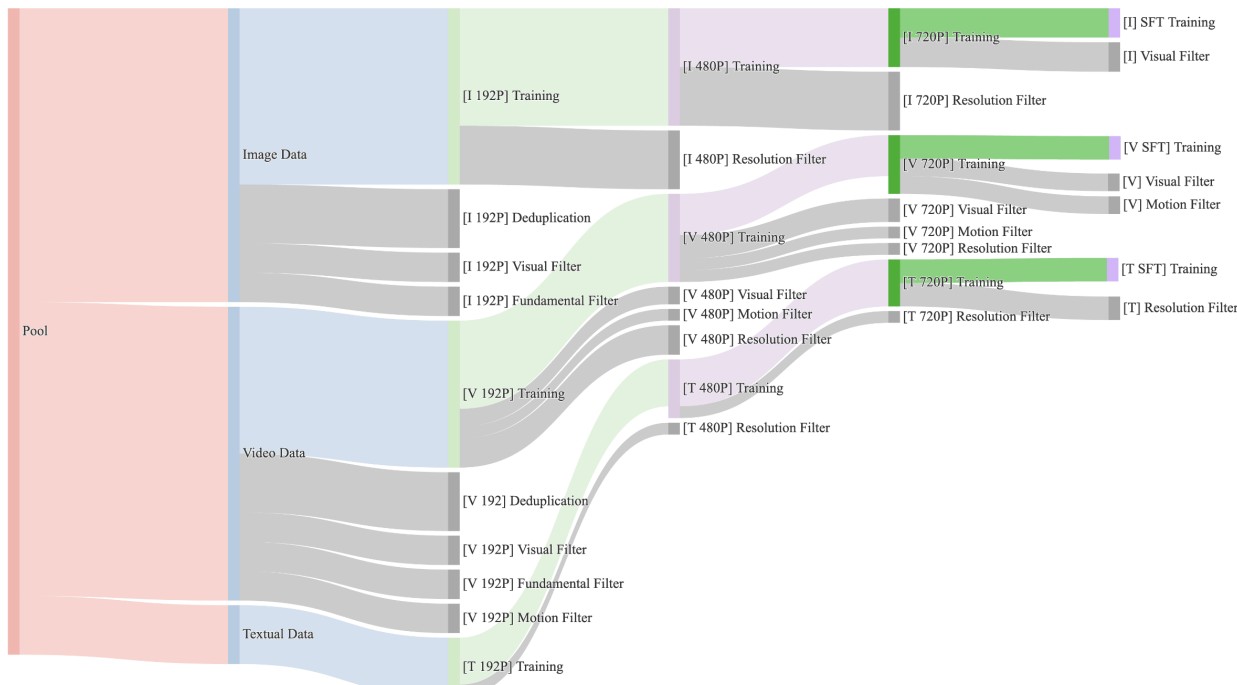

Figure 9: The data preprocessing pipeline of WAN (Wan et al., 2025). WAN first splits data into image, video, and text data. Then, different stages of filters are applied to the data for further proprocessing. Fianlly, the data is splited into pretraining data and SFT data for imrpoving the quality of generated video data.

aesthetic quality, large video motion, and strong semantic consistency) are used for training the model. The overall pipeline is presented in fig. 8. Similarly, SVD (Blattmann et al., 2023a) filters out samples with low text-video alignment evaluated by CoCa (Yu et al., 2022a), static videos (evaluated by optical flow), and high text overlays.

A recent framework, WAN (Wan et al., 2025), as shown in fig. 9, employs a novel data pipeline which includes four fundamental dimensions for data collection and selection. For pretraining data, it evaluates data from fundamental dimensions (text detection, blur detection, synthetic data detection, and etc.), visual quality (including data clustering and scoring), motion quality (which splits data with optimal motion, medium-quality motion, camera-driven motion, low-quality motion, and shaky camera footage), and visual text data. For post-training data preprocessing, WAN improves the quality of image and video data separately. Similarly, to meet the needs of different stages of pre-training, teamMeta (2024) curates 3 subsets of pre-training data with progressively stricter visual, motion, and content thresholds, which leads to more stable training process. While MovieGen and WAN apply post-training to improve the motion and aesthetic quality of the generated videos by finetuning the pre-trained model on a small finetuning set of selected videos, Cosmos WFM (NVIDIA et al., 2025) is fine-tuned in post training to support diverse Physical AI applications, such as autonomous driving, with dedicated post-training datasets.

**Training techniques.** Video generation models (Blattmann et al., 2023a; Zheng et al., 2024c) are usually initialized with image models and pretrained with image data. Then, the models are pretrained with video data for learning efficient video representations. After that, models are trained using high-quality data to form the final video generation models. However, due to the difficulty of the video generation problem, different training techniques (Blattmann et al., 2023a; Xing et al., 2023; Hong et al., 2023a; Yang et al., 2024j) are proposed. For example, CogVideoX (Yang et al., 2024j) employs a multi-resolution frame pack strategy to enable the generation of videos with varied lengths. And a progressive training strategy is employed to generate videos of different resolutions. This progressive training strategy is also employed by other works,

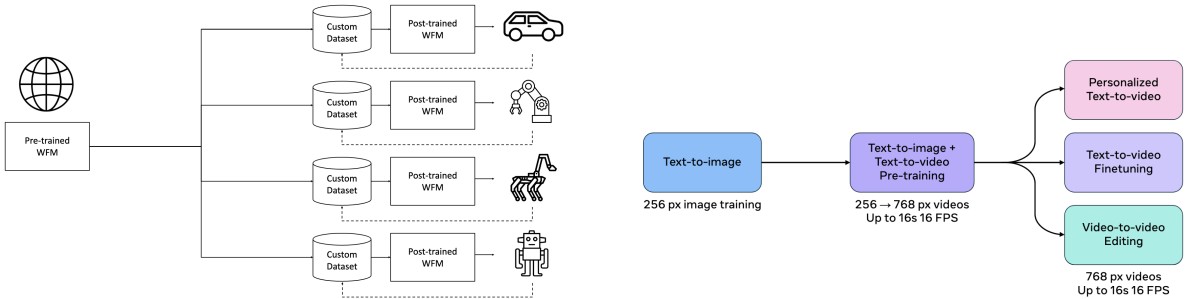

(a) Training pipeline of Cosmos (NVIDIA et al., 2025).   (b) Training pipeline of Movie Gen (teamMeta, 2024).

Figure 10: Training pipelines of Cosmos (NVIDIA et al., 2025) and Movie Gen (teamMeta, 2024). Cosmos first pretrains models and then finetunes on different downstream tasks for better performance while Movie Gen finetunes to achieve better quality of generated videos.

such as Hunyuan-Video (Kong et al., 2025). On the other side, to scale up the training of video generation models, multi-node parallelization and VRAM-efficient scheduling are utilized (Yang et al., 2024j; Zheng et al., 2024c). Moreover, as post training has shined the lights on vision-language models (Cheng et al., 2025; NVIDIA et al., 2025; Yamaguchi et al., 2025), more and more studies (Wan et al., 2025; teamMeta, 2024) include post-training to further improve the quality of generated video, as shown in fig. 10. A practical scenario for post-training is to finetune the pretrained to suit the downstream tasks (NVIDIA et al., 2025), such as autonomous driving and robot manipulation.

**Video generation acceleration techniques.** Due to the high computational requirements of training and inferencing video generation methods, researchers have developed several acceleration techniques. For example, Open-Sora (Zheng et al., 2024c) employs flash attention (Dao et al., 2022), ZeRO (Rajbhandari et al., 2020), and gradient checkpoint for kernel optimization, hybrid parallelism, and larger batch size training. Moreover, Open-Sora employs efficient STDiT as the vision encoder while preprocessing text and video data for acceleration. On the other side, parameter-efficient fine-tuning (PEFT) (Geng et al., 2024a; Jia et al., 2022), which is widely applied in training large foundation (language and vision-language) models (Li et al., 2025a; Touvron et al., 2023a;b), could be used in video generation for training acceleration (Xing et al., 2024b; Pan et al., 2022). We also include a detailed discussion on long video generation acceleration techniques in section 4.5.

### 3.3 Evaluation metrics and benchmarking findings

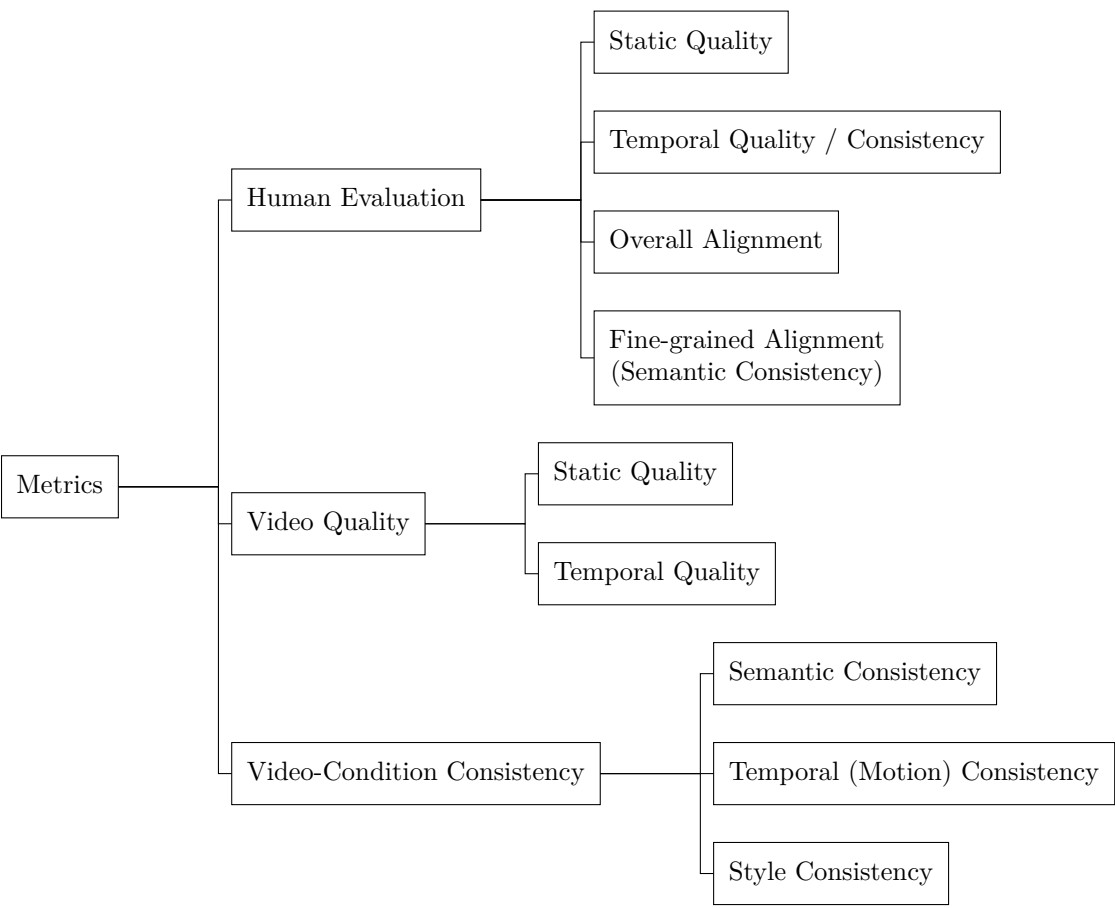

Existing evaluation metrics include human evaluation and auto evaluation. Human evaluation (Liu et al., 2023e; Huang et al., 2024b) always includes several aspects such as static quality, temporal quality (consistency), overall alignment, and fine-grained alignment (semantic consistency). And human evaluation is always employed to test the alignment between auto-evaluation and human evaluators, such as VBench (Huang et al., 2024b), which uses 16 human evaluations on 16 different dimensions mainly focusing on temporal and semantic consistency. A human-verified metric is STREAM (Kim et al., 2023c), testified under the same dimensions using auto-evaluation and human evaluation.

On the other side, as auto-evaluation (Huang et al., 2024b;c; Fan et al., 2024) is cheaper and faster, it is widely applied to evaluate video quality and conditioned quality (semantic consistency and temporal consistency). Video quality usually only contains video as the input of metrics while conditioned quality also uses other modal inputs such as image, text, or audio to evaluate the generated video.

Video quality consists of static quality and temporal/motion quality. 1) For static quality, AIGCBench (Fan et al., 2024) employs a pre-trained optical flow estimation model, *i.e.*, RAFT (Teed & Deng, 2020), and calculates the mean square of the average flow score between adjacent frames from the generated video for estimating the motion effects as Flow-Square-Mean. Dover (Wu et al., 2023a) is also employed for evaluating the static quality from aesthetic and technical perspectives. Moreover, the **number of video frames** is used for accessing the quality. On the other side, FETV (Liu et al., 2023e) employs FID (Heusel et al., 2017) and FVD (Unterthiner et al., 2019) metrics. 2) For temporal/motion quality, AIGCBench assesses it by calculating the average similarity between adjacent frames using CLIP (Radford et al., 2021) noted as GenVideo Clip (Adjacent frames). VBench (Huang et al., 2024b) evaluates it by subject, background, temporal flickering, motion, dynamic degree, aesthetic quality, and imaging quality using multiple pre-trained models including DINO, CLIP, Amt, RAFT, and LAION. Bonneel et al. (2015) proposes a warping

error for improving the consistency between frames. Except for those methods, STREAM (Kim et al., 2023c) is proposed to avoid the problem of FVD, which has a stronger emphasis on the spatial aspect than the temporal naturalness of video and demonstrates considerable instability and diverges from human evaluations. It evaluates temporal and semantic quality at the same time.

Besides those metrics, more and more researchers start to improving motion consistency and aesthetics of generated videos. 1) For accessing motion smoothness, WAN (Wan et al., 2025) employs Qwen2-VL (Bai et al., 2023) with crafted prompts. Similarly, Movie Gen (teamMeta, 2024) accesses the motion completeness and motion naturalness for better evaluating the quality of generated videos. 2) While the motion quality of video evaluates how smooth the video is, the realness and aesthetics evaluate the model's ability to generate photorealistic videos with aesthetically pleasing content, lighting, color, style, *etc.* To access them, Movie Gen recruits a group of human evaluator to measure which of the videos being compared most closely resembles a real video and which of the generated videos has more interesting and compelling content, lighting, color, and camera effects.

The metrics for evaluating conditioned quality can be categorized into semantic consistency, temporal consistency, and style consistency. 1) In semantic consistency, for image/video-video metrics, AIGCBench (Fan et al., 2024) employs Mean Squared Error (MSE First) and Structural Similarity Index Measure (SSIM First) for evaluating the first generated frames. To further understand how the semantics are preserved across frames, AIGCBench calculates the cosine similarity between the input image and each frame of generated video using CLIP (Image-GenVideo CLIP). Moreover, GenVideo-RefVideo SSIM is also employed to measure the spatial structural similarity of the generated videos to the reference videos by calculating the SSIM (Structural Similarity Index Measure) between the corresponding frames of the generated and reference videos as the referenced video is at hand. For Text-to-Video metrics, AIGCBench employs GenVideo-Text Clip and GenVideo-RefVideo CLIP (Keyframes), calculating the similarity between the input text / referenced video and keyframes from generated videos using CLIP. Similar to AIGCBench, FETV (Liu et al., 2023e) propose to use stronger models, such as CLIP finetuned on MSR-VTT, BLIP, and UMP, and VQA models for evaluating the semantic consistency. Moreover, FETV also introduces a novel QA-based metric (Otter-VQA). This metric generates yes-no questions on key elements from the text prompt via the Vicuna model and calculates the average number of questions that receive a positive answer by feeding the generated (or ground-truth) videos and questions into the Video LLM as the score. VBench (Huang et al., 2024b) evaluates the semantic consistency by objects, humans, color, relationships, scene, appearance, and temporal style using ViCLIP, CLIP, Tag2Text, UMT, and GRiT. 2) For temporal consistency, researchers usually directly employ the temporal quality metrics. Recently, TC-bench (Feng et al., 2024c) proposed to focus on the transition of objects, relations, and background for better temporal consistency. 3) For style consistency, VBench employs CLIP and ViCLIP to assess the similarity between generated videos and the style description.

Except for those metrics, VBench (fig. 11) and Video Generation Arena also set up online evaluation platform, providing an accessible leaderboard for researchers.

### 3.4 Industry solutions

As shown Backbone column in Table 1, video generation industry has evolved significantly with the adoption of diffusion models that leverage advanced architectures like DiT, UNet, and U-ViT. DiT (Diffusion Transformer) is widely used in models like W.A.L.T(Gupta et al., 2023), MovieGen, Kling, and Mochi 1(Team, 2024a), excelling in capturing long-range dependencies for sharp, coherent visuals in video generation tasks. UNet, utilized in models like Imagen video, Lavie(Wang et al., 2023g), Seine(Chen et al., 2024e), and VLogger(Zhuang et al., 2024), is known for its ability to handle both local and global features, making it ideal for high-resolution video generation. U-ViT (U-Transformer for video) is another backbone seen in models like VideoCrafter1(Chen et al., 2023a) and VideoCrafter2(Chen et al., 2024b), effective at processing both spatial and temporal dependencies for more accurate video generation. These architectures continue to shape the industry's approach to advanced video generation.

Variational Autoencoders (VAE) are widely used in industry to learn latent video representations. As shown VAE column in Table 1, models such as HunyuanVideo(Kong et al., 2025), Mochi 1(Team, 2024a), and

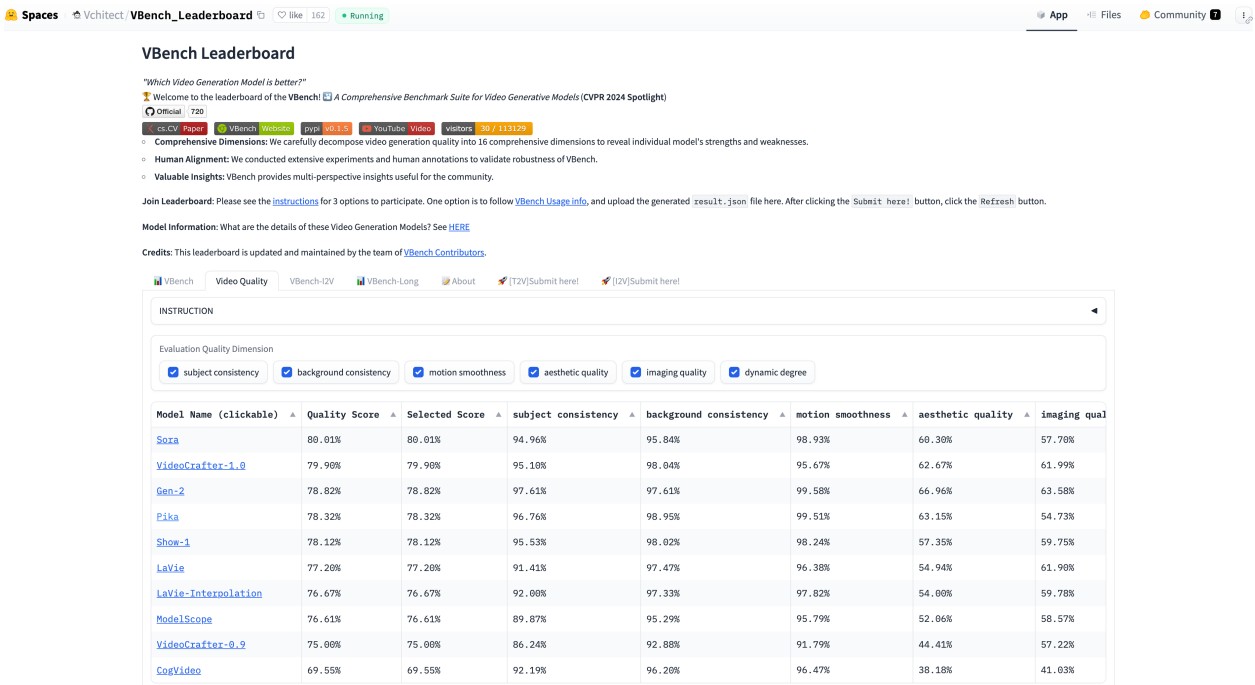

Figure 11: The VBench leader board at Feb. 3rd, 2025. SORA still scores the highest on some of the metrics, *e.g.*, quality score and selected score.

Kling utilize 3D VAE for temporal consistency. On the other hand, models like Lavie(Wang et al., 2023g) and Phenaki(Villegas et al., 2022) utilize 2D VAEs, which offer greater computational efficiency but are less effective at capturing the temporal dynamics of video content. Meanwhile, VQ-VAE, used in models like VLogger(Zhuang et al., 2024) and Vchitect-2.0, introduces vector quantization to improve efficiency by discretizing the latent space. Additionally, Video VAE architectures, such as in VideoCrafter(Chen et al., 2023a) and LTX-Video(HaCohen et al., 2024), are specifically designed for video generation, optimizing the ability to handle both spatial and temporal complexities. The choice of VAE architecture thus directly influences the quality, efficiency, and temporal coherence of video generation.

Text encoders, such as T5-XXL, are essential for aligning video with textual descriptions. Based on the column text encoder in the Table 1, models like Imagen video, W.A.L.T.(Gupta et al., 2023), and MovieGen use T5-XXL for precise video-text alignment, enhancing generative capabilities. Meanwhile, models like Lavie(Wang et al., 2023g) and Seine(Chen et al., 2024e) leverage CLIP-based encoders, which offer flexibility for generating both short clips and full-length videos. T5-XXL excels in large-scale generation, while CLIP provides versatility for diverse visual styles and content.

The scale of models varies, as shown in the parameter column of Table 1, with larger models like Sora (with $\sim 30B$ parameters) and MovieGen (with $\sim 30B$ parameters) offering scalability for large-scale production. Smaller models including Vchitect-2.0 and VideoPoet maintain strong performance while using fewer parameters (1B–2B) to balance performance and computational cost.

Training datasets such as WebVid-10M(Bain et al., 2021) and Vimeo25M(Huang et al., 2023b) are commonly used for large-scale models like Seine(Chen et al., 2024e), Lavie(Wang et al., 2023g) and Show-1 from Table 1, ensuring diverse outputs. Models including Lumiereand VideoPoet choose to gather video and image data directly from public websites, rather than relying on pre-existing datasets.

Resolution plays a key role in video output quality. As shown in the resolution coulmn in the Table 1, GEN-3 Alpha and Sora support high resolutions like $4090 \times 2160$ and $1920 \times 1080$, ideal for cinematic production. In contrast, models like Seine(Chen et al., 2024e) and Phenaki(Villegas et al., 2022) offer lower resolutions like $320 \times 512$ or $64 \times 64$, balancing quality and computational efficiency.

Video duration is also a key consideration. Based on the duration column in the Table 1, W.A.L.T(Gupta et al., 2023) and VLogger(Zhuang et al., 2024) excel in generating short-duration videos for platforms such as social media, while models such as Kling and Mira are capable of creating longer videos, suited for film production or extended content.

GPU requirements for training and inference vary. As shown in the GPU size and GPU hours column in the Table 1, Large models such as Show-1 and MovieGen need substantial GPU resources, with Show-1 requiring around 48 A100 GPUs. Smaller models like LTX-Video(HaCohen et al., 2024) can achieve faster processing, offering practical real-time video generation for commercial use.

The video generation industry continues to grow with models such as MovieGen and Sora catering to high-end production, while consumer-focused models like VLogger(Zhuang et al., 2024) and EasyAnimate (Xu et al., 2024c) offer fast solutions for short-form video content. The challenge remains to optimize video quality while managing computational resources for broader accessibility.

## 4 Applications

### 4.1 Conditions

In this section, we examine a variety of "conditions" that researchers use to guide the video generation process. Unlike purely unconditional or text-based generation, which can lack specific control or produce unsatisfactory temporal consistency, leveraging additional forms of conditioning provides more precise and robust ways to steer the synthesis. These conditions can stem from many sources—images, spatial constraints, camera parameters, audio signals, high-level editing instructions, or even physical simulations—and can serve different roles, from specifying global traits like style and identity to prescribing detailed frame-by-frame motion.

We organize these conditions into subtopics based on the type of guidance they offer. First, we discuss image-based methods (Image condition 4.1.1), where a single reference image or a sequence of images conveys both coarse semantic cues and fine-grained details for video generation. We then explore spatial conditions (Spatial condition 4.1.2), such as trajectories, bounding boxes, or scene layouts, which grant users exact control over object movement and scene composition. Next, we address camera parameter inputs (Camera parameter condition 4.1.3) for 3D-consistent view synthesis and cinematic manipulation. We also cover audio-based approaches (Audio condition 4.1.4) for synchronizing speech, music, or other sound cues with visual content. Finally, we examine methods that focus on high-level editing (High-level video condition 4.1.5) and specialized application constraints (Other conditions 4.1.6), including incorporating physics-based realism, working with physiological signals, and utilizing timestamp information as conditions. Through these diverse forms of conditioning, researchers aim to create video generation systems that balance creative freedom with strong user control, opening the door to a wide range of potential applications.

### 4.1.1 Image condition

Controlling video generation with images is a crucial capability, as images offer visually intuitive and fine-grained control over content that text alone cannot achieve. This functionality serves as a foundational building block for various video applications, such as identity-preserving video generation and image animation.

Depending on how the conditioning image is integrated into the diffusion model, the methods can be broadly categorized into two types: semantic-level image conditioning and fine-grained image conditioning.

**Semantic-level conditioning.** Similar to text-conditioned video diffusion, semantic-level image conditioning provides high-level coarse guidance for video generation. This is typically achieved by encoding the input image into a low-dimensional feature that retains rich semantic information, such as identity. These features are then fed into the video model through cross-attention layers in a manner similar to how text embeddings are incorporated into the network. For instance, DreamVideo (Wei et al., 2024c) employs an identity adapter to encode the identity of the conditioning image, while ID-Animator (He et al., 2024d) uses a face adapter to capture the facial identity of a human face. Additionally, CLIP features are commonly utilized to extract

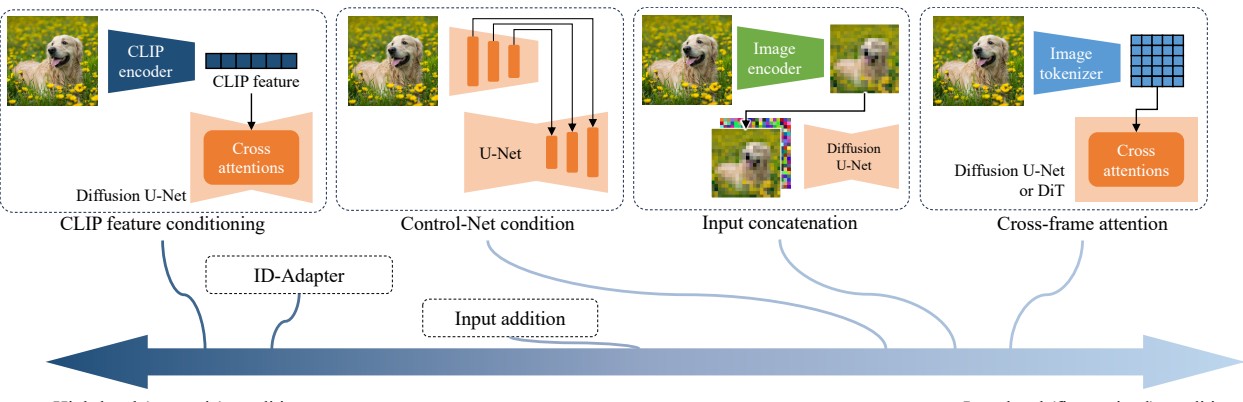

Figure 12: Methods for controlling video generation using image-based conditions. We categorize these methods along a spectrum ranging from high-level semantic conditions to low-level conditions. Additionally, we illustrate four classic approaches commonly used to condition input images.

global information from the image, which is then used to condition video generation (Zhang et al., 2023d; Xing et al., 2023; Wang et al., 2023e; Xu et al., 2024b; Jiang et al., 2024b; Zhao et al., 2024a).

**Fine-grained-level conditioning.** High-level control is not always desirable. In some cases, we aim to fully control every detail of the generated video. At the extreme, fine-grained conditioning means the conditioning image directly corresponds to one of the frames in the generated video. This is achieved by exposing the network to the fine-grained features of the conditioning image.

The most straightforward approach is to concatenate the image (or its latent representation) with the noise input. This concatenation typically occurs along the channel axis, with the conditioning image often serving as the first frame of the training video. This approach is widely adopted in many video generation models, including SVD (Blattmann et al., 2023a), VDT (Lu et al., 2024a), Motion I2V (Shi et al., 2024), AtomoVideo (Gong et al., 2024), and CogVideoX (Yang et al., 2024j). Since the model predicts multiple frames simultaneously and the conditioning image is usually a single frame, the image can either be repeated across all frames or temporally padded with black pixels, accompanied by an additional mask channel to indicate the validity of the condition for each frame. Notably, this concatenation-based approach is also widely used in image generation Brooks et al. (2023); Ke et al. (2024); Zeng et al. (2024c); He et al. (2024c).

Concatenation can also be done along the temporal axis (Shi et al., 2024; Ma et al., 2024b; Zhang et al., 2024j; Gu et al., 2024a; Dai et al., 2023), where the conditioning image is explicitly set as the first frame of the predictions. This method avoids the need for modifications to the model architecture.

While concatenation is often used for conditioning on the first frame, Make Pixel Dance (Zeng et al., 2024a) demonstrates that conditioning on both the first and last frames enables the model to implicitly learn large-scale motion generation, as shown in Fig. 13

MagDiff (Zhao et al., 2024a) designs a pyramid structure to encode conditioning images, effectively capturing contextual information at multiple scales and enhancing the robustness of the input.

PIA (Zhang et al., 2023f) introduces a plug-and-play image conditioning approach that enables seamless conditioning across various personalized text-to-image models without the need for fine-tuning. This is achieved by non-invasively injecting conditioning features into the first convolutional layer of the U-Net.

Image conditioning can also be implemented using a ControlNet-like approach (Zhang et al., 2023b). For instance, DreamVideo (Wang et al., 2023a) incorporates image conditioning by forwarding the conditioning image through the downsampling blocks of the diffusion U-Net and injecting the resulting intermediate feature maps into the video model. Similarly, SparseCtrl (Guo et al., 2024b) employs a ControlNet-like structure but extends it to support temporally sparse conditions, enabling the use of one or more frames as conditioning inputs.

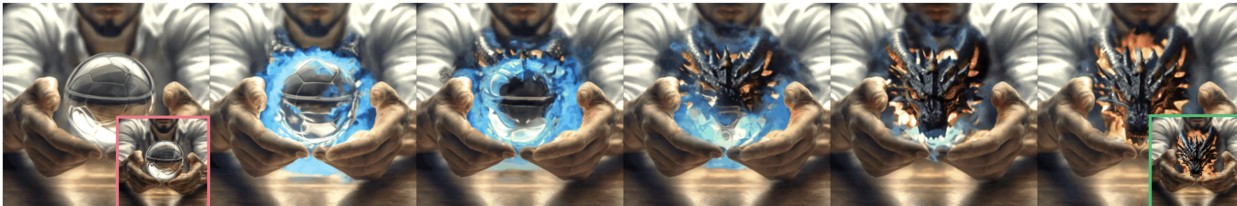

"The ball is in blue fire and turns into a dragon in fire."

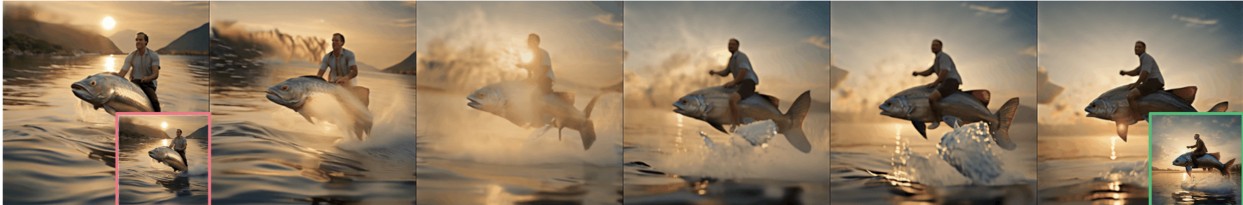

"A man rides on a huge fish, flying from the water into the sky."

Figure 13: Make Pixel Dance Zeng et al. (2024a) shows that by conditioning both on the first and last frame, the model implicitly learns to generate large-scale motion.

I2V-Adapter (Guo et al., 2023b) highlights a potential challenge of concatenating: when fine-tuning a video model initialized from a text-to-video model, directly concatenating the input image can disrupt the fundamental weights. To address this, they propose cross-frame attention mechanisms to enhance the image condition. Specifically, additional cross-frame attention layers are introduced after each self-attention layer, where keys and values are derived from the first frame, and learned queries target each output frame. This architecture is later adopted in EMO (Tian et al., 2024). VideoCrafter (Chen et al., 2023a) also use cross attentions for image conditioning, but uses an additional projection network to align the image embeddings with text embeddings.

Moreover, with the incorporation of optical flow, image conditions can be utilized more explicitly. By warping the image features using optical flow and decoding the warped image, the generated frames effectively "copy" information from specific locations in the conditioning image. This approach is employed in several flow-based video generation methods (Ni et al., 2023; Li et al., 2024k).

**Multi-level condition.** Semantic-level and fine-grained conditions are often combined to achieve more robust and precise control. For instance, both concatenation and CLIP feature cross attention can be integrated into the U-Net together to improve conditioning consistency (Xu et al., 2024b; Xing et al., 2023; Zhang et al., 2023d), VideoBooth (Jiang et al., 2024b) combines CLIP feature cross-attention with image latent cross-frame attention to provide a balance of coarse guidance and fine detail control.

VideoComposer (Wang et al., 2023e) controls the details of generated videos by employing a Spatio-Temporal Condition Encoder, which encodes various modalities, such as images, sketches, depth maps, and edge maps. Simultaneously, it leverages the CLIP embedding of an image to control the style of the generated video.

ConsistI2V (Ren et al., 2024) enhances image conditioning by integrating multiple branches that support both coarse and fine control. These include noise concatenation, image cross-attention, and temporal attention. Additionally, it introduces layout control by using the low-frequency components of the input image as noise initialization for the video generation process.

### 4.1.2 Spatial condition

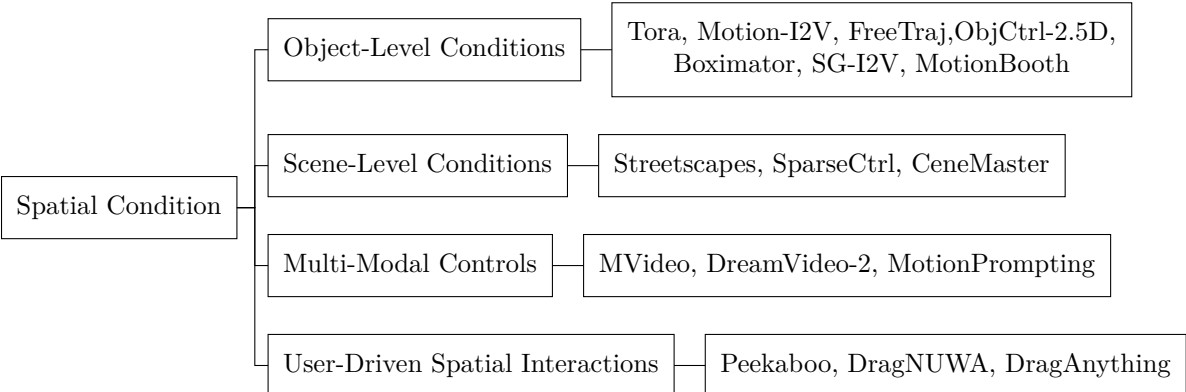

Spatial-conditioned video generation leverages explicit spatial information and constraints to guide the synthesis process, granting users greater control over the layout, movement, and dynamics of video elements. These spatial conditions may appear at multiple levels of granularity—from direct object trajectories and manipulations to structural guidance via depth maps or scene layouts—and can be combined with other modalities to ensure coherence and consistency. Ultimately, spatial conditions form the backbone for controlling motion, be it object-level movement or global camera transitions, resulting in videos that more faithfully align with user intent.

**Object-level conditions.** This category emphasizes controlling the motion and placement of individual objects, often through explicit trajectory specification. Such approaches give users fine-grained command over how objects appear and move throughout the generated scene.

Early examples include Tora (Zhang et al., 2024h), which enables users to input detailed trajectories for multiple objects, resulting in high-fidelity, multi-object videos that precisely follow the prescribed paths. Similarly, Motion-I2V (Shi et al., 2024) accepts sparse trajectory annotations and motion brushes directly on a reference image, ensuring temporal consistency and accurate object-level motion control.

Building upon these ideas, FreeTraj (Qiu et al., 2024a) supports free-form, user-defined trajectories to guide object motion, allowing flexible and intuitive spatial control. By integrating trajectory information into latent space features and employing temporal consistency modules, FreeTraj ensures seamless motion across frames while maintaining coherent scene appearances. Taking this concept further, ObjCtrl-2.5D (Wang et al., 2024i) introduces object-centric trajectory control in a 2.5D latent space. Users can define paths—including rotations, translations, and interactions—across multiple objects. With techniques like keyframe-based interpolation and motion-aware attention, ObjCtrl-2.5D guarantees spatial fidelity and smooth transitions.

Beyond trajectory drawing, object localization methods also facilitate spatial conditioning. Boximator (Wang et al., 2024d) allows users to control object positions through bounding boxes, incorporating a self-tracking mechanism to maintain motion coherence without retraining. Similarly, SG-I2V (Namekata et al., 2024) introduces a training-free approach that leverages bounding boxes and user-defined trajectories in an image-to-video diffusion pipeline. By aligning self-attention feature maps across frames and applying frequency-based post-processing, SG-I2V enforces temporal consistency, preserves high-frequency details, and enables precise zero-shot motion control without additional fine-tuning. Such bounding-box-based approaches complement trajectory-driven methods, offering additional avenues for achieving granular and visually consistent object movements. Additionally, MotionBooth (Wu et al., 2024c) fine-tunes text-to-video (T2V) models for subject-level control, introducing tailored losses (e.g., subject region and subject token cross-attention losses) that preserve object identity while enabling flexible subject movement. Its training-free "motion injection" allows users to define unique object trajectories on the fly with minimal overhead, maintaining fidelity to the customized subject's appearance even under significant motion changes.

**Scene-level conditions.** While object-level conditions dictate the movement of individual elements, scene-level constraints shape the overall layout, geometry, and camera perspective. Methods like Streetscapes (Deng et al., 2024) rely on street maps, height maps, and camera trajectories to ensure spatial coherence and

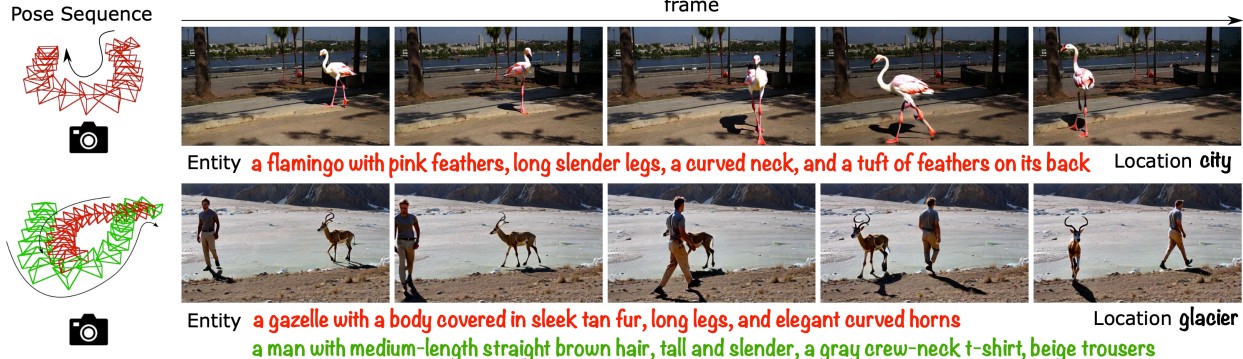

Figure 14: An example of integrating camera poses and object motion control for video generation (Fu et al., 2024a).

temporal stability in city-scale videos. SparseCtrl (Guo et al., 2024b) uses sparse sketches or depth maps at keyframes to propagate structural guidance through the sequence, reducing the need for dense inputs while maintaining spatial fidelity. CineMaster (Wang et al., 2025a) further emphasizes 3D-aware scene manipulation by integrating precise 3D bounding box placement, flexible object and camera motion, and interactive layout control into a two-stage workflow.

**Multi-modal controls.** A natural extension of spatial conditioning involves integrating other modalities, such as text, images, or optical flow, to provide more holistic guidance. MVideo (Zhou et al., 2024a) fuses semantic (textual) and spatial (image and trajectory) inputs, employing motion-aware attention mechanisms to ensure that these heterogeneous signals inform both appearance and temporal dynamics, thus allowing for richer and more contextually grounded video generation.

By combining spatial and trajectory conditions, methods can align object movements and appearances across time. DreamVideo-2 (Wei et al., 2024c) exemplifies this synergy by using a dual-stream diffusion architecture. Its Appearance Stream preserves spatial features while a Motion Stream enforces trajectory-based temporal dynamics. Optical flow maps and frame-wise image conditioning further ensure spatial consistency. Similarly, MotionPrompting (Geng et al., 2024b) maps explicit motion prompts—representing object or camera trajectories—into latent space and refines them through a two-stage generation process, achieving both stable global motion patterns and precise spatial fidelity.

**User-driven spatial interactions.** Beyond automated integration of spatial cues, several works empower users to interactively manipulate scene content. Peekaboo (Wu et al., 2020) enables direct control over object placement and movement via spatial and temporal masks, offering flexible adjustments without retraining the model. DragNUWA (Yin et al., 2023a) allows users to guide arbitrary object movements and camera shifts by combining trajectories with textual and image inputs, unlocking complex video editing through multi-modal manipulation. Extending this idea further, DragAnything (Wu et al., 2024f) leverages draggable points on objects so users can intuitively reposition and pose them across frames, enhancing interactivity and customization within video diffusion frameworks.

In essence, spatial conditions and motion controls are tightly interconnected. Specifying spatial guidance through trajectories, bounding boxes, scene layouts, or depth maps sets the stage for orchestrating how objects and viewpoints evolve over time. By blending object-level trajectories, scene-scale structures, and interactive user cues, these methods collectively push toward more controllable and user-aligned video generation.

### 4.1.3 Camera parameter condition

Camera parameter conditioning, which precisely manipulates camera poses and trajectories, is crucial for achieving realistic 3D consistency in video generation. Equally important is local object motion control—defining how specific entities move within the scene—so that the overall dynamic is both coherent

and visually compelling. Recent works interweave these two aspects, providing new ways to blend cinematographic flexibility with targeted motion editing.

Methods like CameraCtrl (He et al., 2024a) rely on camera pose and trajectory inputs to guide video generation with geometric representations and temporal attention, allowing creators to simulate dynamic viewpoints for virtual reality and film production. Extending these ideas, CVD (Kuang et al., 2024) enforces epipolar constraints to maintain geometric and motion consistency across multiple perspectives. Further exploring 3D constraints, CamCo (Xu et al., 2024b) adopts Plücker coordinates to ensure coherent epipolar geometry, while Cavia (Xu et al., 2024a) employs cross-view and cross-frame attention layers to align spatial and temporal cues in multi-view videos. VD3D (Bahmani et al., 2024d) uses scene representations like depth maps or NeRF, merging appearance and geometry streams for more realistic 3D effects, and AC3D (Bahmani et al., 2024b) integrates Plücker coordinates with hierarchical training and cross-frame attention to manage complex motions across multiple viewpoints. Augmenting these efforts, MotionBooth (Wu et al., 2024c) introduces a lightweight latent shift module for training-free camera motion control, synchronizing smooth camera movements with subject fidelity via concise loss functions and attention alignment. For more in-depth discussion on camera conditioning, please refer to Sec. 4.6.3.

On the object motion side, various approaches link bounding-box–level or trajectory-based editing to camera control for deeper storytelling. Direct-A-Video (Yang et al., 2024i) allows users to specify camera movements such as panning or zooming alongside simple bounding-box manipulation for objects, removing the need for explicit motion annotations. MotionCtrl (Wang et al., 2024j) goes further by letting users provide detailed trajectory paths for both objects and the camera, integrating these into a diffusion-based model to achieve precise movements and strong frame-to-frame consistency. Likewise, 3DTrajMaster (Fu et al., 2024a) incorporates 6DoF pose sequences for multi-entity 3D motion control, using a 3D-motion–grounded injector architecture that fuses text and pose embeddings and a gated attention mechanism within a diffusion pipeline. CineMaster (Wang et al., 2025a) further couples bounding-box–based object manipulation with camera parameter specification in a two-stage pipeline. A Blender-based interface collects 3D bounding boxes and camera trajectories, which are automatically interpolated before guiding a specialized diffusion model that disentangles camera from object motion for enhanced 3D alignment. By merging camera parameters with object trajectories, these methods deliver more comprehensive control over how scenes unfold, enabling cohesive, lifelike video generation.

### 4.1.4 Audio condition

Audio conditions enable synchronization between sound and visuals, crucial for applications like talking head generation and co-speech gesture animation. By leveraging audio inputs, models generate videos that reflect the nuances of speech, emotion, and rhythm inherent in the audio signal.

Several works have focused on generating expressive talking head videos synchronized with speech. EMO (Tian et al., 2024) conditions video generation on audio input and a single reference image, using facial bounding box masks and head movement speed as weak conditions. By integrating audio features from pretrained models like wav2vec into a diffusion-based framework, EMO synchronizes facial expressions and lip movements with the input audio, capturing emotional tones.

Similarly, VASA-1 (Xu et al., 2024e) and DreamTalk (Ma et al., 2023) utilize audio features extracted with wav2vec models to generate talking face videos. VASA-1 incorporates audio features from previous frames to maintain temporal consistency, employing classifier-free guidance for motion generation. DreamTalk uses a transformer-based audio encoder to process audio windows, producing synchronized lip movements and expressions with the help of a lip synchronization expert and a style predictor. Speech2Lip (Wu et al., 2023e) builds on these ideas by adopting a decomposition-synthesis-composition strategy, where a speech-driven implicit model focuses on lip movement in a canonical space while a geometry-aware mapping (GAMEM) handles pose variability for speech-insensitive elements like head movements. A contrastive sync loss further boosts synchronization quality, even with limited training data, yielding sharper visuals and more accurate lip alignment than previous methods. Expanding beyond facial animations, ANGIE (Liu et al., 2022b) addresses co-speech gesture generation by learning motion representations through unsupervised learning. It constructs codebooks of reusable gesture patterns and predicts future motions based on quantized motion codes and

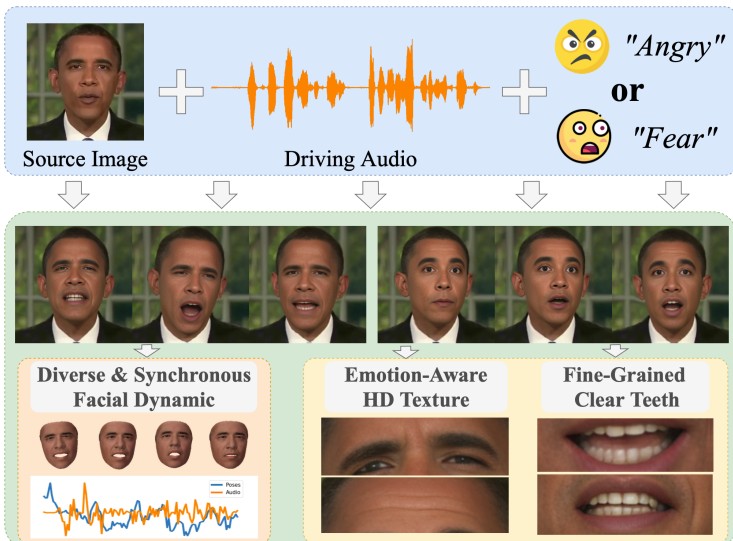

Figure 15: In audio-conditioned video generation, in conjunction with traditional conditions (e.g., image, text), audio information is usually adopted to augment synchronization with talking motion (facial expressions, head movements, lip and teeth dynamics) to create a more realistic and emotionally expressive animation (Tan et al., 2024a).

audio inputs, resulting in natural gesture animations synchronized with speech. Similarly, OmniHuman (Lin et al., 2025) extends audio-driven generation for more comprehensive human animation tasks such as full-body movements and human-object interactions, leveraging a multi-condition training strategy (text, audio, and pose) in a progressive manner, and delivers realistic motion synthesis and robust lip-sync accuracy.

For emotional expressiveness, EMOPortraits (Drobyshev et al., 2024) integrates a speech-driven mode into one-shot head avatar synthesis, using a multi-view dataset of extreme facial expressions (FEED) to generate highly expressive facial animations driven by audio inputs. FlowVQTalker (Tan et al., 2024a) combines normalizing flow and vector quantization to model and generate emotional talking face videos, predicting facial dynamics using audio inputs, a source image, and an emotion label.

To capture subtle expressions and head movements, AniTalker (Liu et al., 2024d) and AniPortrait (Wei et al., 2024a) utilize motion representations learned through self-supervised learning. AniTalker introduces a variance adapter to synthesize speech with controlled attributes, serving as input to a diffusion-based motion generator. AniPortrait employs a two-stage method, obtaining 3D representations from audio inputs and converting them into high-quality portrait animations using a diffusion model.

Additionally, CCVS (Le Moing et al., 2021) (Context-aware Controllable Video Synthesis) incorporates audio tracks as ancillary information to influence the video's rhythm and mood, maintaining temporal coherence by utilizing context frames and object trajectories.

### 4.1.5 High-level video condition

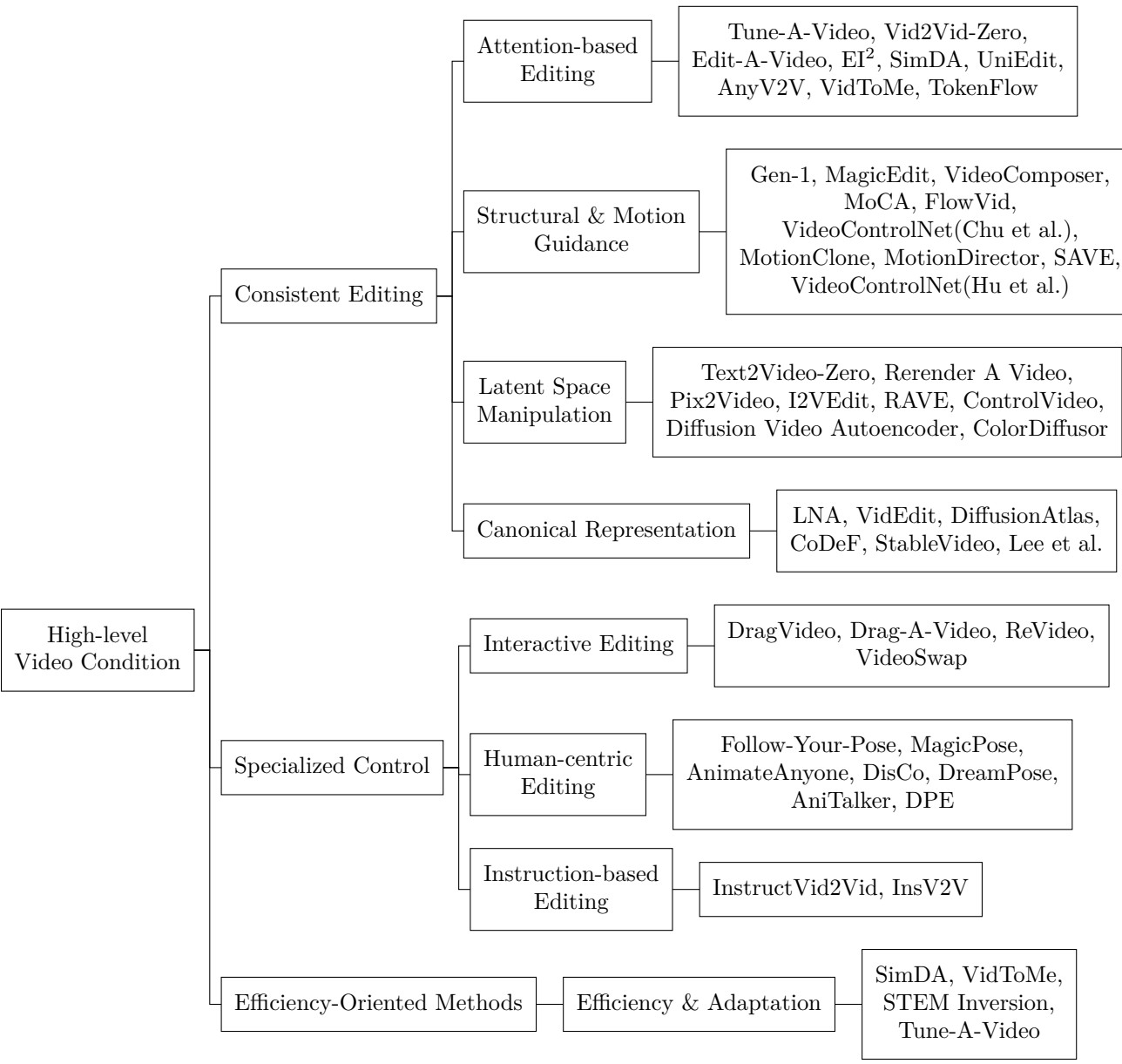

In recent years, diffusion-based video editing has become a powerful paradigm for synthesizing and manipulating videos while maintaining consistency in appearance, motion, and structure. Research in this domain has flourished across different dimensions, including strategies for preserving spatio-temporal coherence, specialized user controls, and approaches that boost efficiency. Below, we group these methods under three broad categories and detail the core ideas within each.

**Consistent editing.** A large body of work addresses how to preserve uniformity across frames, focusing on refined attention mechanisms, structural and motion guidance, latent space manipulation, and canonical representations. In terms of *attention-based approaches*, some of the earliest breakthroughs emerged when text-to-image diffusion models were extended to video. Tune-A-Video(Wu et al., 2023b) introduced sparse-causal spatio-temporal attention to adapt Stable Diffusion for one-shot video editing, while Vid2Vid-Zero(Wang et al., 2023d) developed a dense spatial-temporal attention module and cross-frame attention to capture longer-range temporal dependencies. Edit-A-Video(Shin et al., 2024) likewise addressed temporal flicker by injecting attention maps from previous frames and applying a novel temporal-consistent blending

method. Further enhancements came from methods like $EI^2$(Zhang et al., 2024k) and SimDA(Xing et al., 2024b), which minimized feature shifts via Shift-restricted Temporal Attention Modules, instance centering, spectral normalization, and Latent-Shift Attention to achieve smoother updates with fewer parameters. Some techniques also explicitly combine spatial and temporal features: UniEdit(Bai et al., 2024) and AnyV2V(Ku et al., 2024) fuse self-attention features across frames or integrate DDIM inversions to preserve both content and motion. Token-level operations further enhance frame alignment, with VidToMe(Li et al., 2024h) merging redundant tokens locally and globally, and TokenFlow(Geyer et al., 2023) identifying nearest-neighbor tokens to propagate consistent structure.

Beyond direct attention, some methods leverage explicit structural or motion signals to guide the diffusion process. Gen-1(Esser et al., 2023) demonstrated that providing depth maps alongside noisy latents ensures coherent scene geometry, while MagicEdit(Liew et al., 2023) and VideoComposer(Jiang et al., 2023a) used ControlNet or spatio-temporal encoders for depth, pose, or sketch guidance. Optical flow is also popular for ensuring consistent movement: MoCA(Yan et al., 2023a) projects edits from the initial frame onto subsequent frames through flow warping, FlowVid(Liang et al., 2024a) conditions the diffusion on flow-aligned features, and VideoControlNet(Chu et al., 2023) uses similar flow-based constraints to reduce flicker. Motion-Clone(Ling et al., 2024c) instead relies on internal temporal attention signals to extract dominant motion cues, making it training-free. Splitting appearance and motion into separate pathways further enhances flexibility, as in MotionDirector(Zhao et al., 2023c) (with dual-path LoRA) and SAVE(Song et al., 2023) (with a motion word and motion-aware cross-attention loss). A specialized example is VideoControlNet (Hu et al.)(Hu & Xu, 2023), which segments the video into I-, P-, and B-frames for independent generation, motion-guided updates, and interpolation, respectively.

Latent space manipulation is another powerful direction for preserving consistency. Text2Video-Zero(Khachatryan et al., 2023) and Rerender A Video(Yang et al., 2023b) warp the latent codes of initial or subsequent frames to guide motion while keeping appearance stable. Pix2Video(Ceylan et al., 2023) and I2VEdit(Ouyang et al., 2024b) inject self-attention features from prior frames to ensure visual continuity. Other methods, such as RAVE(Kara et al., 2024) (noise shuffling and the grid trick) or ControlVideo(Zhang et al., 2023e) (an interleaved-frame smoothing pipeline), unify frame latents during denoising. Diffusion Video Autoencoder(Kim et al., 2023a) explicitly disentangles identity, motion, and background within latent representations to simplify partial edits, and ColorDiffusor(Liu et al., 2023a) addresses color drift with dedicated Color Propagation Attention and an Alternated Sampling Strategy.

Finally, canonical representations use a consistent 2D atlas or parameterization to propagate edits across frames. Methods like Layered Neural Atlases(Kasten et al., 2021), VidEdit(Couairon et al., 2023), and DiffusionAtlas(Chang et al., 2023b) map videos into an atlas space, apply edits there, and then map them back to the original frames. CoDeF(Ouyang et al., 2024a) extends this concept using a hash-based design for canonical and deformation fields, while StableVideo(Chai et al., 2023) and Lee et al.(Lee et al., 2023) propagate keyframe edits via atlas-based correspondences, with the latter estimating dense semantic links to handle shape changes.

**Specialized control.** Several methods emphasize more direct or domain-specific control, whether through user interaction, human-centric capabilities, or language-based instructions. Interactive editing systems like DragVideo(Deng et al., 2023) and Drag-A-Video(Teng et al., 2023) adapt point-based image manipulation to the video domain by tracking user-selected points across time and optimizing latent codes for smooth motion. ReVideo(Mou et al., 2024) lets users edit the first frame and then specify new motion trajectories, supported by a three-stage training strategy that decouples content and motion. VideoSwap(Gu et al., 2024b) targets subject swapping via semantic point correspondences and layered atlases, maintaining consistent appearances in dynamic scenes.

Human-centric editing often centers on preserving identity, pose, and expressions across multiple frames. Approaches like Follow-Your-Pose(Ma et al., 2024d) and MagicPose(Chang et al., 2023a) typically train for appearance control first and then refine temporal or pose-specific features using modules like ControlNet. Specialized appearance encoders, as in AnimateAnyone(Hu, 2024) or MotionEditor(Tu et al., 2024), capture high-fidelity details that are integrated into the denoising process for better motion generation. Broader subject or background manipulations appear in DisCo(Wang et al., 2024e), DreamPose(Karras et al., 2023),

and AniTalker(Liu et al., 2024d), leveraging ControlNet-based architectures to separate or recombine different video elements. DPE(Pang et al., 2023) focuses specifically on decoupling pose and expression through a bidirectional cyclic training scheme, skipping 3D Morphable Models for simpler pipelines.

Instruction-based editing harnesses natural language to guide transformations in video, making them accessible to non-experts. InstructVid2Vid(Qin et al., 2024) and InsV2V(Cheng et al., 2023) adapt Prompt-to-Prompt editing for multi-frame scenarios by generating training data from captioning systems. They preserve coherence via specialized objectives like an Inter-Frames Consistency Loss and strategies such as Long Video Sampling Correction.

**Efficiency-oriented methods.** Since diffusion-based pipelines can be computationally intensive, numerous works focus on streamlining editing without degrading consistency. SimDA(Xing et al., 2024b) employs Latent-Shift Attention and lightweight adapters, while VidToMe(Li et al., 2024h) uses Token Merging (ToMe) to compress attention tokens both locally and globally, saving compute cycles across frames. STEM Inversion(Li et al., 2024g) applies an EM algorithm to find compact bases for each frame, reducing the dimensionality of video representations and speeding up inference. On a similar note, Tune-A-Video(Wu et al., 2023b) demonstrates that carefully designed sparse spatio-temporal attention modules can maintain temporal stability while significantly cutting resource usage.

### 4.1.6 Other conditions

Recent advancements in video generation have incorporated various special conditions to enhance control, realism, and applicability. These conditions include audio inputs, physical simulations, trajectories, scene layouts, and physiological signals. Below, we survey these developments, categorizing the works based on the special conditions they employ.

**Physics-based video generation.** Incorporating physics-based conditions ensures that generated motions adhere to real-world dynamics. PhysGen (Liu et al., 2024c) conditions video generation on external forces and physical parameters like gravity and friction to simulate realistic object motion. Starting from a single image, it extracts physical properties, simulates dynamics using a physics engine, and renders the motion using diffusion models, allowing for physically grounded and controllable video sequences.

Similarly, MotionCraft (Aira et al., 2024) uses physics-based optical flow derived from simulations to guide video generation. By warping the latent space of a pretrained diffusion model according to the optical flow, it animates the starting frame with the desired motion, enabling the synthesis of complex movements without retraining the model.

Extending physics-based conditioning to interactive scenarios, VideoAgent (Soni et al., 2024a) introduces multi-agent interactions guided by instructional text prompts and environmental contexts. By simulating agent behaviors that adhere to physical laws, it generates purposeful videos involving multiple interacting agents.

**Physiological signal conditioning.** Incorporating physiological signals opens avenues for health-related applications Synthetic Generation of Face Videos with Plethysmograph Physiology (Wang et al., 2022c) uses a given PPG signal to guide video generation, reflecting physiological signals like heart rate. Starting from a single image, it introduces variations to enhance the training of remote photoplethysmography models.

**Timestamp conditioning.** Timestamp conditions pinpoint precisely *when* events occur, ensuring proper sequencing and duration. MinT (Wu et al., 2024h) exemplifies this by binding each event to a designated time range, using Rescaled Rotary Positional Encoding for coherence, and leveraging scene cut conditioning for seamless transitions. With an LLM-based prompt enhancer to expand event captions, MinT maintains smooth multi-event progression, improves text alignment, and preserves subject integrity, highlighting the importance of time-based control for narrative-rich video generation.

### 4.2 Enhancement

The enhancement of image and video qualities has been a long-lasting and well-studied topic in computer vision. Early literature focused on low-level (i.e., pixel-level) video enhancement, where we process videos

based on low-level photometric or geometric features to resolve specific issues such as video noise, motion blur, *etc.* However, in recent years, the success of generative models including GANs and diffusion models has offered new options to tackle these traditional low-level tasks. In this chapter, we review and discuss the potential of these new generative models on traditional video enhancement tasks including video denoising, video inpainting, video super-resolution, video interpolation and prediction.

### 4.2.1 Video denoising and deblurring

Video denoising aims to enhance video quality by reducing noise and restoring degradation issues like out-of-focus blur, motion blur, and compression artifacts such as moiré patterns (Rota et al., 2023). The noise or degradation defined in these tasks is low-level meaning that it generally does not require much high-level (semantic/object-level) understanding of the video.

Interestingly, the latest diffusion models are inspired and structured with a denoising framework, so they seem to be a direct fit for video denoising. However, they have completely different focuses. The latest diffusion models are mostly designed for highly creative generation tasks, where we use the model to generate from an image of complete noise. The whole process is more about "imagining" what might be hidden beneath the noise, rather than simply "removing" the noise. With large datasets, diffusion models learn the overall distribution of images rather than focusing on refining individual local patches based on appearance. In contrast, video denoising assumes that the input videos, although noisy, should start with enough quality, where at least the main content should be be recognizable. The goal is to reduce noise while keeping the main content unchanged in the video, leaving little room for creativity compared to generative diffusion models.

More broadly, there have been few successful generative models, not just diffusion models, specifically developed for video denoising tasks. The reasons are three-fold.

- **Different focus**: Video denoising requires the model to strictly maintain the original content of the video and avoid introducing extra new information. The task itself limits creativity, so generative modesls generally do not fit very well in this area.

- **Existence of good traditional methods**. Since the task primarily only involves minor local refinement, video denoising is generally not a very challenging task. Also, due to its long history in research, many effective traditional methods already exist and can address the problem reliably. Using generative models for this relatively simple task may seem overkilling.

- **Efficiency**. Compared to traditional methods, generative models are much more complex, making them harder to train and less efficient in practical applications.

### 4.2.2 Video inpainting

Video inpainting is a process used to fill in the missing or corrupted parts of a video Quan et al. (2024). A masked area is usually specified as a part of the input, and the task is aimed at filling in that area using spatial and temporal cues to make the video appear seamless and coherent. Note that the masked area can be specified in various ways depending on the application scenario. Here are some examples.

- In applications where the goal is to restore a corrupted area in the video, the masked area is defined as the region of corruption, and the task mainly focuses on filling in the area based on the local continuity of the frame.

- In applications where the goal is to remove unwanted objects, the mask area is defined as the object masks, and the task focuses more on finding temporal correspondences to fill in the background behind the object.

While traditional methods typically adhere to a predefined problem setting with a specified masked area, recent diffusion methods aim to be more creative by incorporating language instructions and addressing

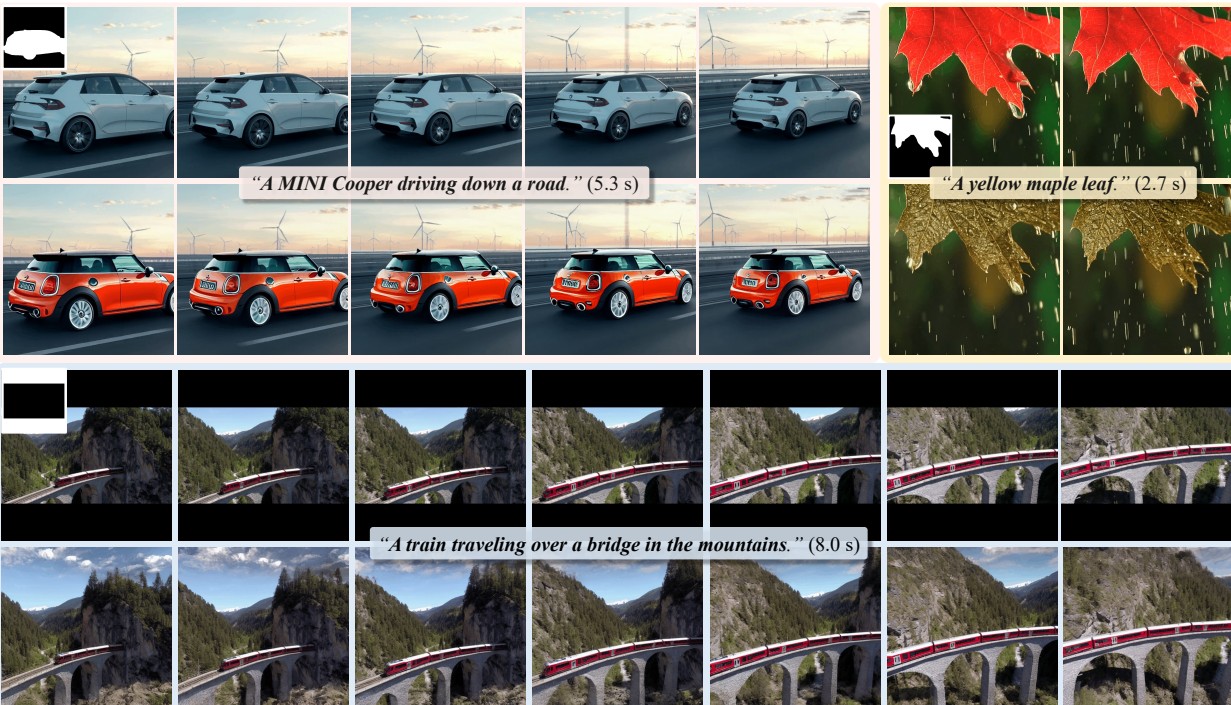

Figure 16: Example of video inpainting with text instructions for video generation. The contents in the masked regions are inpainted under text instrubtions. Original figure from AVID (Zhang et al., 2024i)

more challenging cases through complex generation. One such example is shown in Figure 16, where masks and instructions are provided to enable video editing applications as a video inpainting task (Zhang et al., 2024i).

GAN (Generative Adversarial Network) models were first adopted to solve the inpainting problem from a generative perspective. Starting from image inpainting, Pathak et al. (2016) uses inpainting as a self-supervised task to learn video representations, where an adversarial loss is introduced to produce sharper results. Yu et al. (2018) also introduced local and global WGANs for enhanced results. Recently, Chang et al. (2019) extended this idea to free-form video inpainting, where the masks can have arbitrary shapes such as text captions, and proposed a temporal PatchGAN loss to enhance temporal consistency.

Diffusion moodels have also been explored for inpainting tasks, and such explorations mainly start from image inpainting. RePaint (Lugmayr et al., 2022) applied a pretrained unconditional DDPM and modified the standard denoising strategy by sampling the masked regions from the diffusion model and sampling the unmasked areas from the given image. SmartBrush (Xie et al., 2023) explored image inpaiting with both text and shape guidance with precision control. They try to generate a new object to be filled in the shape specified by the masks. To preserve the background and improve consistency, they add an extra mask prediction module in the diffusion process, which is used to guide the sampling process in the inference stage. For video inpainting, Green et al. (2024) reframe the task as a conditional generative modeling problem tackled with conditional video diffusion models. They use sampling schemes specific to inpainting problems to capture long-range dependencies and show that their model can inpaint diverse novel objects that are semantically consistent with the context.

To better utilize the multi-modal capability of diffusion models, some research work has focused on adding inputs of other modalities, such as text instructions, into video inpainting, paving the way for real applications like text-guided video editing. AVID (Zhang et al., 2024i) builds on existing text-guided image-inpainting models and improves temporal consistency by integrating motion modules. They also proposed a structure guidance module tailored for different types of masks, and a middle-frame attention guidance method for

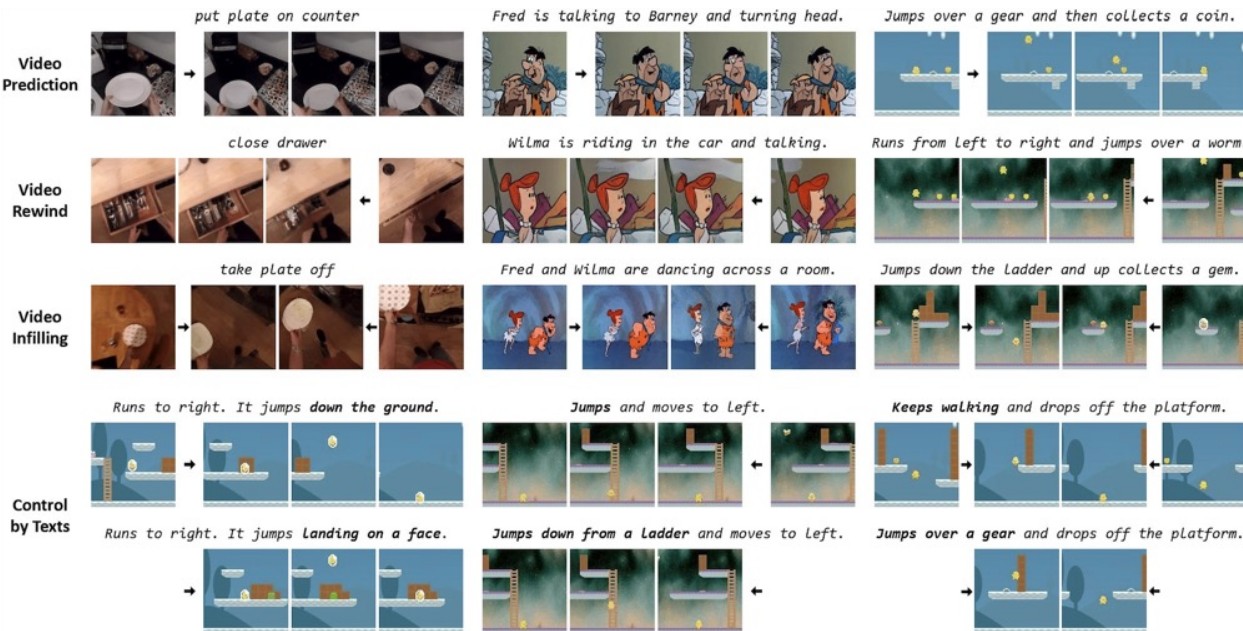

Figure 17: Examples of video extrapolation/prediction (Row 1&2) and interpolation (Row 3). For diffusion models, it is possible to add text instructions for a better control. Original figure from Fu et al. (2023)

longer videos. Subsequently, CoCoCo (Zi et al., 2024) proposes a more efficient motion capture module and an instance-aware region selection method for better "consistency, controllability, and compatibility" (and hence the name "CoCoCo").

Furthermore, UniPaint (Wan et al., 2024) presents an interesting argument that video inpainting and interpolation can be unified and mutually enhanced as both tasks involve filling missing information in the spatial and temproal domains, respectively, for which they propose a spatial-temporal masking strategy during training. Wu et al. (2024b) introduces a pure language-driven video inpainting task, where the specified inpainting masks are no longer needed. Instead, they use natural language instructions, such as instructions to remove objects, to guide and condition the inpainting process.

### 4.2.3  Video interpolation and extrapolation/prediction

Video interpolation aims to generate intermediate frames between two consecutive frames to increase the video's frame rate. In contrast, video extrapolation or prediction focuses on forecasting the next frame following the input frames. Both tasks involve generating new frames along the temporal dimension and thus depend heavily on understanding the motion between frames.

Traditionally, video interpolation and extrapolation/prediction are distinct tasks approached with different methodologies. Video interpolation focuses on finding or synthesizing intermediate frames to reflect the motion between existing frames, while video extrapolation requires more creativity and may rely on external domain knowledge or common sense to help predict future frames. Moreover, video interpolation has clear and direct real-life applications and generally produces more reliable results because it uses temporal cues from both preceding and succeeding frames. As a result, video interpolation has traditionally been of greater interest compared to video extrapolation/prediction.

However, from the perspective of diffusion models, video interpolation and extrapolation can be tackled in a more unified framework. One example is illustrated in Figure 17. Therefore, we discuss both tasks together in this chapter.

With the rise of large pretrained diffusion models, video interpolation and extrapolation are increasingly viewed as conditional generation tasks, rather than relying heavily on motion analysis and image warping.

In both cases, generative methods follow the same approach—generating new frames based on the surrounding context frames. This allows a unified diffusion model to be trained to handle both interpolation and extrapolation.

One early example is MCVD (Masked Conditional Video Diffusion) (Voleti et al., 2022), where a diffusion model is trained to generate current frames conditioned on past and future frames. By randomly masking these frames, the model adapts to different tasks. If both past and future frames are masked, the model performs unconditional video frame generation. If only either past or future frames are masked, it handles past frame or future frame prediction, respectively. When neither are masked, the model performs video interpolation. This random masking enables MCVD to learn video interpolation, prediction, and generation in a unified framework. Similar ideas also appeared in RaMViD (Random-Mask Video Diffusion) (Höppe et al., 2022), but instead of masking in the temporal dimension, RaMViD apply masks in the spatial domain and use a unified framework to tackle video interpolation, infilling, and upsampling.

Moreover, some later methods also specifically explored the power of diffusion models on video interpolation. VIDIM uses cascaded diffusion models, where low-resolution target frames are first generated before generating high-resolution versions conditioned on these generated low-resolution frames(Jain et al., 2024a). LDMVFI explored latent diffusion models to achieve better perceptual generation quality for video interpolation (Danier et al., 2024).

Some other recent work has also adapted video interpolation/extrapolation to specific application scenarios. Fu et al. (2023) proposes MMVG (Multi-modal Masked Video Generation) to tackle a new adapted task called text-guided video completion (TVC) motivated by the uncertainty of frame prediction. Specifically, a video can have multiple potential future outcomes, so it is better to add text guidance to describe a specific future that we want to generate. Apart from that, some work explored interpolation for specific styles such as cartoon (Xing et al., 2024a).

Overall, we have seen great differences between traditional and generative methods.

- **Creativity**: Traditional methods adhere to the original content, motion, and physics in the video, while generative methods can generate in a more creative way.

- **Dependence on motion**: Traditional methods mostly rely on strong motion assumptions such as motion smoothness and linearity, so they may fail if the underlying motion is complex, nonlinear, or ambiguous (Jain et al., 2024a). Also, these smooth motion assumptions often result in overly smoothed predictions by traditional methods. However, diffusion models are less vulnerable to these problems due to their low dependence on motion understanding.

- **Difficulty**: Traditional methods are typically designed for simple settings, where only one intermediate or future frame is predicted from context frames, with short time intervals between them to maintain smooth motion assumptions. In contrast, generative methods can handle more complex tasks, such as interpolating or predicting 5-10 frames at a time, and the context frames can be far apart in time. An extreme example would be having only the first and last frames of a 5-second video. In this case, generative models are well-suited to generate the entire video since most of the information is completely made up. Traditional methods, however, would struggle to handle this scenario due to lack of context. We will discuss more about long video generation in Section 4.5.

### 4.2.4 Video super-resolution

Video super-resolution is another well-established task in video restoration, focusing on enhancing low-resolution videos by upsampling the video frames to produce clearer and sharper high-resolution outputs. Early research focused more on image super-resolution algorithms, followed by video super-resolution methods, which introduced temporal consistency as an additional optimization objective. Some visual examples are shown in Figure 18.

In recent years, diffusion models have also been applied to super-resolution tasks. StableSR (Wang et al., 2024c) explored leveraging the prior knowledge from pre-trained diffusion models to improve super-resolution quality. They employed a frozen Stable Diffusion model to generate high-resolution images conditioned on the

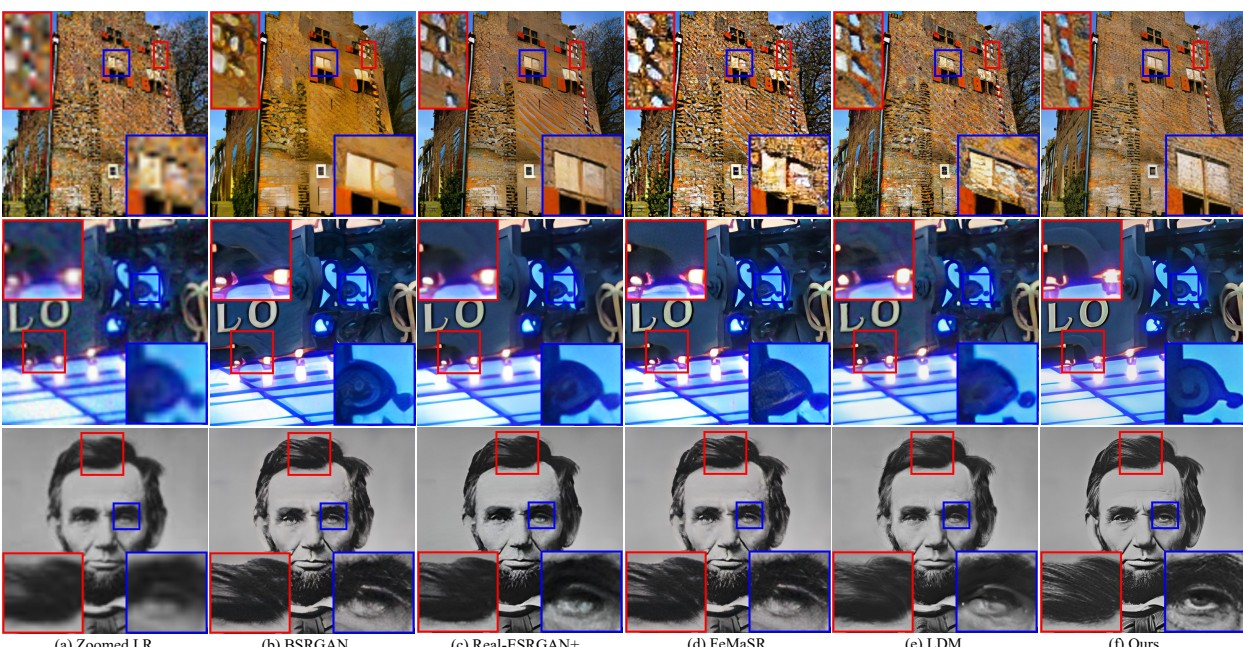

| (a) Zoomed LR | (b) BSRGAN | (c) Real-ESRGAN+ | (d) FeMaSR | (e) LDM | (f) Ours |

Figure 18: Video super-resolution examples of latest methods. Original figure from StableSR (Wang et al., 2024c)

low-resolution features computed from a pretrained super-resolution model. A trainable time-aware encoder module was added to combine the super-resolution features with the diffusion features. Similarly, Yuan et al. (2024b) proposed to utilize a frozen pre-trained text-to-image diffusion model as the base to accomplish text-to-video super-resolution. They first use the image diffusion model to generate high-resolution images one-by-one and then refine the results through a temporal adapter.

Some latest methods have also investigated different ways to combine super-resolution networks (mostaly a VAE) and diffusion networks (U-Net). SATeCo (Chen et al., 2024f) combined the super-resolution (VAE) and diffusion model (U-Net) by inserting the U-Net structure in between the VAE encoder and decoder, together with some adapation or alignment modules. The latest VEnhancer (He et al., 2024b) also shares a similar structure with additional space-time data augmentation and video-aware conditioning, which are trained to enhance AI-generated video on both space and time resolutions. In comparison, Upscale-A-Video (Zhou et al., 2024b) is more designed for long videos. They first utilize the U-Net to generate each video segment given optical flow inputs and perform a global recurrent latent propagation to ensure consistency across a longer time range. The VAE decoder is then applied to decode high-resolution images. Both structures have shown pros and cons on flexibility, fidelity and quality.

In addition to using diffusion models to enhance video resolutions, some research work has also studies how to enhance/adapt low-resolution image/video generation models so that it can also work well on high-resolution tasks. Guo et al. (2025a) propose a tuning-free method through a pivot replacement strategy that leverages reliable semantic guidance from the low-resolution model, as well as a tuning method that integrate learnable multi-scale upsampler modules to learn structural details at the new higher resolution. To follow up, FreeScale (Qiu et al., 2024c) introduces a tuning-free inference paradigm that fuses features from different perceptive scales and extracts the desired frequency for each part of the frame.

### 4.2.5 Combining multiple video enhancement tasks

As we have discussed in the previous chapters, many diffusion methods have unified and tackled multiple enhancement tasks at the same time. To recall a few, UniPaint Wan et al. (2024) combines video inpainting and video interpolation, and MCVD (Voleti et al., 2022) unifies video interpolation and video prediction.

Notably, VEnhancer (He et al., 2024b) is a plug-and-play method used to enhance a low-quality video generated by other generation models by removing artifacts and increasing both of its spatial and temporal resolutions.

This trend mainly stems from the flexibility of diffusion models which can be trained to generate outputs in a task-agnostic manner. As a result, the differences among enhancement models are more based on different training data and strategy instead of model structures.

### 4.2.6 Summary

In summary, recent diffusion-based methods have significantly advanced traditional video enhancement techniques, improving tasks such as video denoising, inpainting, interpolation, extrapolation/prediction, and super-resolution. From this review, we can see a clear trend as follows.

- Diffusion-based methods introduce greater creativity into traditional low-level video enhancement tasks. Leveraging the strength of pretrained diffusion models, these approaches can tackle more complex and challenging scenarios where traditional methods often fall short.

- Diffusion models have also enabled the use of multi-modal inputs, allowing enhancement tasks to be performed with additional inputs like text instructions. This advancement is revolutionary, as text-based input provides an effective means of making the enhancement process more controllable and customizable, opening new possibilities for video editing.

- From the perspective of diffusion models, the distinction between image and video is becoming less significant. Traditional methods often develop image and video enhancement algorithms separately, as video enhancement requires careful attention to temporal correspondences and alignment (the "physics" of motion). In contrast, diffusion models, which are less dependent on explicit motion understanding, allow for a more seamless and scalable adaptation between image and video models.

- Nevertheless, diffusion models can introduce more fidelity issues due to the inherent randomness of the diffusion process. In video enhancement, the output is expected to preserve the original content or subject matter, but diffusion models may introduce new elements that were not originally present. Finding ways to balance fidelity and quality can be an interesting topic for future research.

### 4.3 Personalization

Personalization in video generation is essential for creating content that accurately represents specific subjects, capturing their unique identities, expressions, and motions. Two primary approaches enable this personalization: subject-specific fine-tuning and subject-specific-driven inference. While fine-tuning approaches carefully adapt model parameters to a particular subject, inference-based methods rely on minimal references—often a single image—to guide identity preservation and motion synthesis without retraining.

**Subject-specific fine-tuning.** Subject-specific fine-tuning involves adapting a pre-trained model to a particular subject by retraining or modifying its parameters using data specific to that subject. This approach allows the model to embed detailed subject characteristics directly into its learned representation.

DreamVideo (Wei et al., 2024c) fine-tunes an identity adapter on a few static images of the subject, capturing subject-specific identity traits, while separately training a motion adapter on target motion patterns. This decoupling allows the system to integrate both identity and motion seamlessly into the final output.

Similarly, PersonalVideo (Li et al., 2024c) adopts a fine-tuning strategy to preserve high identity fidelity without degrading the model's motion and semantic capacities. By integrating an isolated identity adapter into the spatial self-attention layers of a pre-trained text-to-video diffusion model, PersonalVideo fine-tunes identity features without disrupting motion dynamics. This method enhances robustness through simulated prompt augmentation, ensuring that even with limited reference images, the model achieves reliable identity preservation alongside customizable style and motion.

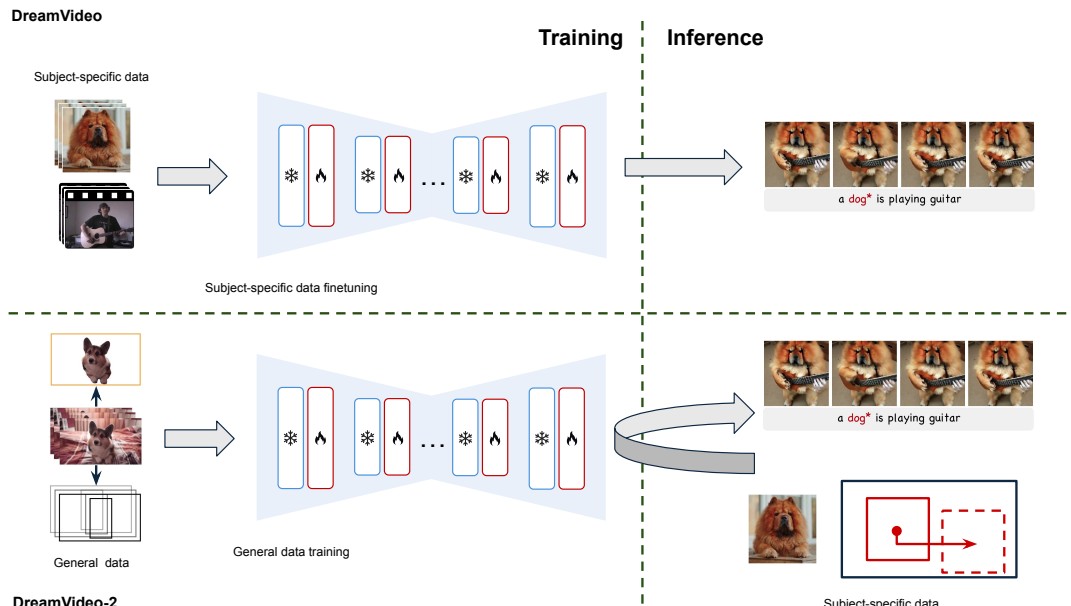

Figure 19: A high-level comparison of two personalization paradigms—subject-specific finetuning and subject-specific-driven inference—illustrated by DreamVideo (Wei et al., 2024c) and DreamVideo-2 (Wei et al., 2024b). In DreamVideo, each new subject (its images and desired motion) is used to finetune the model. In contrast, DreamVideo-2 leverages a pretrained diffusion model trained on general videos (decomposed into subjects and bounding boxes); it then generates videos for a new subject based on a single image and bounding boxes—without requiring additional finetuning.

By extending these techniques to scenarios without specialized video data, Still-Moving (Chefer et al., 2024) leverages text-to-image (T2I) customization and introduces lightweight spatial and temporal adapters for text-to-video pipelines. By training on duplicated still images (or "frozen videos"), it retains motion priors while aligning T2I and T2V features, enabling diverse personalization with only a handful of static images. In parallel, Dynamic Concepts (Abdal et al., 2025) addresses more inherently dynamic concepts—subjects whose characteristic motion is as integral to their identity as appearance—by fine-tuning a Diffusion Transformer in two stages. First, it learns an appearance "basis" from an unordered set of frames, and then refines temporal dynamics by freezing the learned parameters and focusing on motion consistency. Finally, Video Alchemist (Chen et al., 2025a) expands personalization to multi-subject, open-set scenarios by fine-tuning a Diffusion Transformer, and personalizes both foreground objects and backgrounds while mitigating overfitting through targeted data augmentation.

**Subject-specific-driven inference.** Subject-specific-driven inference personalizes video outputs at run-time without retraining the model for each new subject. These methods typically rely on a single reference image (and sometimes auxiliary inputs like audio or text prompts) to guide identity preservation and expression synthesis.

Many audio-driven talking head frameworks fall into this category. EMO (Tian et al., 2024), VASA-1 (Xu et al., 2024e), AniTalker (Liu et al., 2024d), and AniPortrait (Wei et al., 2024a) generate expressive talking head videos from a single reference image and an audio input. They rely on diffusion-based models and carefully designed motion encoding strategies to preserve subject identity while animating expressions and head movements. Approaches like DreamTalk (Ma et al., 2023), EMOPortraits (Drobyshev et al., 2024), and FlowVQTalker (Tan et al., 2024a) extend this concept by incorporating emotional expressiveness, predicting personalized emotion intensity from audio without additional style references, and ensuring identity consistency even under intense expressive variations. Beyond facial animation, ANGIE (Liu et al., 2022b) expands this paradigm to co-speech gesture generation, personalizing gesture patterns from minimal inputs while retaining the speaker's characteristic style.

Recent works further refine and expand subject-specific-driven inference into broader video customization scenarios. For instance, DreamVideo-2 (Wei et al., 2024b) introduces zero-shot subject-driven video generation with precise motion control. Using only a single reference image and bounding box sequences to define the subject's trajectory, it employs reference attention mechanisms and mask-guided motion modules to integrate subject appearance with flexible, fine-grained motion paths—achieving personalization without additional fine-tuning steps.

Building on this idea of inference-time personalization, ConsisID (Yuan et al., 2024a) demonstrates how decomposing identity features into low-frequency (global facial contours) and high-frequency (fine details) components can ensure identity preservation in text-driven synthesis. By integrating these components into a pre-trained Diffusion Transformer, ConsisID consistently captures the subject's essential features across varying poses and expressions, all without retraining for each subject.

Lastly, MCNet (Hong & Xu, 2023) addresses motion transfer scenarios, animating a still image to follow the dynamics of a driving video while preserving identity through an implicit identity representation conditioned memory module. By extracting and reusing structural and appearance details from a single reference image, MCNet maintains subject fidelity throughout complex motions—again, without retraining on the target subject. Together, these approaches illustrate the growing versatility of inference-based frameworks that balance flexible control, identity preservation, and visual fidelity without the burden of subject-specific fine-tuning.

## 4.4 Consistency

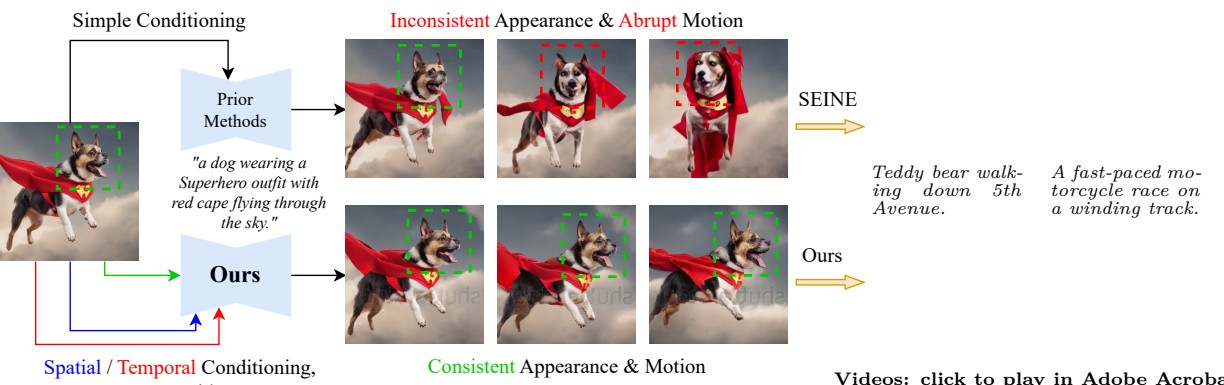

Figure 20: Comparison of image-to-video generation results obtained from SEINE and ConsistI2V. SEINE shows degenerated appearance and motion as the video progresses, while ConsistI2V maintains visual consistency utilizing spatial and temporal conditions.

Improving the consistency of generated videos has been a hot topic recently. Researchers have been working on improving consistency in two different perspectives, *i.e.*, semantic consistency, and motion consistency. Early attempts (Yu et al., 2021; Skorokhodov et al., 2022; Shen et al., 2023; Zhang et al., 2023c) for GAN-based video generation methods mainly add additional discriminators or learned representations for controlling consistency. For example, DI-GAN (Yu et al., 2021) introduces an implicit neural representations-based video generator that improves motion dynamics by manipulating space and time coordinates along with a motion discriminator for temporal consistency. StyleGAN-V (Skorokhodov et al., 2022) employs continuous motion representations as positional embeddings.

**Diffusion models**. Semantic consistency (Ren et al., 2024; Zhang et al., 2024j; Jiang et al., 2024b; Li et al., 2024h; Zhou et al., 2024d; Sun et al., 2023b) ensures the semantic meaning including objects and relationships between objects are smooth between different frames. The biggest problem is semantic drifting, which refers to the gradual shift or change in the meaning or context of visual content as a video progresses. StableVideo (Chai et al., 2023), as a pioneer work, proposes to propagate the representation to the next frame to ensure consistency. Similarly, TokenFlow (Geyer et al., 2023) preserves consistency by explicitly

propagating diffusion features based on inter-frame correspondences using sampled tokens for video editing. GLOBER (Sun et al., 2023a) employs a novel adversarial loss along with conditioning the generation on the global features extracted by a video encoder for global consistency. Specifically, for face video generation, EMO (Tian et al., 2024) preserves visual/identity consistency by encoding a reference image/video/audio and cross-attention for video generation.

Motion consistency consists of multiple aspects, such as video smoothness, reasonable object motion, and stable camera movement, while two different frameworks, including residual-based framework (Deng et al., 2024; Zhou et al., 2024d) and optical flow-based framework, are proposed to ensure the motion consistency of generated videos. For the residual-based methods, they usually add additional motion representation extracted from previous frames or text prompts. As a representative work, Cinemo (Ma et al., 2024b) learns to generate residual images between a previous frame and a later frame for motion control. Moreover, it also incorporates a new noise refinement with discrete cosine transformation for controlling the input noise. VideoComposer (Wang et al., 2023f) uses motion vectors, depth sequences, mask sequences, and sketch sequences to assist in the generation of video frames. On the other side, in addition to the image-based residual, StoryDiffusion (Zhou et al., 2024d) proposes to split the long text prompts into multiple short text prompts to improve consistency.

For the optical flow-based models, they usually learn to generate optical flow between a previous frame and a later frame to maintain consistency. LFDM (Ni et al., 2023) introduces a flow predictor during training and injects flows into the denoising procedure for motion consistency. Similarly, Motion-I2V (Shi et al., 2024) consists of two stages. It first generates optical flows based on the input image and then videos are generated conditioned on the concatenation of these optical flows. For video editing, CAMEL (Zhang et al., 2024b) improves consistency through a frequency-based perspective, where the video is decomposed using low- and high-pass filters. Moreover, it learns additional motion prompts for further stabilize the video.

Apart from these methods, another popular paradigm for consistency is controlling the noise used in diffusion models. PYoCo (Ge et al., 2023) is first proposed to employ a novel correlated noise model, which consists of a noise shared among all frames and an individual noise conditioned on the previous frames. Following this, a $\int$-noise (Chang et al., 2024a) is proposed to warp noise following the flow and preserve the spatial Gaussian property, which is able to preserve the motion and semantic consistency at the same time.

One application of consistency is long video generation, which poses high requirements for motion and semantic consistency. long video generation (Ouyang et al., 2024c) is always achieved by ensuring a smooth transition and semantic consistency among different frames. StoryDiffusion (Zhou et al., 2024d) proposes to use a novel consistent self-attention and a semantic space temporal prediction for object consistency. Similarly, StreamingT2V (Henschel et al., 2024) employs a long-term memory block, a short-term memory block, and also a new blending approach for infinitely long video generation. The short-term block focuses on consistency between adjacent frames, while the long-term block forces the model to depend on the first several frames to avoid forgetting. Later, SEINE (Chen et al., 2024e) proposes to employ a random-mask model for generating transitions based on textual descriptions. Multiple randomly masked images can be seen as images from different scenes and they can provide an overall comprehensive angle to consistency and long video generation. On the other side, 3D scene mesh (Fridman et al., 2023) is also used for generating videos, which can be seen as another application of consistency. Specifically, the video frames and meshes are updated through several pre-trained models and the consistency between different frames is preserved through the iterated refinement of the mesh.

### 4.5 Long video

Long video understanding, including retrieval, classification, and generation, has been a challenging problem for a long time. Videos are always of different lengths and might be infinitely long, such that models find it hard to keep content consistent among different frames. In this part, we will introduce representative methods in long video generation.

Almost all the **traditional long video generation methods** are divide-and-conquer-based methods. They usually first generate keyframes and then generate other frames conditioned on these generated keyframes. TATS (Ge et al., 2022) uses multiple models (interpolation and autoregressive transformers) to ensure the

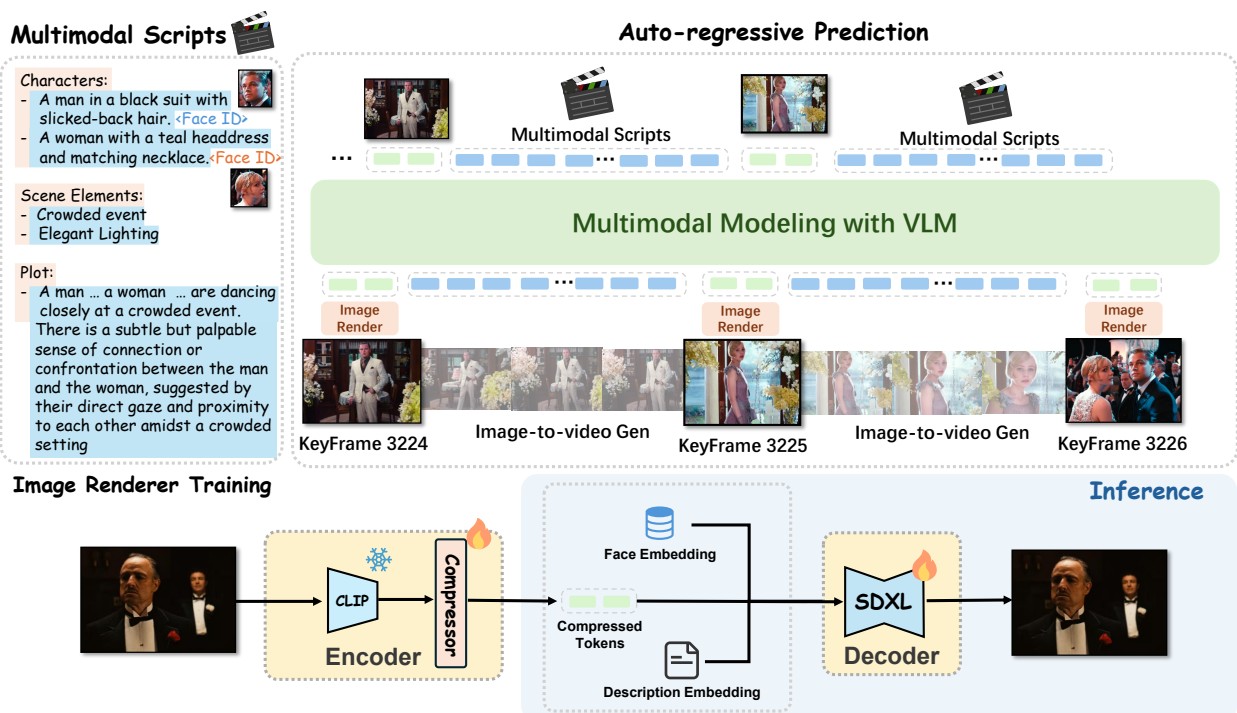

Figure 21: The framework of MovieDreamer. The autoregressive model takes multimodal scripts as input and predicts the tokens for keyframes. These tokens are then rendered into images, forming anchor frames for extended video generation. MovieDreamer ensures long-term coherence and short term fidelity in visual storytelling with the character's identity well preserved.

transition between frames is consistent. Later, to further improve the flexibility of long video generation, Brooks et al. (2022) generates low-resolution sequences, which are then enhanced to high resolution. Besides, CogVideo (Hong et al., 2023a) further interpolates non-key frames with the help of keyframes and text conditions with a two-stage-based framework. Another work by Zhang et al. (2023c) is built upon StyleGAN-v (Skorokhodov et al., 2022), which suffers from undesired content jittering for long video generation. They further proposed a novel B-Spline-based motion representation to ensure temporal smoothness.

**Diffusion models.** Different from traditional models, the solutions in diffusion models are more diverse and can be split into two main categories, *i.e.*, divide and conquer (sliding windows and keyframes), autoregressive models (including sliding windows), and consistency. For **divide-and-conquer**-based methods (EMO), the generation of long videos is split into several stages and for each stage, the video is generated based on the previous one or the input condition for refinement. NUWA-XL (Yin et al., 2023b) employs a hierarchical method for generating long videos. Specifically, a global diffusion model is first applied to generate the keyframes across the entire time range, and then local diffusion models recursively fill in the content between nearby frames. An earlier work (Harvey et al., 2022) proposes a RAG-style generation pipeline for generating adapted long video frames continued on any other videos along with a novel generation scheme (attention map) for not forgetting the early frames. Gen-L-Video (Wang et al., 2023b) also employs a hierarchical strategy to construct long videos through a set of short videos. FreeNoise (Qiu et al., 2024b), as an example, employs a sequence of noises for long-range correlation and performs temporal attention over them by sliding windows. Moreover, it also uses a motion injection method to condition the generation of long videos on multiple text prompts. Besides, instead of considering directly generating long videos, MAVIN (Zhang et al., 2024a) chooses to first generate several small clips and then generate the transition between them to maintain consistency with a boundary frame guidance module and a noise sampling strategy.

On the other side, autoregressive methods are drawing more and more research attention. For example, a sliding window, which can be seen as an application of autoregressive methods, is used to enable a short-video

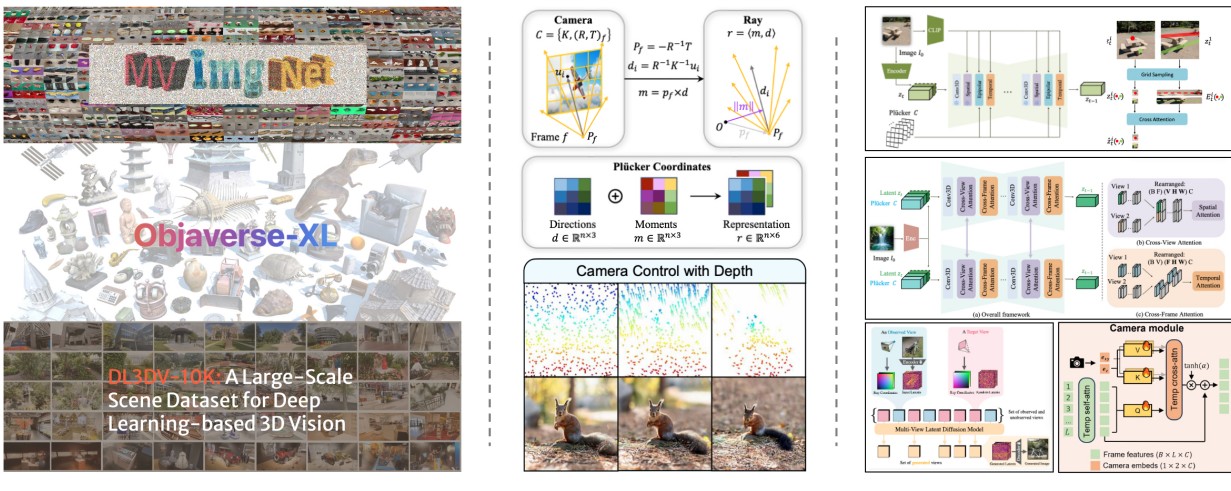

Figure 22: For a video diffusion model to be 3D aware, the general paradigm includes training on 3D dataset, adding camera control to the model, and introducing 3D-aware model architectures.

generator to generate long videos while the hidden representation of the video is interpolated when conditioning on different prompts. Moreover, many methods (Gao et al., 2024; Zhao et al., 2025; Xie et al., 2024a) are proposed with multimodal autoregressive models. MovieDreamer uses a multimodal autoregressive model with diffusion models to generate long videos conditioned on multimodal scripts. The multi-modal autoregressive model is designed to capture the key semantics in the multimodal conditions. (Xie et al., 2024a) further shows that existing models can be easily extended to autoregressive video diffusion models by varying the noise level instead of using a stable noise level.

Apart from these methods, more models are proposed to address the high computational requirements and the efficiency using diverse strategies and solutions (Diffusion Forcing). Specifically, to speed up the generation procedure, Video-Infinity (Tan et al., 2024b) is proposed to distribute the generation for one long video to multiple GPUs with the proposed clip parallelism and dual-scope attention modules. It is able to generate videos up to 2,300 frames in approximately 5 minutes with $8 \times$ Nvidia 6000 Ada GPUs (48G), which is 100 times faster than the current methods.

### 4.6   3D-aware video diffusion

The world exists in three spatial dimensions plus time, while a video represents a 2D projection of this 3D world. Consequently, video diffusion models should adhere to certain 3D constraints. These models generate multiple frames simultaneously and capture inter-frame relationships through attention layers spanning different frames. Existing research, such as SORA (Li et al., 2024i), demonstrates that by training on large-scale datasets, networks can implicitly learn 3D priors embedded within the data.

While this approach highlights a promising direction for learning 3D priors, there remains significant value in explicitly incorporating 3D awareness into video diffusion models. Doing so could improve 3D controllability and facilitate easier and more effective learning for the model.

### 4.6.1   Training on 3D dataset

Video diffusion models are typically trained on in-the-wild videos that feature arbitrary camera motions and dynamic scenes, making it challenging for these models to learn robust 3D priors. To enhance a model's ability for 3D generalization and multi-view consistency, one straightforward approach is to fine-tune the video diffusion model using datasets that explicitly capture 3D characteristics.

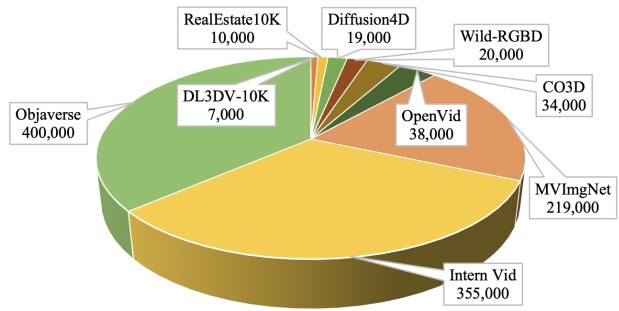

Figure 23: A visualization of the 3D datasets used by Cavia (Xu et al., 2024a).

SV3D (Blattmann et al., 2023a) fine-tunes a pre-trained stable video diffusion model on orbiting videos of 3D synthetic objects from the GSO (Downs et al., 2022) and OmniObject3D (Wu et al., 2023d) datasets. These orbiting videos include both static orbits, where the camera circles an object at regularly spaced azimuths with a fixed elevation, and dynamic orbits, where the camera trajectory is irregular. Similarly, V3D (Chen et al., 2024g) fine-tunes on the Objaverse dataset, leveraging 360-degree orbiting videos, along with proprietary datasets featuring similar setups (Junlin Han, 2024; Melas-Kyriazi et al., 2024).

Generative Camera Dolly (Van Hoorick et al., 2024) introduces two synthetic datasets, Kubric-4D and ParallelDomain-4D, which provide multi-view videos of dynamic scenes. By fine-tuning a Stable Video Diffusion (SVD) model solely on these synthetic datasets, they achieve zero-shot generalization to real-world novel view synthesis tasks.

CamCo (Xu et al., 2024b) fine-tunes a video diffusion model on real-world camera fly-through data from the WebVid10M dataset (Bain et al., 2021), using estimated camera poses. This enables the model to generate scene-level videos with precise camera controls. Similarly, CVD (Kuang et al., 2024) combines RealEstate10K (Zhou et al., 2018b) and WebVid10M (Bain et al., 2021) to train a camera-controllable video diffusion model.

Diffusion4D (LIANG et al., 2024) extends video diffusion into the realm of 4D generation by fine-tuning on orbiting videos where both the camera and object are in motion, enabling models to capture spatiotemporal dynamics more effectively.

Cavia (Xu et al., 2024a) focuses on camera-conditioned image-to-multi-view-video generation. They train their model on a large-scale dataset of approximately 0.4 million videos annotated with camera poses. These videos are sourced from a diverse collection of datasets, including Wild-RGBD (Xia et al., 2024), MVImgNet (Yu et al., 2023c), RealEstate10K (Zhou et al., 2018b), CO3D (Reizenstein et al., 2021), DL3DV-10K (Ling et al., 2024b), Objaverse (Deitke et al., 2022; 2023), Diffusion4D (LIANG et al., 2024), OpenVid (Nan et al., 2024), and InternVid (Chen et al., 2023b). A visualization of the dataset collections used in Cavia can be seen in Fig. 23.

HumanVid (Wang et al., 2024h) introduces a human-centric dataset enriched with camera annotations, enabling precise modeling of human-centric video generation.

KFC-W (Chou et al., 2024) introduces an auxiliary task of multi-view inpainting for training video models. This approach enables the model to learn 3D priors from large-scale, unposed internet photos.

CineMaster (Wang et al., 2025a) proposes a customized data labeling pipeline that leverages existing depth prediction (Yang et al., 2024e;f), instance segmentation (Ravi et al., 2024) and scene reconstruction (Zhang et al., 2024d) models to generate labeled data. This enables a 3D-aware model to train on large-scale, in-the-wild video data.

### 4.6.2 Architecture for 3D diffusion models

In addition to data-driven approaches for learning 3D priors, another promising strategy involves designing 3D-specific model architectures that incorporate inductive biases for generating view consistent images. Traditional U-Net based video diffusion models typically capture inter-frame relationships using temporal

attention layers, which are implemented through per-pixel attention along the temporal dimension. While this enables information flow between frames at the same spatial positions, it is suboptimal for achieving 3D awareness.

Recently, multi-view diffusion models (Shi et al., 2023d; Tang et al., 2023; Yang et al., 2024b; Liu et al., 2024b;f; Hu et al., 2024a; Kant et al., 2024; Yang et al., 2024d; Oh et al., 2023; Tang et al., 2024) have gained significant attention for static 3D scene generation. By employing cross-frame attention, these models effectively enhance the spatial relationships necessary for 3D consistency. For static 3D scene generation, epipolar-constrained attention (Xu et al., 2024b; Huang et al., 2023c; Zheng et al., 2024a) further enforces strict adherence to 3D constraints, ensuring greater structural accuracy.

This concept has also been extended to multi-view video diffusion models (Li et al., 2024a; Xie et al., 2024b; Xu et al., 2024a). By integrating both cross-frame and cross-view attention mechanisms, video models are fine-tuned to support 4D generation, enabling them to produce spatiotemporally consistent videos that adhere to 3D principles.

In DiT-like structures (Gao∗et al., 2024; Chou et al., 2024; Liang et al., 2024b; Gu et al., 2025; Wang et al., 2025a), cross-frame attention is inherently achieved as tokens for different temporal and spatial locations are jointly mixed through transformer, allowing the model to learn 3D priors directly from data.

### 4.6.3 Camera conditioning

Cameras serve as a bridge between the 3D world and 2D images. Given a 3D scene, intrinsic and extrinsic camera parameters determine how the 3D scene is projected into a 2D image. Adapting a video diffusion model to be 3D-aware involves conditioning the model with camera parameters, allowing users to control the generated views. Typically, fine-tuning is required after introducing this additional conditioning, as discussed in Sec. 4.6.1.

Several commonly used approaches exist for conditioning video diffusion models with camera parameters:

- **MLP-based embedding**. A straightforward method is to use a fully connected multi-layer perceptron (MLP) to embed camera parameters. These parameters are often represented as elevation and azimuth angles, as in SV3D (Voleti et al., 2024; Yang et al., 2024c), panning and zooming movement (Yang et al., 2024h), or as a relative camera extrinsic matrix (Van Hoorick et al., 2024; Wang et al., 2024k; 2025a). This embedding serves as a global condition, typically introduced into the diffusion model via cross-attention layers.

- **Ray map**. A camera defines a mapping from each pixel location in image space to a ray in 3D space. This mapping can be represented as a "ray map," which stores the ray origin and direction for each pixel location instead of RGB color. The ray map forms a spatially aligned representation with six channels, matching the resolution of an image. This method is particularly suited for scene-level video diffusion models requiring pixel-aligned conditioning (Gao∗et al., 2024; Watson et al., 2024; Wu et al., 2024e).

- **Plücker coordinates**. Plücker coordinates are an alternative to a ray map, parameterizing a ray as $(\mathbf{o} \times \mathbf{d}, \mathbf{o})$, where $\mathbf{o}$ is the ray origin, $\mathbf{d}$ is the normalized ray direction, and $\times$ denotes the cross product. This representation provides a uniform way to encode rays, making it a suitable positional embedding for video diffusion models (Xu et al., 2024b;a; Bahmani et al., 2024d;b; He et al., 2024a; Kuang et al., 2024; Yao et al., 2024; Wang et al., 2024h; Zheng et al., 2024a; Liang et al., 2024b).

- **Point-based**. Camera conditioning can also be achieved implicitly by motions of 2D or 3D points. For instance, I2VControl-Camera (Feng et al., 2024b) employs point trajectories in the camera coordinate system as control signals, enabling precise control over the generated views. Similarly, Motion Prompting (Geng et al., 2024a) converts camera poses into image-space point trajectories. This is achieved by projecting a 3D point cloud, estimated using a monocular depth estimator, onto 2D points in image space. Diffusion as Shader (Gu et al., 2025) leverages a tracking video of dense 3D points as control signals to train a video DiT model, enabling detailed control over both camera and motion controls.

### 4.6.4 Inference-time tricks

Interestingly, there are a few efforts being made to explore whether it is possible to directly use a pre-trained video diffusion model for 3D-aware video generation without any fine-tuning. ViVid-1-to-3 (Kwak et al., 2024) combines a video diffusion model with Zero-1-to-3 XL to jointly denoise multi-view videos, effectively leveraging the 3D priors from Zero-1-to-3 XL alongside the temporal consistency inherent in video diffusion models. NVS-Solver (You et al., 2024) guides the sampling process of a pre-trained Stable Video Diffusion (SVD) model by iteratively modulating the score function with the given scene priors represented derived from warped input views, effectively achieving novel view synthesis from sparse input views. CamTrol (Hou et al., 2024) directs video generation by employing DDIM inversion noise derived from point cloud renderings, enabling 3D-aware video synthesis without the need for model fine-tuning.

## 5    Benefits to other domains

### 5.1    Video representation learning

Video representation learning (Ravanbakhsh et al., 2024), as a foundational task in video understanding, it provides downstream tasks with powerful backbones and useful representations. Early methods (Seo et al., 2020; Wu et al., 2022a) primarily relied on 2D Convolutional Neural Networks (CNNs) (He et al., 2016; Krizhevsky et al., 2012) to extract spatial features from individual frames. These methods treated video data as sequences of static images, which limited their ability to capture temporal dynamics. To address this, researchers began incorporating Recurrent Neural Networks (RNNs) (Hopfield, 1982) and Long Short-Term Memory (LSTM) (Hochreiter & Schmidhuber, 1997) networks to model temporal dependencies across frames. However, these approaches (Liu et al., 2016; 2018) often struggled with capturing complex, long-range temporal relationships and integrating spatial-temporal information efficiently.

The development of 3D CNNs marked a significant advancement, enabling simultaneous modeling of spatial and temporal features by applying convolutions across both space and time dimensions. This shift allowed for more holistic video representation (Tran et al., 2015), leading to better performance in tasks such as action recognition (Yang et al., 2019; Kong et al., 2017) and video captioning (Yan et al., 2022). More recently, attention mechanisms and Transformers (Vaswani et al., 2017) have been introduced to further enhance video representation learning (Zhao et al., 2022; Gabeur et al., 2020; Cao et al., 2022; Botach et al., 2022; Wu et al., 2022b). These models, especially those utilizing self-attention, can capture long-range dependencies and complex interactions between frames, leading to more robust and generalizable video representations. The evolution of video representation learning has been characterized by a gradual shift from static image processing to dynamic, sequence-based modeling, reflecting the unique challenges of video data.

On the other side, video-language learning (Radford et al., 2021; Li et al., 2022) has been boosted by the integration of advanced models (Vaswani et al., 2017; He et al., 2016) and the increasing complexity of tasks that involve understanding both visual and textual information. Early attempts at video representation learning (Lei et al., 2021) relied heavily on separate processing of video and language inputs, where modality-dependent models were used to extract features from video frames and textual data independently. However, this approach often fails to capture the intricate relationships between the two modalities. The introduction of Transformer-based models (Ye et al., 2023) marked a significant advancement, enabling more effective modeling of cross-modal interactions through self-attention mechanisms. These models can jointly process video and language inputs (Zhao et al., 2023b), leading to improved performance on tasks, *e.g.*, video captioning (Li et al., 2023), video retrieval (Wang et al., 2024b), and video question answering (Pan et al., 2023).

As shown by previous works in image generation (Chi et al., 2024), a pretrained video generative model might be able to serve as prior, loss, or regularize in learning video representations. Recently, a pioneer work (Yeh et al., 2024a) employs image and video generative models for hateful visual content recognition. It queries well-trained SOTA image and video generative models to automatically scale up training data pairs for hateful visual content recognition, achieving superior performance.

## 5.2 Video retrieval

Video retrieval, mainly video-text retrieval (VTR), which involves cross-modal alignment and abstract understanding of temporal images (videos), has been a popular and fundamental task of language-grounding problems (Wang et al., 2020a;b; 2021; Yu et al., 2023b). Most existing conventional video-text retrieval frameworks (Yu et al., 2017; Dong et al., 2019; Zhu & Yang, 2020; Miech et al., 2020; Gabeur et al., 2020; Dzabraev et al., 2021; Croitoru et al., 2021; Liu et al., 2025b; Wang et al., 2025c) focus on learning powerful representations for video and text and extracting modality-dependent representations with modality-dependent encoders. Inspired by the success of self-supervised pretraining methods (Devlin et al., 2019; Radford et al., 2019; Brown et al., 2020) and vision-language pretraining (Li et al., 2020; Gan et al., 2020; Singh et al., 2022) on large-scale unlabeled cross-modal data, recent works (Lei et al., 2021; Cheng et al., 2021; Gao et al., 2021; Ma et al., 2022; Park et al., 2022; Wang et al., 2022a;b; Zhao et al., 2022; Gorti et al., 2022; Luo et al., 2022; Liu et al., 2022d;a; Cao et al., 2022) have attempted to pretrain or fine-tune video-text retrieval models (Gorti et al., 2022; Lei et al., 2021).

With the rapid progress in video generation, leveraging the advancement in video generation (Chen et al., 2024c; Muaz et al., 2023) to advance video retrieval by reranking (Zhou et al., 2024e; Dutta et al., 2021) or other methods becomes more and more promising. On the other side, retrieval augmented generation could also help faithful video generation (Arefeen et al., 2024; Rosa, 2024; Tevissen et al., 2024; He et al., 2023a).

## 5.3 Video QA and captioning

Video question answering (VideoQA) (Zhong et al., 2022; Wu et al., 2024a) is a task to predict the correct answer given a question and a video. Early work (Zeng et al., 2017; Xu et al., 2017; Zhao et al., 2017; Jang et al., 2017) focused on simpler factoid questions with the introduction of datasets like MSVD-QA and TGIF-QA. Techniques during this period primarily relied on attention mechanisms and memory networks to handle the spatial and temporal complexities of video data. Later, more challenging datasets emerged, such as NExT-QA, which emphasized inference and reasoning over merely recognizing visual facts. This shift necessitated the development of more advanced models, including Graph Neural Networks (GNNs) for relational reasoning and Transformers for their powerful sequence modeling capabilities, particularly when enhanced with cross-modal pre-training strategies.

Recently, the focus has increasingly shifted towards models that can handle multi-modal inputs and perform more complex reasoning tasks. Knowledge-based VideoQA datasets and techniques have also attracted more and more attention, pushing the boundaries of what VideoQA systems can achieve. Modular networks and neural-symbolic approaches are being explored to improve flexibility and interpretability, addressing the need for models that can reason about causal and temporal relationships in video content. On the other side, understanding long videos could be a big challenge to current models as long videos require a more comprehensive understanding among different frames and a long term memory.

In the future, pre-trained models originally designed for video generation can be used to enhance VideoQA by predicting future frames or generating hypothetical scenarios to answer complex questions. Also, text-to-video generation methods can be used for generating diverse VideoQA benchmarks.

## 5.4 3D and 4D generation

One emerging area of research focuses on generating 3D and 4D scene representations, such as textured meshes, Neural Radiance Fields (NeRF) (Mildenhall et al., 2021), and Gaussian Splatting (GS) (Kerbl et al., 2023). These representations hold significant potential for applications in 3D content creation and immersive virtual reality. However, due to the heterogeneous nature of 3D and 4D representations, and the scarcity of large-scale 3D and 4D datasets, it is difficult to design feed-forward neural networks capable of directly generating such representations.

To address these challenges, a common approach involves distilling knowledge from video generative models to reconstruct multiple viewpoints of a 3D scene. This strategy leverages advancements in video generation to synthesize complex 3D structures effectively. Recent studies have explored the potential of video diffusion

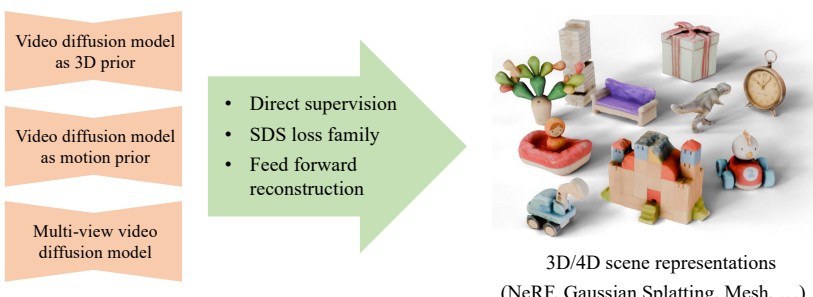

Figure 24: A common paradigm for 3D and 4D generation is by distilling 3D or motion knowledge from video generation models to 3D and 4D scene representation.

models in facilitating 3D and 4D generation, offering promising avenues for bridging the gap between video and high-dimensional scene representation.

### 5.4.1 Video diffusion for 3D generation

When generating 3D scenes from 2D models using multiple views, a major challenge is to maintain view consistency, which requires capturing inter-frame correlations across the generated views. Video diffusion models provide a general solution to inter-frame correlations. By treating the temporal dimension as different viewpoints, pre-trained video diffusion models can be fine-tuned into multi-view diffusion models. Several methods have been proposed to use a 2D video generation model for 3D scene generation:

V3D (Chen et al., 2024g) leverages the predictions of fine-tuned video diffusion models as direct supervision to generate 3D Gaussian splatting or meshes. SV3D (Voleti et al., 2024) employs the outputs of a fine-tuned Stable Video Diffusion model to supervise a Neural Radiance Field (NeRF) in the coarse stage, then uses SDS loss to refine high-frequency details of DMTet (Shen et al., 2021) initialized from the coarse stage. IM-3D (Melas-Kyriazi et al., 2024) adopts an instructive NeRF-to-NeRF (Haque et al., 2023) pipeline for distillation. This involves iteratively adding noise to rendered results and using a diffusion model to denoise the images, which then serve as targets for updating the 3D representation. VFusion3D (Junlin Han, 2024) demonstrates the potential of fine-tuned video diffusion models as data generators for creating large-scale 3D datasets. These datasets can be used to train feed-forward 3D generation models. Hi3D (Yang et al., 2024c) follows a coarse-to-fine 3D generation pipeline. It first generates low-resolution orbiting videos using a text-to-video model, then refines these into high-resolution orbiting videos with a video-to-video model. Both models are fine-tuned from pre-trained video diffusion models with camera condition extensions. The high-resolution orbiting videos are subsequently used to reconstruct 3D meshes. ReconX (Liu et al., 2024a) fine-tunes video diffusion models for sparse-view novel view synthesis, enabling efficient reconstruction of 3D scenes from limited input views. With recent advancements in large reconstruction models (LRMs) (Hong et al., 2023b; Zhang et al., 2024e), Wonderland (Liang et al., 2024b) demonstrates that by training a feed-forward 3D reconstruction model from the latent space of a video generative model, remarkable scene generation results can be achieved. Some examples are shown in Fig. 25.

### 5.4.2 Video diffusion for 4D generation

The incorporation of the additional time dimension in 4D generation significantly increases the complexity of developing and training feed-forward networks. Video diffusion models, functioning as motion generators, offer a promising solution by enabling the extraction of scene dynamics and motion information for 4D generation.

Several approaches leverage video diffusion models to train dynamic Neural Radiance Fields (NeRFs) or other 4D representations: MAV3D (Singer et al., 2023), 4D-fy (Bahmani et al., 2024c), Dream-in-4D (Zheng et al., 2024b), Align-Your-Gaussians (Ling et al., 2024a), and Animate124 (Zhao et al., 2023e) apply Score Distillation Sampling (SDS) loss along the temporal axis to train dynamic NeRFs using video diffusion models. These methods enable the models to learn motion dynamics directly from video diffusion. Vidu4D

Figure 25: By training a feed-forward reconstruction model from a video generative model, Wonderland Liang et al. (2024b) achieves remarkable 3D scene generation results.

(Wang et al., 2024g) and STAG4D (Zeng et al., 2024b) propose robust reconstruction pipelines to reconstruct dynamic Gaussian Surfels or Gaussian splatting with monocular videos generated by video diffusion models. PLA4D (Miao et al., 2024a) animates static 3D Gaussian splatting generated by 3D diffusion models, using video diffusion predictions combined with a motion alignment model. TC4D (Bahmani et al., 2024a) uses SDS loss to supervise dynamic NeRFs, incorporating a trajectory-conditioned video model that disentangles global and local motion components, facilitating the generation of large-scale motions.

Dynamic motion knowledge can also be combined with spatial 3D knowledge from multi-view diffusion models: EG4D (Sun et al., 2024a) use a combination of SVD and SV3D models to directly generate multi-view videos, which are then used to supervise a 4D Gaussian splatting. 4Real (Yu et al., 2024a) generates 4D videos by using video diffusion model to supervise fixed-viewpoint video and freeze-time video with SDS loss. The use of deformable 3D GS ensures the temporal and view consistency. DreamGaussian4D (Ren et al., 2023) employs a training-free video-to-video pipeline that uses pre-trained video diffusion models to refine the dynamic textures of generated mesh sequences.

Several approaches aim to fine-tune video diffusion models into 4D-aware diffusion models, enabling the unified learning of spatial and temporal information. Such a direct 4D generator could be further distilled into a 4D scene representation for better temporal and view consistency or real-time rendering.

Diffusion4D (LIANG et al., 2024) fine-tunes a video diffusion model conditioned on camera pose and timestamps to generate orbiting views of dynamic scenes, which are then used to optimize dynamic Gaussian splatting. 4Diffusion (Zhang et al., 2024c) and Animate3D (Jiang et al., 2024a) train multi-view video diffusion models capable of simultaneously generating videos from different viewpoints, and then train dynamic NeRFs with SDS loss. VideoMV (Zuo et al., 2024) trains a multi-view video diffusion model and a feed-forward Gaussian splatting reconstruction model that directly predicts GS from sparse views generated by the multi-view video diffusion model. Vivid-ZOO (Li et al., 2024a) reuses pre-trained 2D video and multi-view diffusion models to bridge the gap toward 4D generation.

SV4D (Xie et al., 2024b) and CAT4D (Wu et al., 2024e) train multi-view video diffusion models conditioned on monocular reference videos, enabling video-to-4D tasks, as shown in Fig. 26.

# 6 Ethical Considerations

While video generation has been largely advanced by high-quality datasets (LAION, 2023; Miech et al., 2019; Yang et al., 2024a; Chen et al., 2024d), advanced model architectures (Lee et al., 2022; Seo et al., 2022),

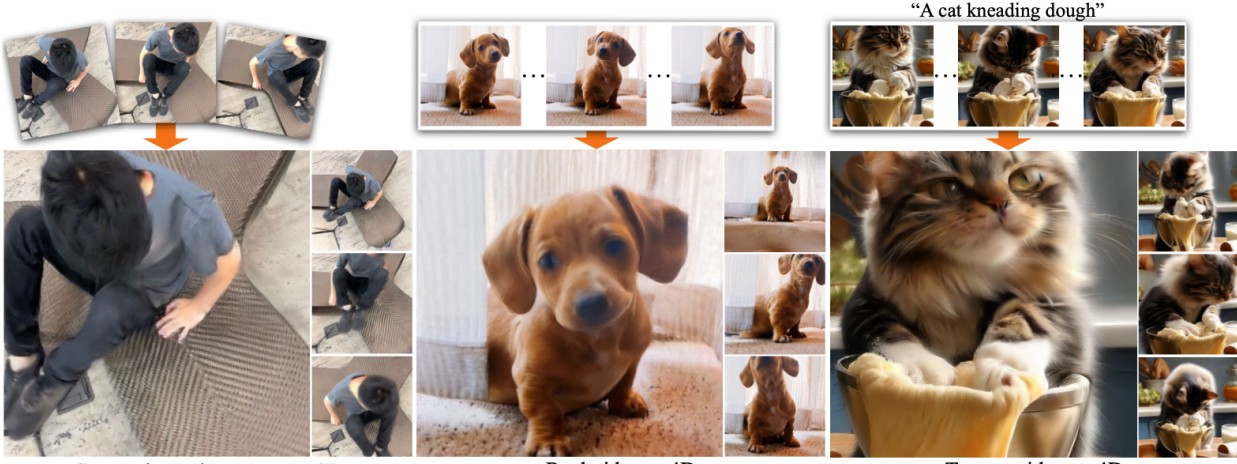

Figure 26: CAT4D (Wu et al., 2024e) trains a multi-view video diffusion model conditioned on a monocular video, which enables 4D view synthesis from casually captured monocular videos.

and training techniques (Wan et al., 2025; teamMeta, 2024), they (Wan et al., 2025; Zheng et al., 2024c; teamMeta, 2024) simultaneously raise critical ethical concerns that demand careful consideration.

**Deepfakes and Misinformation**. Creating realistic fake videos of real people without consent raises serious concerns about non-consensual intimate imagery (Zhao et al., 2021; Wu et al., 2022c; Rössler et al., 2018; Liu et al., 2025a), political manipulation, and spreading false information. This technology can undermine trust in authentic media and enable harassment campaigns. While image-based deepfake detection (Dang et al., 2020) has been successful, video deepfake video detection (Kundu et al.) remains a serious challenge to this area.

**Consent and Privacy**. Using someone's likeness, voice, or personal content to generate videos without explicit permission violates their privacy rights and autonomy (Rosenblat et al., 2025; Zhang et al., 2024f; Chen et al., 2024a). This extends to using training data that may contain copyrighted or private material without proper authorization.

**Bias and Representation**. AI models trained on biased datasets can perpetuate harmful stereotypes (Gallegos et al., 2024; Chen et al., 2024a), underrepresent certain groups, or generate content that reinforces discriminatory patterns. The bias in generated videos (Nadeem et al., 2025) could amplify existing social inequalities and marginalize vulnerable communities.

**Legal and Regulatory Challenges**. Current laws (Goanta et al., 2023) struggle to address AI-generated content, creating uncertainty around liability, intellectual property rights (Zhang et al., 2024f; Chen et al., 2024a), and regulatory compliance. This legal ambiguity can enable harmful uses while hindering beneficial applications.

**Transparency and Disclosure**. There's an ongoing debate about when and how AI-generated content should be labeled. Clear disclosure (Chang et al., 2024b; Yoo et al., 2024; Chen et al., 2024a) helps viewers make informed decisions about the content they consume, but enforcement remains challenging. While applying watermarks on generated video (Hu et al., 2024b) could be a potential solution, more and more approaches should be explored to emphasize these considerations.

**Quality Control and Safety**. Video generation systems may produce harmful, offensive, or dangerous content if not properly constrained. Ensuring robust content filtering (Poppi et al., 2024; Yoon et al., 2025; Liang et al., 2025b; Liu et al., 2025a) and safety measures is crucial to prevent the creation of videos that could cause harm or violate policies.

**Computational Resources and Environmental Impact**. Training and running video generation models requires significant computational power, leading to substantial energy consumption and environmental costs (Nag et al., 2024) that should be considered in the development and deployment of these technologies.

Despite that, researchers have also proposed benchmark datasets (Miao et al., 2024b; Pang et al., 2024; Chen et al., 2025b; Yeh et al., 2024b; Wang et al., 2025b) to access the safety of text-to-video generative models.

## 7 Conclusion and outlook

This survey presented a comprehensive analysis of diffusion-based video generation, tracing its evolution from traditional GAN-based approaches to current state-of-the-art methodologies. Through our systematic examination, we have highlighted how diffusion models have addressed many historical challenges in video generation while introducing new opportunities and considerations for future research. The field of diffusion-based video generation has demonstrated remarkable progress in generating temporally consistent and visually compelling videos.

Our analysis reveals several key insights: First, the transition from GAN-based methods to diffusion models has significantly improved training stability and output quality, though computational efficiency remains a crucial challenge. Second, the integration of diffusion models with various architectural innovations has enhanced temporal coherence and motion consistency, particularly in long video generation. Third, the application of video generation methods has broadened considerably, from low-level vision tasks to complex video understanding problems.

Despite these advances, several critical challenges remain:

- Computational Efficiency: Future research should focus on reducing the computational demands of both training and inference, particularly for real-time applications.

- Long-Video Generation: While progress has been made in handling long video generation, maintaining consistency and coherence in long videos remains challenging.

- Adhering Physical Rules: Ensuring generated videos adhere to physical laws and maintain realistic object dynamics requires continued attention.

- Ethical Considerations: As these technologies become more powerful, developing robust frameworks for addressing bias and preventing misuse becomes increasingly important.

Corresponding to these challenges, we anticipate several promising research directions:

- **Efficient Architectures and Training Strategies**. The computational demands of video generation models present a significant barrier to widespread adoption and real-time applications (Bolya & Hoffman, 2023; Li et al., 2024j; Kim et al., 2023b; Sun et al., 2025; Kahatapitiya et al., 2024). Future research should focus on developing token-level optimizations such as adaptive token selection and learned pruning mechanisms that can identify and eliminate redundant spatial-temporal information while preserving critical motion dynamics. Additionally, hierarchical generation frameworks that decompose video synthesis into coarse-to-fine stages can leverage temporal coherence to reduce computational overhead. Novel training paradigms including progressive training (Wan et al., 2025), curriculum learning for temporal sequences, and specialized attention mechanisms for video data offer promising avenues for achieving better quality-efficiency trade-offs.

- **Integration of Physical Priors and Domain Knowledge**. Current video generation models often produce visually plausible but physically implausible results (Kang et al., 2025a), limiting their applicability in domains requiring realistic motion and object interactions. Future research should systematically incorporate physical understanding through physics-informed architectures that explicitly model laws such as conservation of momentum, gravity, and collision dynamics (Bansal et al., 2024; Kang et al., 2025b; Bansal et al., 2025). For specialized applications such as medical video

generation (Cao et al., 2024; Li et al., 2024f) or autonomous driving simulations (Fu et al., 2024b), incorporating domain-specific knowledge through structured priors can enhance both realism and utility. This requires developing modular frameworks that can easily integrate external knowledge sources and differentiable physics simulators into the generation pipeline.

- **Hybrid Generative Approaches**. The integration of diffusion models with other generative frameworks (Li et al., 2025b; Xie et al., 2025) presents opportunities to leverage the complementary strengths of different approaches while mitigating their individual limitations. Future research should explore combinations of diffusion models with GANs (Wang et al., 2023j) for improved training stability and generation speed, autoregressive models (Lee et al., 2022) for better temporal coherence, and variational autoencoders (Pandey et al., 2022) for enhanced controllability. Additionally, investigating unified frameworks (Guo et al., 2025b; Zhang et al., 2025) that can seamlessly switch between different generative paradigms based on content requirements and computational constraints offers promising directions for more versatile video generation systems.

- **Model Safety and Ethical Considerations**. As video generation technology becomes more sophisticated and accessible, ensuring model safety and addressing ethical concerns (Miao et al., 2024b; Pang et al., 2024; Chen et al., 2025b; Yeh et al., 2024b; Wang et al., 2025b; Hu et al., 2024b) becomes increasingly critical, such as copyrights, data consent, privacy, and biases.

As the field continues to evolve, the relations between diffusion-based video generation and related domains, such as video understanding and 3D generation, will likely drive further innovations. The community's focus on addressing both technical and ethical challenges will be crucial in realizing the full potential of this technology while ensuring its responsible development and deployment.

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
