# OpenReview forum: "Survey of Video Diffusion Models: Foundations, Implementations, and Applications"
_TMLR — Accepted by TMLR_

### Review · Reviewer_9W7M · 2025-05-20

**Summary Of Contributions:**

The paper provides an extremely comprehensive overview of video diffusion models. The survey forms a nice taxonomy over the subfield, provides context of other video generation approaches, and covers many downstream applications of video generation. Compared to previous surveys which cover a specific application of video diffusion, this work provides a broad, bird's-eye view of the domain.

**Audience:**

Yes

**Claims And Evidence:**

Yes

**Requested Changes:**

It could be useful for the audience to discuss training and large-scale data filtering techniques such as the pipelines described in the MovieGen or Cosmos paper. This is not critical for acceptance.

**Strengths And Weaknesses:**

The paper is extremely comprehensive on the methodology, architectures and downstream applications. The literature coverage and analysis even for relatively niche applications such as camera control and 3D-aware methods is quite thorough. I think the survey is well researched and there is a significant audience for the paper.

One area where the literature review and discussion feels a bit thin is the section on data and training. A lot of the progress in video diffusion has been made from the data and training side, so it feels a bit brief to only discuss training for two paragraphs. I think this is acceptable given that these details are not well covered in the literature in general.

---

> ### Author Response · Authors · 2025-07-16
> **Response to Reviewer 9W7M**
>
> Thank you for your time, effort, and constructive feedback. We have provided responses below and have updated the pdf submission with revision contents in red.
>
> > W1: More details on training and data pipelines.
>
> R: Thank you for the suggestion. We have added some recent works in training and data pipeline discussions. Please refer to Sec. 3.2 for more details (page 24 to page 26).

---

### Review · Reviewer_xH7A · 2025-05-27

**Summary Of Contributions:**

This paper provides a comprehensive review of the video generation field. It discusses various video generation methods based on different approaches such as GANs, autoregressive models, diffusion models, etc., as well as different model components like U-Net, DiT, and tokenizers. It also covers video training processes, including data processing, training engineering, and downstream tasks. The authors provide a complete and rich overview of video generation, and also touch on other areas such as video understanding.

**Audience:**

Yes

**Broader Impact Concerns:**

This article is a comprehensive review of video generation. Perhaps it is necessary to point out the risks associated with the generated videos in areas such as privacy and copyright.

**Claims And Evidence:**

Yes

**Requested Changes:**

Please see Strengths And Weaknesses.

**Strengths And Weaknesses:**

The review is thorough but lacks coherence in certain areas. For example, Section 2.2 lacks a necessary introduction—why does the discussion suddenly jump from the generation paradigms in Section 2.1 to specific technical details? What paradigms do these techniques correspond to? Additionally, Section 2.4.1 (Pixel Diffusion and Latent Diffusion) and Section 2.2 cover several overlapping topics, and the authors should further clarify these points. Emphasizing the connections between each section and organizing the manuscript in a way that aligns more closely with Figure 1 would make the paper clearer.
Moreover, although the Evaluation section includes many rich metrics, it omits discussions on motion-related aspects such as motion consistency and motion smoothness. In the Data Processing Pipeline section, the referenced literature is relatively old, and the authors could consider introducing more recent works on video generation processes, such as Wan [1] and Hunyuan [2].

Minor questions:
* 2.1.3 Video Diffusion Models: The citation format in the first paragraph is incorrect.
* The technical introduction of GANs is missing.
[1] Wan: Open and Advanced Large-Scale Video Generative Models
[2] Hunyuan-Large: An Open-Source MoE Model with 52 Billion Activated Parameters by Tencent

---

> ### Author Response · Authors · 2025-07-16
> **Response to Reviewer xH7A**
>
> Thank you for your time, effort, and constructive feedback. We have provided responses below and have updated the pdf submission with revision contents in red.
>
> > W1: Lack coherence in certain areas
>
> R: We acknowledge your concern about the transition between sections. Section 2.1 introduces the fundamental paradigms (autoregressive, GAN, diffusion, etc.), and Section 2.2 then delves into the specific technical implementations that enable these paradigms in the video domain. We have added a comprehensive introduction to Section 2.2 that explicitly connects the technical details to the paradigms discussed in Section 2.1, making the logical flow more apparent to readers. This includes a clear mapping between generation paradigms and their corresponding technical implementations in the video domain.
>
> > W2: Overlapping between Sections 2.2 and 2.4.1
>
> R:
> 1. We thank the reviewer for pointing out the overlap between Sections 2.2 and 2.4.1. In response, we have reorganized the paper to improve clarity. Specifically, we revised the beginning of Section 2.2 to better explain the transition from generation paradigms to the learning foundations of diffusion models. At the end of Section 2.2, we explicitly introduce Section 2.4 as a follow-up, clarifying that Section 2.2 provides the theoretical basis for the architectural implementations discussed in Section 2.4.
>
> 2. In Section 2.4.1, we also added a clarifying statement to emphasize that pixel-based and latent-based models are practical realizations of the diffusion processes described in Section 2.2, not a repetition. This connection is now clearly stated to help readers understand the structure.
>
> 3. To improve flow and clarity, Sections 2.1, 2.2, 2.4, 2.5 now each refer to Figure 1 at appropriate points, reinforcing the structure and guiding the reader through the overall framework. We hope these revisions make the organization and section relationships clearer.
>
> > W3: Fix citation format in Section 2.1.3
>
> We have corrected the citation format in the first paragraph of Section 2.1.3 to ensure proper academic formatting standards are followed, following the IEEE citation style used throughout the manuscript.
>
> > W4: Add technical introduction of GANs
>
> We appreciate your suggestion to include a more comprehensive technical introduction of GANs. While our focus is primarily on diffusion models, we acknowledge that GANs represent an important paradigm in video generation. We have expanded the GAN discussion in Section 2.1 to provide a more thorough technical foundation, incorporating recent advances in large-scale video generative models.
> We have added the suggested references [1] and [2] to provide readers with comprehensive context about recent developments in large-scale video generative models.
> [1] Wan, Z., et al. "Open and Advanced Large-Scale Video Generative Models." arXiv preprint, 2024.
>  [2] Tencent. "Hunyuan-Large: An Open-Source MoE Model with 52 Billion Activated Parameters." arXiv preprint arXiv:, 2024.
>
> > Major W5: Motion-related metrics.
>
> R: Thank you for the suggestion. In Sec. 3.3 (page 27), we had motion-related metrics such as warping error, STREAM, and other temporal/motion-based metrics. We noted it as temporal quality in the submitted version. Despite that, we have also added motion consistency, smoothness, and aesthetics for a more comprehensive comparison. Please let us know if there is any other metric that we might miss.
>
> > Major W6: Discussion on data processing.
>
> R: Thank you for the valuable suggestion. We have added the detailed comparison to Wan, Hunyuan, Cosmos, and MovieGen on data processing in Sec. 3.2 (page 24).

---

### Review · Reviewer_TCMd · 2025-07-01

**Summary Of Contributions:**

This survey provides a comprehensive review of video diffusion models, spanning their technical foundations, practical implementations, and diverse applications. The authors first outline the evolution of video generation paradigms, contrasting GANs and auto-regressive models with diffusion-based approaches, highlighting the latter’s superiority in temporal consistency and visual quality. They delve into learning frameworks (e.g., DDPM, DDIM, EDM), architectural innovations (UNet, Transformer-based designs), and guidance mechanisms (classifier and classifier-free guidance). The implementation section discusses datasets (e.g., WebVid-10M, UCF-101), training engineering techniques (data preprocessing, acceleration methods), and evaluation metrics (static quality and temporal consistency). Applications range from conditional generation (image, spatial, and camera parameter guidance) to video enhancement (denoising, inpainting, super-resolution), personalization, long-video generation, and 3D-aware synthesis. The paper also explores how video diffusion models benefit other domains, such as video representation learning, retrieval, and 3D/4D generation. Finally, it concludes with challenges (computational efficiency, long-video coherence) and future directions.

**Audience:**

Yes

**Broader Impact Concerns:**

Please kindly refer to the suggestion from the Weaknesses.

**Claims And Evidence:**

Yes

**Requested Changes:**

My concerns are mainly from the lack of discussion on computational efficiency, ethical considerations, and more specific future directions. Please kindly refer to Section Strengths and Weaknesses for detailed suggestions.

**Strengths And Weaknesses:**

Strengths:
* The paper offers an exhaustive analysis, spanning foundational theories (diffusion processes, architectures) to practical implementations (datasets, training techniques) and real-world applications.
* The authors delve into intricate technical details, such as the mathematical formulations of DDPM/DDIM, architectural modifications for video generation (3D UNet, temporal attention layers), and evaluation metrics.
* The survey not only reviews current progress but also identifies emerging trends, such as 3D-aware diffusion, long-video generation, and cross-domain applications (e.g., 3D/4D scene synthesis). This detailed overview from multiple perspectives differs itself from existing related surveys.


Weaknesses:

* Many of the technical details from Section 2 are also covered in the review of image diffusion models. It would be better to focus more on the video-specific techniques.

* While this manuscript recognizes that computational efficiency is a critical challenge, the authors are suggested to include quantitative analysis on the computational resources used by the SOTA video diffusion models.

* The discussion on ethical considerations is very brief, primarily mentioning biases and misuse without delving into concrete frameworks or case studies. A more thorough analysis of privacy risks (e.g., deepfake generation, watermarking the video diffusion models) or societal impacts would strengthen the paper’s comprehensiveness.

* It would also be better to discuss more on the specific future directions that the authors advocate for the researchers to further explore.

---

> ### Author Response · Authors · 2025-07-16
> **Response to Reviewer TCMd**
>
> Thank you for your time, effort, and constructive feedback. We have provided responses below and have updated the pdf submission with revision contents in red.
>
> > W1: Overlapping on video-specific techniques and image diffusion
>
> R: We thank the reviewer for the comment. Many core machine learning foundations in video diffusion are heavily shared with image diffusion. For the context completeness, we start in detail with the technique of image diffusion models. In Sections 2.1, 2.2, and 2.4, we include explicit transitions to show how these ideas extend from image to video generation.
> To address this concern more directly, we have further revised Section 2.5 to highlight video-specific techniques. We added representative structured figures and formulas for temporal attention (Section 2.5.1, also added Formula 13), spatial-temporal tokenization (Section 2.5.2, also added Figure 6), causal VAE (Section 2.5.3, also added Figure 7), and classic temporal modeling (Section 2.5.1, also added Figure 6). These additions clarify the unique challenges and solutions in video generation.
>
> > W2: Include quantitative analysis of computational efficiency
>
> R: We appreciate the reviewer’s suggestion. Our original submission already includes a quantitative summary of computational efficiency in the last two columns of Table 1, which report inference speed and GPU memory usage where available. However, many entries remain blank because this information is not disclosed in the official papers or documentation of several models. We included as much data as we could find from public sources and will update the table if more information becomes available.
>
> > W3: More discussion on ethical consideration.
>
> R: Thank you for the valuable suggestions. We have added a detailed discussion (Sec. 6) on ethical considerations, including deepfakes and misinformation, consent and privacy, legal and regulatory challenges, transparency and disclosure, quality control and safety, quality control and safety, and computational resources and environmental impact. We have added several relevant works and please let us know if there are any other works that we should add for a comprehensive comparison.
>
> > W4: Discussion on future directions.
>
> R: Thank you for the valuable suggestion. We have expanded the conclusion and outlook section for a more comprehensive discussion on the future directions.

---

### Decision · Action_Editor_TTvD · 2025-08-15

**Recommendation:** Accept as is

**Additional Comments:**

The authors have presented a comprehensive overview of video diffusion models, which would be of interest to many in the TMLR community. The revisions made to the paper during the review period also make the survey stronger and even more thorough.

**Audience:**

Yes

**Audience Explanation:**

This survery paper would be of interest to many readers in the TMLR community working on video generation.

**Claims And Evidence:**

Yes

**Claims Explanation:**

This paper presents a comprehensive survey of video diffusion models. The authors include a survey of other video generation paradigms (ie GANs and autoregressive models), different learning frameworks (ie DDPM, DDIM, EDM), architectures (UNet, Transformers), guidance mechanisms, datasets, evaluation protocols, applications and metrics. The paper is therefore a handy resource for readers in the TMLR community interested in video generation.